# OrbitZoo: Real Orbital Systems Challenges for Reinforcement Learning

**Alexandre Oliveira**
NOVA University of Lisbon
Caparica, Almada, Portugal
aan.oliveira@campus.fct.unl.pt

**Katarina Dyreby**
NOVA University of Lisbon
Caparica, Almada, Portugal
k.dyreby@campus.fct.unl.pt

**Francisco Caldas**
NOVA University of Lisbon
Caparica, Almada, Portugal
f.caldas@campus.fct.unl.pt

**Cláudia Soares**
NOVA University of Lisbon
Caparica, Almada, Portugal
cam.soares@fct.unl.pt

## Abstract

The increasing number of satellites and orbital debris has made space congestion a critical issue, threatening satellite safety and sustainability. Challenges such as collision avoidance, station-keeping, and orbital maneuvering require advanced techniques to handle dynamic uncertainties and multi-agent interactions. Reinforcement learning (RL) has shown promise in this domain, enabling adaptive, autonomous policies for space operations; however, many existing RL frameworks rely on custom-built environments developed from scratch, which often use simplified models and require significant time to implement and validate the orbital dynamics, limiting their ability to fully capture real-world complexities. To address this, we introduce OrbitZoo, a versatile multi-agent RL environment built on a high-fidelity industry standard library, that enables realistic data generation, supports scenarios like collision avoidance and cooperative maneuvers, and ensures robust and accurate orbital dynamics. The environment is validated against various real satellite constellations, including Starlink, achieving a Mean Absolute Percentage Error (MAPE) of $0.16\%$ compared to real-world data. This validation ensures reliability for generating high-fidelity simulations and enabling autonomous and independent satellite operations. This project is open source[1] and has a dedicated project page[2].

## 1 Introduction

Since the dawn of the space age in 1957, humanity has successfully launched approximately 20,000 satellites into Earth's orbit [23], of which only about 50% remain operational. These satellites are crucial for our daily life, providing critical services such as global communication, navigation, weather forecasting, Earth observation, and scientific research. However, these advances come with many problems.

Earth's orbital environment hosts an estimated 140 million debris objects, with approximately 1 million of these being larger than 1 cm – large and fast enough to cause catastrophic damage upon impact [22]. Collisions with space debris generate more debris, leading to further collisions. This chain-reaction known as the Kessler syndrome, results in an exponential increase in debris, threatening

---

[1]https://github.com/orbitzoo/orbit_zoo
[2]https://orbitzoo.github.io/

39th Conference on Neural Information Processing Systems (NeurIPS 2025).

the long-term sustainability of Earth's orbits [39]. If no further measures are taken to address this issue, Earth's orbits could become unusable.

More recently, the congestion in Lower Earth Orbit (LEO) is likely to undergo a major change of scale for mega-constellations of telecommunication satellites, leveraging tens of thousands of satellites [9]. The increasing density of satellites and debris, particularly in LEO [11], presents formidable challenges for Space Traffic Management (STM), related to the accurate monitoring and tracking of space objects and development of adequate decision-support frameworks [47], compounded by issues of data scarcity and uncertainty. Despite the growing availability of orbital data and tools, current solutions for satellite maneuvers – ranging from orbital transfers to collision avoidance – remain heavily reliant on manual processes. Human experts must navigate an ever-increasing complexity of scenarios, making critical decisions under time constraints with incomplete or inaccurate information [26]. As the volume of satellites and orbital debris continues rising, these traditional methods are quickly becoming unsustainable, which demands the development of new strategies to minimize our impact, such as debris removal [7] and faster, more capable, and autonomous intelligent systems for decision-making. Adaptive control systems [4] provide robustness to uncertainty but struggle with complex, dynamic, and nonlinear space systems. RL, on the other hand, excels in real-time adaptation to such environments [77].

We examine how RL applications to orbital dynamics are shaped by two key factors and limited by a lack of standardization. First, the choice of data-generation tools affects the trade-off between simplicity and realism. Second, RL mission frameworks are often built from scratch, requiring extensive validation of orbital dynamics. To address these challenges, we introduce OrbitZoo, an environment that combines fast, high-fidelity orbital data generation with RL development. The contributions of this paper fall into three categories:

- **Data Generation**: Built on Python and with a robust space dynamics library on its background, OrbitZoo generates high-fidelity orbital data by incorporating realistic forces and perturbations, providing accurate datasets essential for machine learning and strategy validation, while making use of parallel computations for fast propagation;

- **Reinforcement Learning**: OrbitZoo is standardized for RL research, leveraging the Petting-Zoo library [76] to support multi-agent RL (MARL) with a Partially Observable Markov Decision Process (POMDP) structure, while also making use of the parallelism provided by the space dynamics library to properly scale to systems with thousands of bodies. This integration enables the development, training, and benchmarking of intelligent satellite maneuvering strategies for single and multi-agent missions in cooperative, competitive, or mixed scenarios;

- **Customizable Framework with Visualization**: OrbitZoo's modular design allows users to define scenarios with an arbitrary number of bodies with or without thrusting capabilities, incorporate custom models, and adapt the environment to specific needs, with clear separation of abstraction levels. It also features an interactive 3D visual component, making it versatile for a wide range of applications while providing an accessible way to understand orbital behaviors and decision-making processes.

## 2    Related Work

Existing efforts in this domain can be broadly categorized into three areas: high-fidelity tools for **generating and simulating orbital dynamics data**, **RL for orbital maneuvering**, and **MARL environments for orbital dynamics**.

**Data Generation and Simulation Tools.**    Orekit [45] is one of the most comprehensive open-source libraries for astrodynamics and orbital mechanics, offering advanced capabilities for precise orbit determination [46, 58, 16], propagation and associated uncertainties [6], attitude determination, and trajectory analysis [19]. Its modular, Java-based architecture allows users to model systems ranging from simple Newtonian attraction to highly complex scenarios incorporating detailed gravity fields and perturbations, such as atmospheric drag, solar radiation pressure (SRP), and third-body effects. While versatile, Orekit's steep learning curve and reliance on Java can pose challenges for users unfamiliar with its technicalities.

Poliastro [62], a Python-native library, offers a more accessible alternative, featuring tools for orbit propagation and transfer planning. Although it lacks the high-fidelity modeling of Orekit, Poliastro integrates with the Cesium library [17] for accurate 3D geospatial visualization, making it suitable for simpler scenarios or users with limited expertise in orbital mechanics. Systems Tool Kit (STK) [5], the leading commercial orbital simulation software, offers extensive capabilities for mission planning and operational analysis. Despite its advanced visualization features and high precision, STK's substantial cost restricts its accessibility to well-funded organizations.

**Reinforcement Learning for Orbital Dynamics.**   RL has been widely applied to satellite maneuvering, often relying on custom-built environments and simplified dynamical models. Many studies use the Circular Restricted Three-Body Problem (CR3BP) [49, 24, 73, 41, 12] for tractable simulations, though it falls short in capturing real-world perturbations, making it challenging for non-experts to extend RL to realistic space missions. For **LEO station-keeping**, [31] implemented Proximal Policy Optimization (PPO) [69] with a 4th-order Yoshida integrator, modeling gravitational forces and atmospheric drag, while [75] used Soft Actor-Critic (SAC) [28] with dynamics developed from scratch that also included third-body forces and SRP.

Orekit has also been employed in RL frameworks with more accurate dynamics. [40], for instance, used Orekit and Deep Deterministic Policy Gradient (DDPG) [43] for **low-thrust transfers**. In **geostationary orbit (GEO)**, [15] used Orekit and a synchronous variant of A3C [50] for a perigee-raising maneuver. Despite these contributions, all relied on simplified or mission-specific physics without multi-agent or uncertainty support, limiting their generalization to more complex scenarios.

**Collision Avoidance Maneuvers (CAMs)** have received growing attention. [38] introduced ColAv-Gym, a single-agent CAM environment in LEO using Orekit and PPO. It leverages real Conjunction Data Messages (CDMs) to retroactively reconstruct satellite and debris orbits for training. In contrast, [72] and [13] employed synthetic debris scenarios in Gym-based environments incorporating Orekit and an adaptation of SpaceNav [27]. These studies investigated different observation and reward strategies, modeling the problem as a POMDP with temporal features captured by Long Short-Term Memory (LSTM) [32] layers in the Deep Q-Network (DQN) [51] architecture, showcasing how RL algorithms with discrete action spaces and recurrent networks can be useful in such scenarios.

**Multi-Agent Reinforcement Learning Environments.**   MARL has recently gained traction in orbital dynamics, with applications in satellite coordination and task assignment, although with limited focus on maneuvers. [34] introduced the RL-Enabled Distributed Assignment (REDA) algorithm, using Poliastro and DQN for task allocation across satellite constellations. However, REDA lacked integration with realistic orbital data. Similarly, [84] applied Multi-Agent PPO (MAPPO) [83] for multi-satellite observation planning, combining STK and MATLAB for a custom simulation framework but without leveraging industry-standard astrodynamics libraries like Orekit. [60] developed a MARL framework for cooperative formation flying of multiple spacecraft considering only Newtonian attraction, while [20] investigated multi-agent CAM using a framework [54] that aggregates local neighborhood information through a graph neural network (GNN) [64], considering two dynamic models [61, 80] that did not account for SRP and third-body forces.

## 3   Background: Challenges in Multi-Agent RL for Orbital Dynamics

A major challenge in applying RL to orbital dynamics lies in bridging the Reality Gap between simulation and real-world operations. This requires incorporating high-fidelity dynamics, accounting for uncertainties, and validating simulations against real-world data. According to what was discussed in Sec. 2, the essential capabilities of an orbital RL environment include: support for multi-agent interactions (both cooperative and competitive scenarios), integration with industry-standard simulators such as Orekit or Poliastro (rather than custom-built models), high-fidelity dynamics with gravity fields and perturbations, support for algorithms with continuous control, realistic body and thrust modeling (capturing distinct spacecraft characteristics and continuous thrust integration), interactive real-time visualization, and public code availability.

As shown in Tab. 1, **OrbitZoo** distinguishes itself by containing all the aforementioned capabilities. Unlike previous environments based on simplified models, OrbitZoo incorporates multiple perturbative forces, supports Cartesian, Keplerian, and equinoctial representations, enables realistic body and thrust modeling through diverse integration methods, propagates uncertainty, and offers several

Table 1: Comparison of RL environments for orbital dynamics.

| Work | Multi-Agent RL | Industry-Standard Simulator | High-Fidelity Dynamics | Continuous Control | Realistic Bodies and Thrust | Interactive Visualization | Publicly Available |
|---|---|---|---|---|---|---|---|
| Kolosa (2019) [40] | | ✓ | | ✓ | ✓ | | ✓ |
| Miller (2019) [49] | | | | ✓ | | | |
| Herrera (2020) [31] | | | ✓ (drag only) | ✓ | ✓ | | ✓ |
| Federici (2021) [24] | | | | ✓ | | | |
| Sullivan (2021) [73] | | | | ✓ | ✓ | | |
| Casas (2022) [15] | | ✓ | ✓ | ✓ | ✓ | | |
| Bonasera (2022) [12] | | | ✓ | ✓ | | | |
| Dolan (2023) [20] | ✓ | | ✓ (J2 only) | ✓ | ✓ | | |
| Bourriez (2023) [13] | | ✓ | ✓ | | ✓ | | |
| LaFarge (2023) [41] | | | ✓ (third bodies only) | ✓ | ✓ | | |
| Zhang (2023) [84] | ✓ | ✓ | | ✓ | | | |
| Qingyu (2023) [60] | ✓ | | | ✓ | ✓ | | |
| Holder (2024) [34] | ✓ | ✓ | | ✓ | | | |
| Solomon (2024) [72] | | ✓ | ✓ | ✓ | ✓ | | |
| Kazemi (2024) [38] | | ✓ | ✓ | ✓ | ✓ | | |
| **OrbitZoo (ours)** | ✓ | ✓ | ✓ | ✓ | ✓ | ✓ | ✓ |

methods to assist in the development of RL missions or multi-agent systems analysis. Its publicly available nature further establishes it as a comprehensive platform for advancing learning-based space operations.

### 3.1 Multi-Agent Reinforcement Learning

RL provides a framework for decision-making under uncertainty, where agents learn through trial and error to maximize cumulative rewards [74, 70]. In the context of MARL, multiple agents operate in a shared environment, each pursuing individual or collective goals. MARL presents unique challenges in orbital dynamics, where environments are partially observable, highly dynamic, and governed by complex physical interactions.

MARL builds upon Markov Decision Processes (MDPs) [74, 10] and Partially Observable Markov Decision Processes (POMDPs), commonly used to model scenarios where agents cannot access the full environment state [37]. A POMDP is defined by the tuple $\langle S, A, P, R, O, \Omega \rangle$. At each timestep $t$, an agent observes $o_t \in O$ based on the current state $s_t \in S$, selects an action $a_t \in A$, and transitions to a new state $s_{t+1}$ with probability $P(s_{t+1}|s_t, a_t)$, receiving a reward $R(s_t, a_t)$. The goal is to learn a policy $\pi(a_t|o_t)$ that maximizes expected returns over time.

The introduction of Deep Q-Networks (DQN) [51] revolutionized RL by combining deep function approximation with Q-learning, making it possible to tackle discrete decision-making tasks from raw, high-dimensional observations. This success established the foundation for subsequent methods that aimed to generalize deep RL to continuous domains. Deep RL algorithms such as DDPG [43] and PPO [69] have been instrumental in this extension, albeit the topic had been explored earlier [71]. In MARL, the interaction between agents is often modeled as a multi-agent POMDP (MA-POMDP), where each agent $i$ observes $o_t^i$, performs an action $a_t^i$, and receives a reward $r_t^i$. PPO and variants, such as MAPPO [83], are well-suited for these scenarios, allowing a stable decentralized or centralized training of agents in shared environments. This is particularly important for missions involving constellations of satellites, where each satellite may need to coordinate, compete, or cooperate with others to achieve shared objectives in complex – and potentially noisy – orbital dynamics.

### 3.2 Challenges in Bridging the Reality Gap

Simulating orbital dynamics presents a substantial reality gap, driven by the need for high-fidelity physical modeling and the incorporation of real-world uncertainties. This section highlights the main challenges of applying RL to orbital dynamics, whose understanding relies on the fundamental concepts presented in Appendix B.

**Complex Orbital Dynamics.** Orbital dynamics follow Newton's laws of motion and gravitation, requiring accurate modeling of perturbative forces such as atmospheric drag, SRP, and third-body effects. Simplified propagators like the Simplified General Perturbations (SGP) model are computationally efficient but lack the precision required for RL tasks. In contrast, Orekit's numerical propagators [45] provide high-fidelity simulations by incorporating complex perturbations, enabling realistic long-term trajectory prediction and maneuver planning. The choice of integration method in numerical propagation also affects realism: classical approaches such as Gauss' variational equations (GVEs) and fixed-step integrators like the 4th-order Yoshida or RK4 [14] are commonly used for orbital propagation due to their simplicity and efficiency, but they lack the adaptive precision of higher-order variable-step schemes such as Dormand-Prince [21].

**Coordinate Systems and State Representations.** Orbital states are commonly represented using Keplerian elements, but these suffer from singularities in cases like circular or equatorial orbits. Equinoctial elements avoid such issues and are often preferred in RL applications for their robustness. However, Cartesian coordinates remain essential not only for numerical integration but also for tasks like computing inter-body distances or collision probabilities. An RL framework should therefore support all major representations. In multi-agent settings, maintaining consistent reference frames and relative state representations is particularly challenging, as agents may operate in different orbital regimes.

**Thrust Modeling and Control.** Thrust actions are often defined in local reference frames like RSW (Radial, Along-track, Cross-track). A polar parametrization – using thrust magnitude, deviation angle, and azimuthal angle – provides a realistic and constrained action space, offering flexibility in modeling satellite maneuvers. This representation can be converted to the RSW frame for compatibility with numerical propagators. In multi-agent settings, additional challenges arise as each spacecraft may have distinct thrust capabilities, fuel constraints, and control dynamics, making coordinated maneuver planning and learning more complex.

**Exploration in High-Dimensional Action Spaces.** Exploration in continuous action spaces, such as those encountered in orbital dynamics, presents significant challenges, particularly in multi-agent settings where agents may operate in joint state and action spaces with cooperative or adversarial interactions. To address this complexity and the wide range of possible learning objectives, it is essential to design a modular framework that supports both on- and off-policy RL methods, as well as discrete and continuous action spaces. Such flexibility enables consistent experimentation and comparison across diverse mission scenarios. Moreover, having the capability to handle multi-objective RL (MORL) [30] is important, as spacecraft missions often involve trade-offs (e.g., fuel vs. time vs. risk) that must be balanced within the policy learning process while coordinating multiple agents.

**Ephemeris Data and Validation.** Ephemerides offer time-stamped predictions of orbital positions and velocities, enabling validation of simulated trajectories. For instance, Orekit's numerical propagator demonstrates high accuracy when compared with Starlink satellite ephemeris data. By tuning physical parameters – such as drag and reflection coefficients – the simulation closely matches real-world behavior, helping bridge the reality gap in RL applications.

### 3.3 Multi-Agent Coordination and Scalability

In MARL for orbital dynamics, agents operate in partially observable environments with decentralized policies. While MARL algorithms commonly assume fully cooperative scenarios (Dec-POMDP), in reality agents often assume partial or fully competitive scenarios (MA-POMDP). In such cases, task coordination, interference avoidance, and scaling to large constellations, that is, groups of satellites working together to accomplish shared objectives, become key challenges. EPyMARL [57] extends PyMARL for flexible agent collaboration in MARL environments, while MARLlib [36], compatible with PettingZoo environments like OrbitZoo, supports a broad set of algorithms and architectures. Federated Learning (FL) and Federated RL (FRL) [59] further enable decentralized learning and coordination in both cooperative and competitive settings while maintaining information privacy. Nonetheless, challenges inherent to the space environment – such as competition, communication constraints, uncertainty, and heterogeneity between agents – make it challenging to apply standard MARL algorithms, highlighting the need for new, tailored approaches.

Table 2: Summary of reinforcement learning challenges and how OrbitZoo addresses them.

| Research Gap | Limitations of Standard Environments | Enabled by OrbitZoo |
| --- | --- | --- |
| Sim-to-Real Transfer | Neglects real-world orbital perturbations and environmental uncertainties. | Realistic dynamics with harmonic gravity fields, drag, SRP, third-body forces, uncertainties and variable-step integration methods. |
| Continuous Control | Impulsive or unconstrained control models. | Time-continuous thrust with fuel and actuation constraints. |
| Multi-Agent Coordination | Mostly single-agent, fully observable setups. | Decentralized agents with partial observability via PettingZoo. |
| Reward Design | Dense, artificial reward signals. | Flexible, physically grounded reward functions. |
| Safety and Adversarial Learning | Rarely models safety or adversarial dynamics. | Supports pursuit–evasion and safe maneuvering tasks. |
| Visualization | Limited (2D) or no visualization for debugging. | Real-time 3D inspection of agent behavior. |

Recent solutions leverage C++ engines [85] or CPU/GPU parallelization for faster training [63] and propagation [42] (see Appendix B.3). Scalability remains challenging when using high-fidelity numerical propagation, as each body experiences distinct forces that evolve continuously over time.

## 3.4 Realistic Simulation Environments

Realistic orbital dynamics environments should integrate (1) high-fidelity dynamics that account for the most relevant forces and perturbations (e.g., gravity fields, drag, SRP, third-body effects), (2) body and thruster specific characteristics, including shape and dimensions, reflection and drag coefficients, and specific impulse, (3) flexible state and action spaces, as different missions may require distinct representations, (3) multi-agent reinforcement learning and federated reinforcement learning capabilities, (4) scalability and parallelization to handle an increasing number of bodies, and (5) reproducibility and extensibility. However, most existing MARL environments for orbital dynamics do not fully integrate these capabilities (Tab. 1).

Beyond the orbital domain, similar challenges have been addressed in other fields where partial observability, decentralized control, and coordination are critical. In energy systems, MARL has been used to optimize distributed generation and communication under uncertain and partially observable states [56]. In autonomous robotics, macro-action (joint policies) and communication-aware learning frameworks have been proposed to improve coordination under local observation constraints [81, 55]. Similarly, unmanned aerial vehicle (UAV) swarm control has used graph-based methods to address scalability and partial observability in large-scale UAVs coordination and confrontation tasks [82]. These efforts highlight the growing importance of standardized, high-fidelity benchmarks across domains.

## 4 Novel Reinforcement Learning Challenges in OrbitZoo

In the emerging era of autonomous systems operating in the physical world, RL is evolving beyond synthetic benchmarks and addressing real-world constraints such as actuation limits, uncertainty, safety, and coordination. OrbitZoo is designed to support this shift. It provides a modular testbed to rigorously investigate RL challenges grounded in high-fidelity dynamics while abstracting just enough to enable algorithmic insight. Tab. 2 summarizes the main RL challenges addressed by OrbitZoo, which are discussed in detail below.

**Sim-to-Real Transfer and Grounded Dynamics.** Sim-to-real gaps constitute a significant barrier in deploying RL policies in physical systems. While Orekit handles high-fidelity propagation,

OrbitZoo provides the interfaces needed to test RL algorithms in scenarios with uncertain initial conditions, realistic gravity fields (Holmes–Featherstone harmonics [35]), drag (computed from historical weather data), SRP (accounting for the occlusion of the Sun by the Moon), and third-body forces (from all planets in the solar system, as well as the Sun, Moon, and the barycenters of the Earth–Moon system and the solar system). Validation against Starlink ephemerides (Sec. 5) shows that these perturbations can be tuned to match real trajectories, allowing research on robust policy learning under realistic dynamics.

**Continuous Control and Thrust Constraints.** OrbitZoo supports physically plausible, time-continuous control through polar thrust parameterizations and constant-magnitude thrusts. Unlike many benchmarks that assume instantaneous or impulsive maneuvers, OrbitZoo models thrust as a continuous acceleration integrated over time, constrained by available fuel and propulsion limits. This formulation forces agents to learn realistic, sustained maneuvers, making the environment well suited for studying constrained RL and planning under resource limitations. Classical maneuvers with simplified dynamics and impulsive actions can still be simulated in OrbitZoo (Appendix E.4).

**Multi-Agent Coordination with Partial Observability.** Real-world orbital operations often involve multiple autonomous spacecraft interacting under limited information – for example, in formation flying, debris tracking, or pursuit/evasion scenarios. Each agent must make decisions based only on local or delayed observations, while the overall behavior emerges from their collective interaction. OrbitZoo enables such decentralized and partially observable setups through its PettingZoo-based architecture, where each agent receives its own observation and action space while sharing the same dynamic environment.

This design enables the study of cooperative and competitive MARL paradigms. Cooperative tasks include coordinated thrusting for formation maintenance or rendezvous, while adversarial tasks involve pursuit, evasion, or interference. OrbitZoo also supports centralized training with decentralized execution (CTDE), where a global critic accesses full system information during learning, but agents act autonomously at runtime. With support for parallel trajectory collection, OrbitZoo offers a scalable testbed for studying coordination, communication, and robustness in multi-agent orbital control – areas largely unexplored in standard RL benchmarks.

**Reward Shaping under Sparse and Delayed Feedback.** Designing informative yet physically grounded reward signals is a central challenge in orbital control. Rewards must capture long-term mission objectives while remaining interpretable within physical constraints. OrbitZoo provides a flexible reward framework that integrates both inter-body and body-specific metrics, enabling the formulation of rich and diverse learning objectives. Inter-body quantities – such as relative distance or line-of-sight conditions – allow the definition of cooperative or competitive multi-agent tasks. In parallel, each body exposes individual physical attributes such as fuel consumption and mass variation, which can be incorporated into agent-specific reward functions.

This modular reward interface enables experimentation across dense and sparse regimes, as well as multi-objective trade-offs between performance, safety, and efficiency. It supports studies on credit assignment, reward misspecification, and curriculum learning [53] in complex multi-agent orbital environments, where delayed effects and coupled dynamics make the reward landscape inherently nontrivial.

**Adversarial and Safety-Critical Learning.** OrbitZoo's modular setup supports adversarial and safety-critical scenarios, such as pursuit–evasion, intentional jamming, or collision avoidance. The environment includes inter-body metrics like the probability of collision (POC) and body-specific measures of propagation uncertainty, allowing agents to quantify and respond to risk in real time. Combined with partial observability and fuel-aware dynamics, these features enable research on robust policy design, safe exploration, and adversarial resilience in continuous orbital domains.

**Visualization and Debugging.** OrbitZoo offers a real-time, integrated 3D visualization tool that aids policy inspection and failure case diagnosis. Although not a core contribution, this represents, to the best of our knowledge, the first Python-based RL framework with a built-in, real-time orbital visualization interface, as summarized in Appendix A. This feature enhances interpretability, reproducibility, and debugging by providing an intuitive link between learned behaviors and the underlying orbital dynamics.

# 5 Experiments and Results

To evaluate the effectiveness of OrbitZoo in modeling diverse orbital missions and supporting RL methods, we conducted a series of experiments using several deep learning algorithms (i.e., DQN, DDPG and PPO), as well as FRL applied to PPO in a multi-agent setting. These include the development of a mission analogous to one created in a different environment, orbital transfers (OTs), collision avoidance maneuvers (CAMs), a multi-agent geostationary orbit (GEO) constellation coordination problem using independent and federated learning, and a validation experiment comparing OrbitZoo's simulations against real-world Starlink ephemeris data. In the following missions, we consider an RL agent as a satellite with limited maneuvering capabilities, learning a strategy (policy) to apply thrusts that maximize the expected return (Sec. 3.1). Details can be found in Appendix E.

## 5.1 Single-Agent Hohmann Maneuver

The Hohmann transfer experiment was chosen as a benchmark to evaluate OrbitZoo's ability to support RL in a high-fidelity orbital dynamics environment. Details of the experiment can be found in Appendix E.4. As a classic problem in orbital mechanics, the Hohmann transfer is analytically solvable and provides a clear reference for comparing RL-derived solutions with theoretical optima [33].

**Setup.** The agent observes the satellite's current orbital state through equinoctial coordinates and receives a reward based on the reduction of transfer error while minimizing fuel consumption. The action space corresponds to polar thrust parameters $(T, \theta, \phi)$, where $T$ is the thrust magnitude and $(\theta, \phi)$ define the thrust direction. The environment incorporates perturbative forces, including drag.

**Results.** The RL agent learned near-optimal strategies for a 30 km altitude Hohmann transfer, matching theoretical semi-major axis values. Deviations in other elements (e.g., inclination) arose from thrust misalignments. Contrary to the classical optimal maneuver, the agent adapted to new perturbations and reached the target successfully. These results validate OrbitZoo's ability to model classical problems in complex orbital dynamics and highlight areas for refining policy and reward design. Figure 17 shows the optimized transfer with minimal fuel use. The trajectory closely matches the theoretical solution (Figure 1), confirming OrbitZoo's high-fidelity simulation.

## 5.2 Single-Agent Collision Avoidance Maneuver

CAM missions in RL are often focused on station-keeping while preventing collisions with other objects. Many existing approaches assume perfect knowledge of the current state of all bodies in the environment, simplifying the task to maximizing the Euclidean distance from other bodies while maintaining proximity to a nominal orbit. In reality, however, the states of these bodies are known with uncertainty, which must be accounted for in operational planning. In this mission, a more realistic approach to CAM is adopted by explicitly modeling state uncertainty and its evolution over time. While the environment handles propagation using high-fidelity dynamics that closely resemble real-world physics, the prediction of future states – and the corresponding probability of collision (POC) [2] – relies on a simplified Newtonian attraction model, representing a low-fidelity simulation.

**Setup.** Two bodies – debris and a satellite with maneuvering capabilities – are synthetically instantiated in the same Cartesian position with some uncertainty, and with symmetric Cartesian velocity, creating a short-term encounter at that moment. By propagating both bodies backward in time, we create the initial conditions for the start of each episode, enough for the satellite to perform the needed maneuvers and ultimately avoiding a collision. Due to the uncertainty, a POC $> 10^{-6}$ is considered as high collision risk, and the agent is tasked at lowering it while staying near its initial orbit.

**Results.** In many existing CAM approaches, the RL agent is trained using discrete action spaces, since maneuvers are only required when a collision risk exists. Accordingly, a DQN was employed for training the agent. To investigate whether applying thrust in specific directions could lead to improved performance, the continuous version of PPO was also implemented. In this setup, the action space included a decision variable ($0 \leq \delta \leq 1$) indicating whether thrust should be applied ($\delta > 0.5$). For training, the environment used Newtonian attraction with atmospheric drag to model the actual

dynamics. During evaluation, however, all available perturbations in OrbitZoo were enabled to assess the generalization capability of the trained agents. Results show that both algorithms learned effective policies under the (simplified) training dynamics, but PPO demonstrated superior performance under realistic conditions, exhibiting a greater ability to reduce the POC. More details are presented in Appendix E.6.

## 5.3 Multi-Agent GEO Constellation Coordination

This experiment demonstrates OrbitZoo's ability to simulate and train multi-agent systems for satellite constellation management in geostationary orbit (GEO), using both independent and federated learning. More details can be found in Appendix E.7.

**Setup.**    A four-satellite constellation in GEO aimed to maintain equal angular separation and altitude while minimizing fuel use through small thrusts limited to the GEO orbital plane ($\phi = 0$ in polar parameterization). Each agent observes the anomalies of other satellites, its own semi-major axis and eccentricity, and outputs thrust commands via a decentralized policy. Training used PPO with generalized advantage estimation (GAE) [68]. At the start of each episode, the satellites are placed at random position along the GEO orbit.

**Results.**    Figure 2 shows a view of the OrbitZoo interface of the constellation's configuration after 4 days for a given initial configuration. Generally, the agents successfully obtained a larger angular separation while not drifting away from GEO. The policies are also able generalize to unseen perturbations, such as third body forces (Sun and Moon), SRP and drag. However, the constellation is not able to increase the overall angular separation within the given time in some configurations.

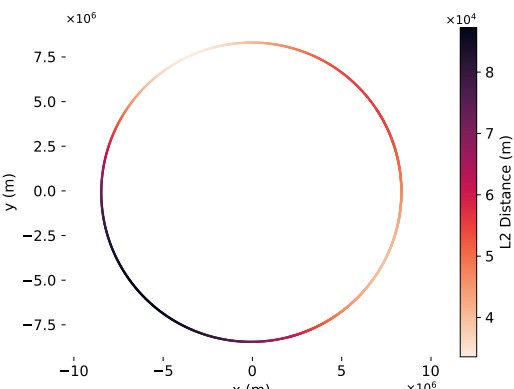

Figure 1: The L2 error between the optimal and Experiment 2 maneuvers stays low over long orbits, with most error due to minor inclination shifts. However, agents were trained to minimize equinoctial element differences, not Euclidean distance.

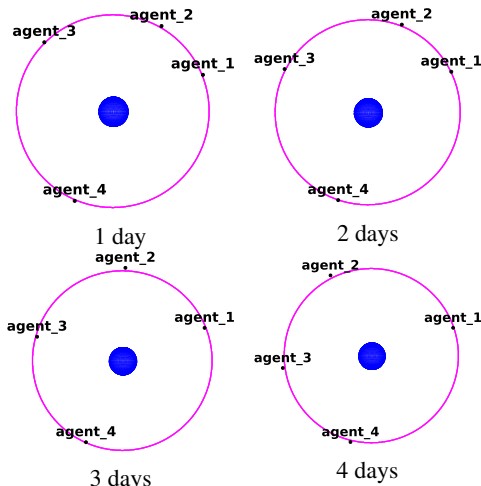

Figure 2: Constellation evolution over 4 days (seed 44). The purple circle represents GEO.

## 5.4 Validation Against Real-World Data

To evaluate OrbitZoo's ability to bridge the reality gap, we configured the environment to simulate Starlink satellites and compared its output to publicly available ephemeris data in Space-Track. Further details on the analysis can be found in Appendix G.

**Setup.**    Thirty-one Starlink satellites with publicly available data were selected for the validation experiment. Bayesian optimization was employed to tune parameters such as the drag coefficient, reflection coefficient, and satellite radius to match the ephemeris data. The RMSE between the OrbitZoo-propagated trajectories and the ephemeris data was used as the primary metric for evaluation.

**Results.** As summarized in Table 3, OrbitZoo achieved varying levels of accuracy across 31 satellites, with mean RMSE values ranging from 24.14 meters to 1924.90 meters over 16.6-hour propagation. While some satellites closely matched the Starlink ephemeris data, others showed significant deviations, likely due to limited information on their physical properties. Figure 3 illustrates the residuals between the propagated and observed trajectories, demonstrating the accuracy of OrbitZoo's physical modeling for the relevant horizon of two hours.

| Group | Mean RMSE (meters) |
| --- | --- |
| Low RMSE | 24.14 |
| Medium RMSE | 83.75 |
| High RMSE | 1924.90 |

Table 3: Root mean square error (RMSE) values, in meters, for three groups of satellites (a total of 31 satellites), separated based on an RMSE threshold. The "Low RMSE" group includes satellites with RMSE values below 50, the "Medium RMSE" group includes satellites with values between 50 and 100, and the "High RMSE" group is for satellites above 100. The estimated error is derived from a 16.6 hour propagation (1000 steps).

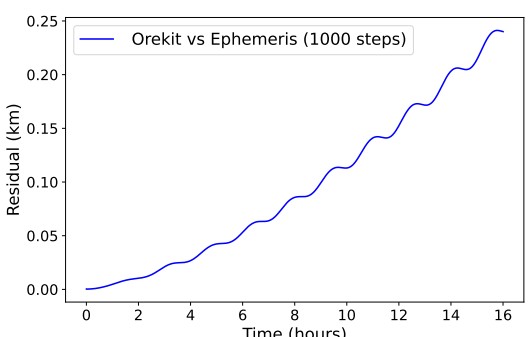

Figure 3: Residuals between OrbitZoo-propagated trajectories and Starlink ephemeris data for satellite 44744 in blue. The residuals remain low over the validation interval, confirming OrbitZoo's high fidelity.

## 5.5 Additional Experiments

Many RL missions and environments developed by other authors (as shown in Table 1) are not publicly available, hindering reproducibility and making it challenging to assess the complexity of their implementation. Among the few exceptions is the work by D. Kolosa [40] and A. Herrera [31], whose code is openly accessible and supports the results presented. In this context, we replicate both authors first missions in Appendix E.2 and Appendix E.3, respectively, to illustrate how OrbitZoo provides a flexible framework for developing and analyzing such missions. For completeness, we present an additional experiment: a chase-target scenario (Appendix E.5), where a satellite pursues a moving object in a higher orbit. For each mission, we provide a brief overview, followed by the experimental setup, results, key challenges, and directions for future improvement.

## Conclusions

Through comparisons with existing environments and a diverse set of experiments – including a **Hohmann transfer maneuver**, a **geostationary constellation coordination task**, and **validation using real-world Starlink ephemeris data** – we demonstrated that RL policies trained in **OrbitZoo** can achieve near-optimal control while remaining consistent with realistic satellite behavior. These results establish **OrbitZoo** as a reliable benchmark for RL-based autonomy in space operations. To assess generalization and robustness, we employed multiple RL algorithms and different levels of realism, highlighting how continuous action spaces generally enhance performance under realistic orbital dynamics. Beyond RL, **OrbitZoo** serves as a **publicly available platform** designed to support researchers in aerospace engineering, satellite operations, and machine learning. It provides a modular, high-fidelity environment for studying RL in realistic orbital settings. While grounded in orbital dynamics, its abstractions and challenge structures are representative of broader autonomy domains. The platform fosters reproducible experimentation and enables the exploration of autonomous decision-making for future space applications.

**Acknowledgments.** We thank the anonymous reviewers for their valuable input and for helping strengthen the manuscript. This work was partially supported by NOVA LINCS (UID/04516) funded by FCT IP, and the Neuraspace AI Fights Space Debris project (project code C626449889-00463050, operation code 2022-C05i0101-02), co-funded by Recovery and Resilience Plan and NextGeneration EU Funds, www.recuperarportugal.gov.pt.

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

## Appendix Table of Contents

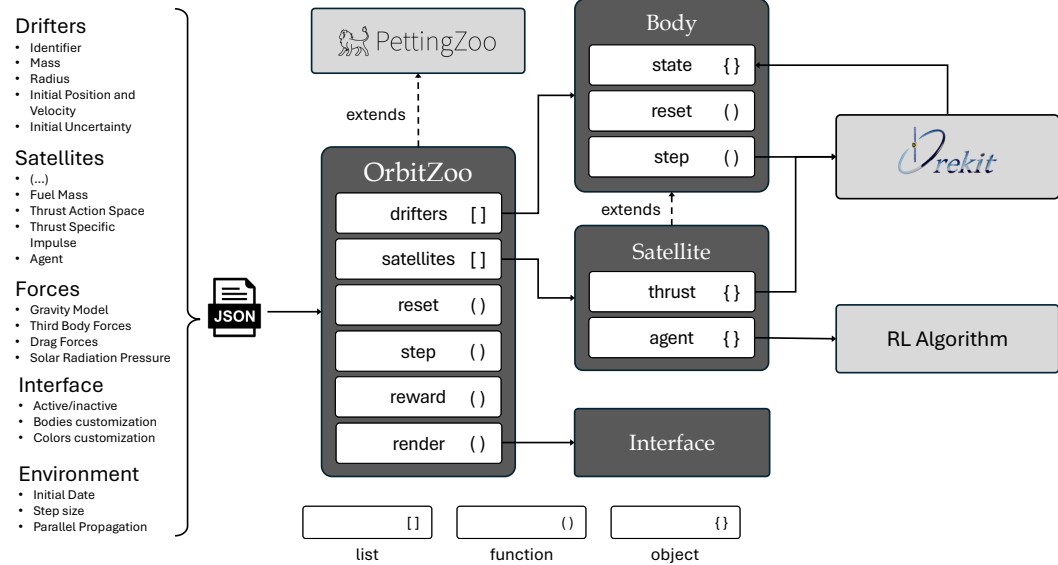

Figure 4: High-level overview of OrbitZoo's architecture. A JSON file describing the orbital system is provided to OrbitZoo, which then serves as an interface for data generation, single- and multi-agent RL mission development, or orbital dynamics analysis.

# A    OrbitZoo: A Framework for Multi-Agent RL in Orbital Dynamics

OrbitZoo is a flexible and modular Python environment for RL designed to address the challenges of applying MARL to high-fidelity orbital dynamics. It overcomes key limitations in existing tools, such as a lack of configurability and visualization, restricted support for realistic perturbative forces, and limited multi-agent capabilities. While the *vanilla* OrbitZoo environment does not implement specific missions, RL algorithms, and training logic, it can easily be extended and configured for such cases.

## A.1    Architecture and Design

OrbitZoo's architecture is designed to be modular and extensible. Figure 4 shows the primary modules, where darker boxes represent developed classes and lighter colors indicate integrated external components. This modular design allows users to implement and customize each component independently, ensuring compatibility with diverse RL algorithms and experimental setups.

### A.1.1    Core Modules

**Body: Modular Propagation of Orbital Dynamics.**    The *Body* class is the foundation of OrbitZoo and represents physical entities in the environment. Each body instance contains an individual numerical propagator, which is used to compute future states with great accuracy (position, velocity, uncertainty and mass) knowing the body's current state and active forces. Upon initialization, each body must contain a unique identifier (name), a dry mass, a radius, an expected Cartesian position and velocity $\mu = (x, y, z, \dot{x}, \dot{y}, \dot{z})$, and uncertainties associated with each of these elements $(\sigma_x, \sigma_y, \sigma_z, \sigma_{\dot{x}}, \sigma_{\dot{y}}, \sigma_{\dot{z}})$, which are internally used to construct the covariance matrix $\Sigma$ with these uncertainties as its diagonal entries. Optionally, the initial date can be provided, allowing an accurate representation of perturbative forces acting on the satellite at a specific moment, namely drag. When resetting the environment, the actual position and velocity of the body are sampled from a multivariate normal distribution $\mathcal{N}(\mu, \Sigma)$. Although numerical propagation is performed in Cartesian coordinates in the background, each body instance provides methods to retrieve its current state in multiple representations – Cartesian, Keplerian, or equinoctial. Some static methods also provide information related to two given bodies, such as the distance between them, line of sight without Earth's intersection, Time of Closest Approach (TCA), or Probability of Collision (POC), which are useful for a large range of RL missions.

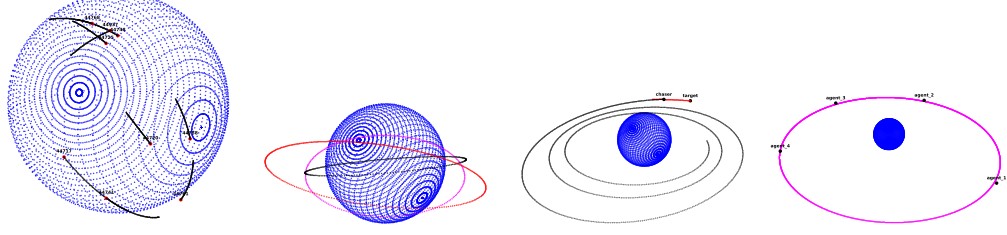

(a) A system propagating several Starlink satellites.   (b) A system with 3 different Keplerian orbits.   (c) A single-agent mission: Chase Target.   (d) A multi-agent mission: Constellation in GEO.

Figure 5: Frames of different systems on OrbitZoo's interface.

**Satellite: Extending Bodies with Thrust Capabilities.**   The *Satellite* class extends the *Body* class to incorporate propulsion systems and agent parameters, essential for RL tasks requiring control and maneuverability. Satellites are configured with: (1) polar thrust parametrization $(T, \theta, \phi)$ and action space, including maximum thrust $(T_{\max})$, deviation angle $(\theta_{\max})$, and azimuthal angle $(\phi_{\max})$ for realistic thrust modeling; (2) initial fuel mass and thrust-specific impulse $(I_{\text{sp}})$ for long-duration missions; and (3) agent parameters. The agent parameters are arbitrary and serve as a standardization, depending on the specific requirements of the implemented algorithm for initialization. In addition to the instance methods provided by the *Body* class, a *Satellite* includes fuel-related methods that can be used both to define episode termination conditions and to shape reward functions.

**Interface: Interactive Visualization.**   The *Interface* class provides interactive 3D visualization of the orbital environment, making OrbitZoo particularly useful for debugging, analysis, and presentation. It supports: (1) customizable visual components, such as central body, equatorial grids, Keplerian orbits, velocity and thrust vectors, and satellite trails; (2) real-time updates of system states, including timestamps, body names, and orbital parameters; and (3) flexible camera perspectives for inspecting orbital trajectories and multi-agent interactions. Figure 5 demonstrates some visualization capabilities, ranging from single-agent missions to multi-agent constellations.

Additionally, we demonstrate this tool through four videos that highlight its capabilities in various scenarios. The first video offers an overview of the user interface, followed by demonstrations of the Hohmann maneuver mission, the GEO constellation mission, and a chase-target mission. These videos can be accessed through the following links: Interface Video; Hohmann Maneuver Mission; GEO Constellation Mission; Chase Target Mission. The interface is built upon play3d [1], a library that leverages 2D perspective projections to create interactive 3D environments.

**Environment: High-level Interaction.**   The *Environment* (class OrbitZoo) serves as the primary interface for users. It integrates the above modules to provide a streamlined workflow for designing and executing RL missions. Key features include: (1) high-level methods for retrieving orbital state information, managing agents, and configuring scenarios; (2) flexibility to define single-agent or multi-agent missions, supporting tasks such as station-keeping, orbital transfers, and collision avoidance; and (3) compatibility with PettingZoo, enabling seamless integration with existing RL workflows. At a high level, the environment implements the four fundamental methods common to RL frameworks: `reset`, `step`, `reward`, and `render`.

### A.2   Use Cases

Currently, OrbitZoo focuses on systems orbiting Earth, reflecting the growing challenges posed by the rapid increase in satellites and debris within LEO. The framework simulates realistic gravity fields and perturbations – such as atmospheric drag, solar radiation pressure, and third-body effects – providing a high-fidelity environment for RL in near-Earth operations while also supporting realistic control maneuvers. Nonetheless, by modifying a few physical attributes, such as the gravitational parameter of the central body, the environment can be easily adapted to represent other celestial systems.

Typical RL-based use cases (as explored in this paper) include autonomous orbit transfer, formation flying, rendezvous, collision avoidance and debris tracking, where agents must make real-time deci-

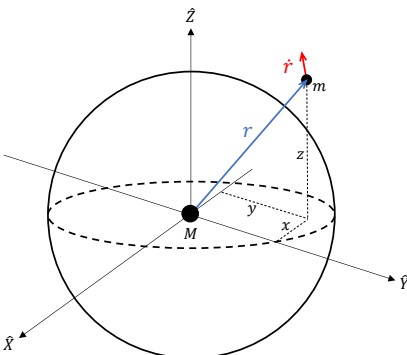

Figure 6: Representation of the Cartesian position ($r$) and velocity ($\dot{r}$) of a small body with mass $m$ orbiting a large central body with mass $M$.

sions in complex dynamical regimes. While the current focus remains Earth-centric, the modular design of OrbitZoo supports future extensions to other planetary or multiple central body environments, effectively providing a high-fidelity environment for CR3BP scenarios.

Notably, RL is only one component of OrbitZoo. The environment can be used independently of RL to test classical maneuver strategies, analyze orbital evolution, or study the intrinsic dynamics of multi-body systems under perturbations. This flexibility allows researchers to validate deterministic control laws, benchmark optimal control solutions, or simply observe long-term dynamical behavior under realistic forces.

## B  Orbital Mechanics

This section provides an overview of key orbital mechanics concepts to support a clearer understanding of the concepts presented throughout the paper. We begin by explaining the most commonly used coordinate systems, then discuss our approach to thrust representation, and conclude with how body propagation is usually handled in orbital dynamics.

### B.1  Coordinate Systems

Orbital dynamics is typically described using different coordinate systems, each offering unique advantages depending on the problem at hand. Considering a large central body (such as Earth) as an inertial frame, Cartesian coordinates provide a straightforward and intuitive representation of a body's position and velocity in space as $r = (x, y, z)$ and $\dot{r} = (\dot{x}, \dot{y}, \dot{z})$, respectively, as shown in Figure 6. While useful for direct numerical computations, such as altitude or distance between bodies, this state rapidly changes and often lacks the deeper insights into the actual orbital shape and orientation, which many missions focus on, such as the closest point (periapsis) or farthest point (apoapsis) from the orbiting body.

Keplerian elements, as Figure 7 shows, describe the orbital motion using five parameters: the semi-major axis ($a$), eccentricity ($e$), inclination ($i$), argument of perigee ($\omega$) and longitude of the ascending node ($\Omega$). Within this orbit, the anomaly (which can be represented in three ways, as seen in Figure 8) is an angle that indicates the current body position within that orbit, measuring from the periapsis. The mean anomaly is commonly used due to its linear evolution over time. When there are no perturbative forces (including maneuvers) that change the inclination, all points of the orbit are included in a two-dimensional plane, called the orbital plane. In these cases, Cartesian positions and velocities vectors are always within this plane.

These elements directly relate to the orbital geometry and are particularly useful for characterizing two-body motion. However, $\omega$ and $\Omega$ can become undefined or ambiguous in special cases, such as circular or equatorial orbits. These scenarios, referred to as singularities, pose significant challenges for numerical integration and optimization problems. Equinoctial elements are therefore preferred for RL, as they avoid these singularities by using five parameters:

$$a = a \quad e_x = e\cos(\omega + \Omega) \quad e_y = e\sin(\omega + \Omega) \quad h_x = \tan(i/2)\cos(\Omega) \quad h_y = \tan(i/2)\sin(\Omega),$$

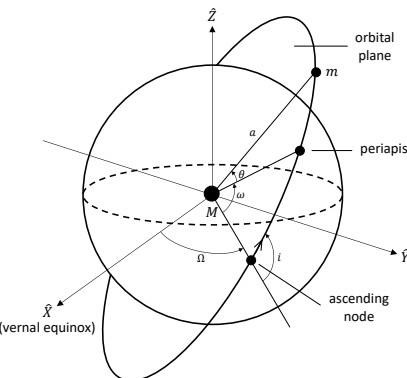

Figure 7: Representation of Keplerian Orbital Elements. Here, it is assumed that the orbit is perfectly circular ($e = 0$), and consequently, the semi-major axis ($a$) corresponds to the Euclidean distance to the primary focus ($M$).

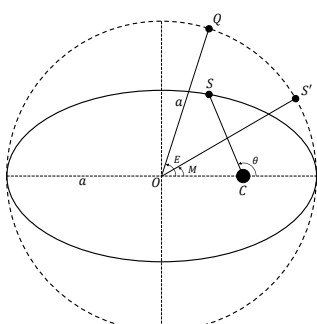

Figure 8: Types of anomaly. A body $S$ orbits a central body $C$ in an elliptical orbit with semi-major axis $a$. The true anomaly ($\theta$) is the actual angular position of $S$ measured from the periapsis (closest point to $C$). The mean anomaly ($M$) is the angle assuming uniform motion over the same orbital period (represented by body $S'$). Eccentric anomaly ($E$) is an intermediate angle used to relate $M$ and $\theta$, representing the position of $S$ on the circular orbit ($Q$). Despite evolving at different rates, all types of anomaly reach periapsis (0 rad) and apoapsis ($\pi$ rad) at exactly the same moment.

where $e_x$ and $e_y$ represent components of the eccentricity vector, and $h_x$ and $h_y$ describe the inclination vector.

### B.2 Thrust Representation

Spacecraft control often employs local reference frames (centered on the body) for thrust actions due to their intuitive representation, such as RSW. Given a satellite's Cartesian position $r$ and velocity $\dot{r}$, it is straightforward to compute the radial ($\hat{R}$), cross-track ($\hat{W}$) and along-track ($\hat{S}$) unit vectors:

$$\hat{R} = \frac{r}{\|r\|} \qquad \hat{W} = \frac{r \times \dot{r}}{\|r \times \dot{r}\|} \qquad \hat{S} = \hat{W} \times \hat{R}, \tag{1}$$

where $\times$ represents the cross product and $\| \cdot \|$ is the L2 norm. This coordinate system places the spacecraft at the origin, with $\hat{R}$ pointing towards the central body, $\hat{W}$ pointing perpendicular to the orbital plane, and $\hat{S}$ pointing perpendicular to $\hat{W}$ and $\hat{R}$ (approximately the same direction as $\dot{r}$).

In this approach, the thrust usually consists of a vector $\mathbf{T}_{\text{RSW}} = (T_R, T_S, T_W)$ that represents the thrust magnitude on each axis of the RSW frame. The action space is then limited to a maximum thrust magnitude $T_{\max}$ on each component: $\mathbf{T}_{\text{RSW}} \in [-T_{\max}, T_{\max}]^3$.

A more realistic and versatile approach to modeling the thrust action space is to adopt a polar parametrization, representing the thrust as $\mathbf{T} = (T, \theta, \phi)$. A visual comparison can be seen in Figure 9. This parameterization limits the thrust to the action space $\mathbf{T} \leq (T_{\max}, \theta_{\max}, \phi_{\max})$. When

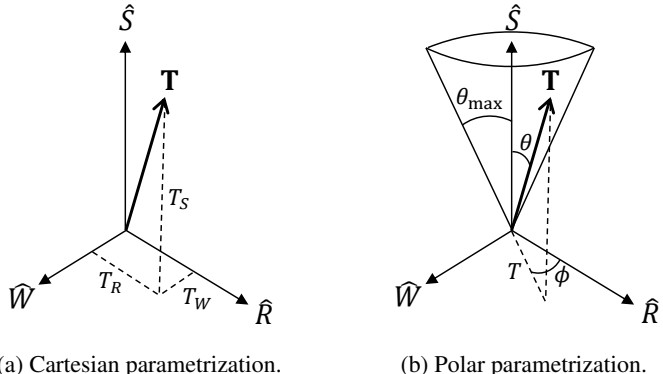

(a) Cartesian parametrization.  (b) Polar parametrization.

Figure 9: Comparison of Cartesian and Polar parameterizations of the thrust vector $\mathbf{T}$.

$\theta_{\max} = \pi$ rad and $\phi_{\max} = 2\pi$ rad, the satellite can apply thrust in any possible direction. This parameterization naturally constrains the thrust to a cone-shaped region, providing a physically realistic limit on the directions in which the satellite can apply force, offering a more controlled and flexible representation of the satellite's maneuvering capabilities. Finally, the thrust vector in the RSW frame can be obtained via the following transformation:

$$\mathbf{T}_{\mathrm{RSW}} = T(\cos\theta \hat{S} + \sin\theta(\cos\phi \hat{R} + \sin\phi \hat{W})). \tag{2}$$

Now consider that the spacecraft has a thrust system that produces a force $\mathbf{T} \in \mathbb{R}^3$. Similarly to the gravitational acceleration, the actual acceleration provoked by this thrust system comes from Newton's second law of motion:

$$\mathbf{T} = m a_{\mathrm{thrust}} \Leftrightarrow a_{\mathrm{thrust}} = \frac{\mathbf{T}}{m}. \tag{3}$$

This expression indicates that propulsive capability is inversely related to the spacecraft mass; hence, as propellant is expended, the spacecraft is capable of higher acceleration values when using the same force. The rate at which mass is lost is given by:

$$\dot{m} = -\frac{\|\mathbf{T}\|}{I_{\mathrm{sp}} g}, \tag{4}$$

where $I_{\mathrm{sp}}$ is the specific impulse (a measure of the efficiency of the propulsion system) and $g$ is standard gravity. A higher $I_{\mathrm{sp}}$ value means that the spacecraft can generate more thrust per unit of propellant mass, which makes them more efficient by saving fuel.

## B.3 Propagation

The motion of celestial bodies is governed by Newton's law of universal gravitation, which states that each body exerts a gravitational force on every other body. For a two-body system, the force acting on a satellite or debris with mass $m$ due to a central body with mass $M$ is given by:

$$\mathbf{F} = -\frac{GMm}{\|r\|^2}\hat{r}, \tag{5}$$

where $G$ is the gravitational constant and $\hat{r}$ is the unit vector of position $r$. Using Newton's second law, the resulting acceleration of the smaller body is expressed as:

$$\ddot{r} = -\frac{GM}{\|r\|^2}\hat{r} = -\frac{\mu_E}{\|r\|^2}\hat{r}, \tag{6}$$

where $\mu_E$ is called the standard gravitational parameter of Earth.

Although this model forms the foundation of orbital mechanics and is commonly used in RL, analytical models like SGP incorporate additional perturbative forces – such as atmospheric drag, SRP, and Earth's oblateness – to provide a fast and relatively accurate method of propagating bodies.

Numerical propagators provide a powerful alternative to SGP by solving the equations of motion through numerical integration, enabling precise modeling of complex orbital dynamics. Unlike SGP, numerical propagators can incorporate a wide range of perturbative forces ($a_{\text{env}}$), including higher-order gravitational harmonics, third-body effects, atmospheric drag, and SRP. This flexibility allows them to handle scenarios requiring high precision, such as long-term orbit predictions and mission-critical maneuvers.

Assuming the state of a spacecraft is characterized by its Cartesian position $r$, velocity $\dot{r}$ and mass $m$, the state vector $s = (r, \dot{r}, m) \in \mathbb{R}^7$ can be propagated in time using integration methods. The Runge-Kutta (RK4) [14] method is a numerical integration technique used to solve first-order differential equations, which can be used to approximate the unknown function $s$ dependent on time $t$. Since the state corresponds to a second-order system, it can be rewritten as a first-order system:

$$f(t,s) = \frac{d}{dt}s = \frac{d}{dt}\begin{bmatrix} r \\ \dot{r} \\ m \end{bmatrix} = \begin{bmatrix} \dot{r} \\ \ddot{r} \\ \dot{m} \end{bmatrix} = \begin{bmatrix} \dot{r} \\ -\frac{\mu_E}{\|r\|^2}\hat{r} + \frac{\mathbf{T}}{m} + a_{\text{env}} \\ -\frac{\|\mathbf{T}\|}{I_{\text{sp}}g} \end{bmatrix}. \tag{7}$$

By knowing the state of the spacecraft at a given moment, we create the initial conditions $t_0$ and $s_0$. To get an approximation of the state after a step size $\Delta t$, we define:

$$s_{\Delta t} = s_0 + \frac{\Delta t}{6}(k_1 + 2k_2 + 2k_3 + k_4), \tag{8}$$

where

$$k_1 = f(t_0, s_0),$$
$$k_2 = f\left(t_0 + \frac{\Delta t}{2}, s_0 + \Delta t\frac{k_1}{2}\right),$$
$$k_3 = f\left(t_0 + \frac{\Delta t}{2}, s_0 + \Delta t\frac{k_2}{2}\right),$$
$$k_4 = f\left(t_0 + \Delta t, s_0 + \Delta t k_3\right).$$

The time dependence of $f(s,t)$ in equation 7 enables the modeling of forces that are active only at specific times, such as environmental accelerations $a_{\text{env}}$ or thrust maneuvers.

To propagate the uncertainty, one can use the state transition matrix (STM) to analytically approximate the expected uncertainty found in Monte Carlo simulations.

Assuming we are only interested in the uncertainty of the position $r$ and velocity $\dot{r}$, the state can be simplified to $s = (r, \dot{r})$, therefore representing the dynamics as $f(t,s) = (\dot{r}, \ddot{r})$. By calculating the Jacobian $A \in \mathbb{R}^{6\times6}$, which linearizes the dynamics around $t_0$:

$$A = \frac{\partial f}{\partial s} = \begin{bmatrix} \frac{\partial \dot{r}}{\partial r} & \frac{\partial \dot{r}}{\partial \dot{r}} \\ \frac{\partial \ddot{r}}{\partial r} & \frac{\partial \ddot{r}}{\partial \dot{r}} \end{bmatrix} = \begin{bmatrix} 0 & I \\ \frac{\partial \ddot{r}}{\partial r} & \frac{\partial \ddot{r}}{\partial \dot{r}} \end{bmatrix}, \tag{9}$$

it is possible to approximate the STM ($\Phi \in \mathbb{R}^{6\times6}$) as:

$$\Phi = I + A\Delta t. \tag{10}$$

By knowing the position and velocity covariance matrix at $t_0$ ($\Sigma_0 \in \mathbb{R}^{6\times6}$), the propagated covariance ($\Sigma_{\Delta t}$) becomes:

$$\Sigma_{\Delta t} = \Phi\Sigma_0\Phi^T. \tag{11}$$

## C Formal Model Definitions

For completeness, we provide the mathematical details of the methods referred to in the main paper.

### C.1 Double Deep Q-Network (DDQN)

Double Deep Q-Network (DDQN) [29] is an improvement over the standard Deep Q-Network (DQN) [51] algorithm used in RL for environments with discrete action spaces. Although powerful, DQN suffers from overestimation bias (the tendency to overestimate action values due to using the same network for both action selection and action evaluation). DDQN addresses this by decoupling these two roles using two networks: an online network with parameters $\theta$ used for selecting actions, $Q_\theta(s_t, a_t)$, and a target network with parameters $\theta^-$ used for evaluating the value of the selected actions without constant deviations, $Q_{\theta^-}(s_t, a_t)$.

After an agent interacts with the environment and stores several experience tuples $(s_t, a_t, r_t, s_{t+1})$ in a buffer, DDQN samples a batch $B$ of experiences from the buffer and minimizes the mean squared error (MSE) between the current Q-value, $Q_\theta(s_t, a_t)$, and the estimated real Q-value, $Q_\theta^*(s_t, a_t)$:

$$\min_\theta \frac{1}{|B|} \sum_{(s_t, a_t, r_t, s_{t+1}) \sim B} (\underbrace{r_t + \gamma Q_{\theta^-}(s_{t+1}, \pi_\theta(s_{t+1}))}_{Q_\theta^*(s_t, a_t)} - Q_\theta(s_t, a_t))^2, \tag{12}$$

where $|B|$ is the size of the batch, $0 \leq \gamma < 1$ is the discount factor and $\pi_\theta(s_{t+1}) = \max_a Q_\theta(s_{t+1}, a)$ corresponds to the best action according to the current parameters $\theta$. Periodically, the target parameters $\theta^-$ are updated with the parameters of the online network, $\theta$.

Exploration in DQN is guided by the epsilon-greedy strategy, where at each time step, the agent selects a random action with probability $\epsilon \in (0, 1)$, and with probability $1 - \epsilon$, it chooses the action with the highest estimated Q-value. This probability $\epsilon$ typically starts relatively high to encourage exploration early in training and gradually decreases over time, allowing the agent to increasingly exploit the strategies it has learned.

### C.2 Deep Deterministic Policy Gradient (DDPG)

DDPG [43] is an off-policy RL algorithm that contains an actor and a critic network, as it stores experiences in a buffer gathered in interactions with old policies, similar to DQN. The actor network with parameters $\theta$ receives a state and produces an action (representing the policy, $\pi_\theta(a_t|s_t)$), and the critic network with parameters $\phi$ receives a state and an action, and produces a single value (representing the Q-value function, $Q_\phi(s_t, a_t)$). Similarly to DQN, DDPG also contains a target actor with parameters $\theta^-$ and a target critic with parameters $\theta^-$.

For optimization, both networks are dependent from the value that the other outputs. While the actor network tries to maximize the expected value produced by the critic (equation 13), the latter tries to approximate the Q-values to the true estimated ones by a MSE loss function (equation 14):

$$\max_\theta E\left[Q_\phi(s_t, \pi_\theta(s_t)\right] \tag{13}$$

$$\min_\phi \frac{1}{|B|} \sum_{(s_t, a_t, r_t, s_{t+1}) \sim B} (\underbrace{r_t + \gamma Q_{\phi^-}(s_{t+1}, \pi_{\theta^-}(s_{t+1}))}_{Q_\phi^*(s_t, a_t)} - Q_\phi(s_t, a_t))^2. \tag{14}$$

The target network can be updated using the soft update method, rather than directly copying the online network. This method uses a parameter $\tau \in (0, 1)$ to keep a percentage of the old targets, preventing large, abrupt changes in target values:

$$\theta^- \leftarrow \tau \cdot \theta + (1 - \tau) \cdot \theta^-. \tag{15}$$

Exploration in DDPG can be achieved using the Ornstein-Uhlenbeck process [79], which introduces temporally correlated noise $x_t$ to the actions produced by the actor network. This noise evolves over time and is updated at each time step according to:

$$x_{t+1} = x_t + \theta(\mu - x_t)\Delta t + \sigma\sqrt{\Delta t} \cdot \mathcal{N}(0, 1), \tag{16}$$

where $\theta$ determines how strongly the noise is pulled back toward the mean $\mu$, $\sigma$ controls the magnitude of the noise, and $\Delta t$ is a small time increment used to discretize the continuous process.

## C.3 Proximal Policy Optimization (PPO)

The actor loss in PPO [69] is defined based on the ratio $r_t(\theta)$, which measures the probability of taking an action $a_t$ under the current policy compared to the old policy:

$$r_t(\theta) = \frac{\pi_\theta(a_t|s_t)}{\pi_{\theta_{\text{old}}}(a_t|s_t)}. \tag{17}$$

The clipped objective ensures stability during training by preventing excessively large updates to the policy:

$$\min_\theta \mathbb{E}\left[\min\left(r_t(\theta)\hat{A}_t, \text{clip}(r_t(\theta), 1 - \epsilon, 1 + \epsilon)\hat{A}_t\right)\right], \tag{18}$$

where $\epsilon$ controls the trust region, and $\hat{A}_t$ is the advantage function. The advantage $\hat{A}_t$ is computed recursively using the temporal difference (TD) error:

$$\delta_t = r_t + \gamma V_\phi(s_{t+1}) - V_\phi(s_t), \tag{19}$$

$$\hat{A}_t = \delta_t + \gamma\lambda \cdot \hat{A}_{t+1}, \tag{20}$$

where $\lambda$ is a decay factor and $V_\phi(s_t)$ is the value function approximated by the critic network.

Since the actor network in PPO outputs the expected action the agent should take given a state, exploration is naturally handled by sampling from a probability distribution. Specifically, an action is sampled from a normal distribution centered at the policy output with a given standard deviation $\sigma$:

$$a_t \sim \mathcal{N}(\pi_\theta(a_t \mid s_t), \sigma). \tag{21}$$

The standard deviation $\sigma$ typically starts high to encourage exploration in the early stages of training and is gradually decreased over time to allow the agent to exploit the strategies it has learned.

## C.4 Generalized Advantage Estimation (GAE)

Generalized Advantage Estimation (GAE) [68] provides a flexible way to compute the advantage by balancing bias and variance through the $\lambda$ parameter:

$$\hat{A}_t = \sum_{l=0}^{\infty}(\gamma\lambda)^l\delta_{t+l}. \tag{22}$$

For long episodes, such as those encountered in orbital dynamics, GAE effectively stabilizes training by reducing variance in advantage estimation.

## C.5 Federated Learning (FL)

In MARL environments, there are several approaches to collaboratively train agents. A common method is independent learning, where each agent is trained separately and treats other agents as part of the environment's dynamics. In this setting, collaboration arises indirectly through a shared reward function, which must be identical for all agents to ensure coordinated behavior.

However, Federated Learning (FL) offers a more stable and scalable alternative by enabling agents to share knowledge during training while keeping their local data private. FL is a machine learning paradigm in which multiple entities (e.g., agents or devices) collaboratively train a global model without sharing raw data. Instead, they exchange model parameters or gradients, which are aggregated to improve a shared model.

Applied to RL, FL can be used to exchange learned parameters between agents rather than raw experience tuples. This is particularly beneficial in POMDP environments, where each agent has only a limited view of the full state. By federating value functions or policy networks, agents can better estimate the value of a given state $s_t$, leading to more stable and data-efficient learning while maintaining a level of privacy and decentralization.

A specific case of FL is horizontal federated learning (HFL) [59], which occurs when all agents have the same observation and action spaces. In such scenarios, multiple identical agents operating in parallel can be treated as instances of a single agent in multiple environments. This setting allows for

straightforward parameter sharing and can be seen as data augmentation across parallel environments, enhancing generalization and learning speed.

One common HFL algorithm that can be used for recent RL algorithms (e.g. DDPG and PPO) is Federated Averaging (FedAvg) [48], by training a shared critic through a central server (e.g., ground station). In an environment with $N$ agents, the server starts by sending the current global parameters $(w_g)$ to every agent: $w_i \leftarrow w_g, \forall i \in \{1, ..., N\}$ (where $w_i$ are the local parameters of agent $i$). Then, agents interact with the environment for an arbitrary number of steps, gathering a buffer of experiences $B_i$, which are then used to locally optimize the weights $w_i$ by an RL algorithm. After training, these weights are gathered by the server and averaged:

$$w_g = \sum_{i=1}^{N} \frac{|B_i|}{\sum_{k=1}^{N} |B_k|} w_i, \tag{23}$$

where $|B|$ is the number of experiences in buffer $B$. Finally, the process is repeated by sending these new global parameters to each agent. When we assume that the experiences gathered by each agent are independent identically distributed (IID), FedAvg converges.

# D  Computational Performance

Real orbital systems can consist of hundreds or even thousands of bodies, each requiring propagation at every step. Additionally, different forces may act on different bodies, and each body can have unique properties such as shape, attitude, or drag/reflection coefficients.

OrbitZoo imposes no strict limit on the number of bodies in a system, with scalability constrained only by the available hardware (D.1). However, it is essential to assess how increasing the number of bodies and introducing complex dynamics impact simulation speed and parallelization efficiency. These aspects are examined in the following subsections (D.2 and D.3).

Although not yet implemented, OrbitZoo experiments could benefit from libraries mentioned in 3.3 to accelerate training.

## D.1  Hardware Specifications

OrbitZoo supports systems of varying sizes and complexity, making hardware requirements dependent on the specific system being modeled, particularly in terms of CPU power and memory. For the following experiments and evaluations, the hardware used is detailed in Table 4, with the GPU utilized solely for training RL agents.

Table 4: Hardware specifications.

| Hardware | Specification |
|----------|---------------|
| CPU | Intel(R) Core(TM) i3-8100 CPU @3.60 Hz, 3600 Mhz, 4 Cores, 4 Logical Processors |
| GPU | NVIDIA GeForce GTX 1050 Ti |
| RAM | 16.0 GB, 2933 Mhz |

## D.2  Scalability

One of the simplest metrics for assessing scalability is simulation speed. Specifically, the time required to perform a single propagation step for all bodies in the system. Since multiple factors influence this speed, we focus on addressing the following questions:

- How does the simulation speed evolve with the addition of bodies?
- How does the simulation speed evolve with the addition of forces acting on the bodies (more realism)?

To address the first question, we evaluate system performance when propagating 1, 5, 10, 100, 1000, and 10000 bodies. For the second question, we analyze simulation speed starting with a simple

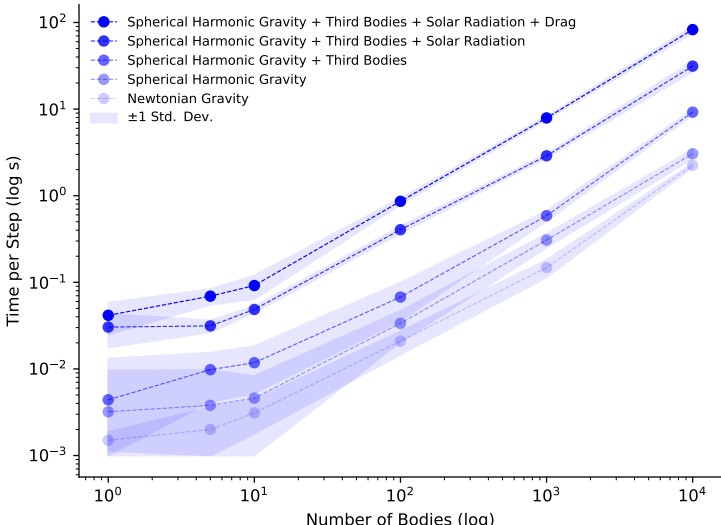

Figure 10: Time per step (in seconds) by number of bodies and active forces. Each point represents an average of 100 propagation steps of 10 seconds. Shaded regions indicate $\pm 1$ standard deviation.

Newtonian gravity model assuming a perfectly spherical central body. We then introduce a more complex gravity model using spherical harmonics (HolmesFeatherstone) and further incorporate perturbative forces, including third-body effects (Sun and Moon), SRP, and drag.

Results in Figure 10 show that OrbitZoo exhibits approximately $O(n)$ time complexity per step, where $n$ is the number of bodies in the system. This trend becomes more evident as $n$ increases. However, for smaller systems (1 to 10 bodies), the effects of parallelization are noticeable, as the hardware and OrbitZoo distribute computations across multiple threads. Additionally, the inclusion of perturbative forces increases the step time in a roughly linear manner.

Scalability and performance also depend on user-specific implementations. For example, in large LEO satellite constellations, drag is a crucial perturbative force for realistic trajectory modeling. However, drag computation relies on body shape, which is assumed to be unique for each satellite by default. As a result, each propagator stores its own atmospheric data, significantly increasing memory usage. By assuming uniform body characteristics (e.g., identical shapes), a shared force instance can be used across propagators, drastically reducing memory consumption.

### D.3 Parallelization

As shown in Figure 4, OrbitZoo integrates with two key external components: PettingZoo and Orekit. PettingZoo provides a framework for sequential or parallelized step computations, while Orekit handles propagation of orbital dynamics within each step. Since orbital MARL missions assume all agents act simultaneously rather than in turns, OrbitZoo propagates systems in a parallel manner, so it should naturally take advantage of available tools for faster computations.

OrbitZoo leverages PettingZoo's parallelization by implementing the ParallelEnv class, and Orekit's parallelization by implementing the PropagatorsParallelizer class (a way to simultaneously propagate several bodies). To assess the impact of this choice, we compared simulation times between parallel and sequential step computations (as it was presented in Figure 10).

Figure 11 shows that when using few active forces, the sequential propagation is faster than the parallel, even for a large number of bodies. However, when systems become more computational demanding – with many perturbative forces – the parallel mode becomes the fastest. Given that a single RL episode can consist of hundreds or thousands of steps, this improvement significantly enhances simulation efficiency, where using the parallel mode can decrease a single episode by several minutes.

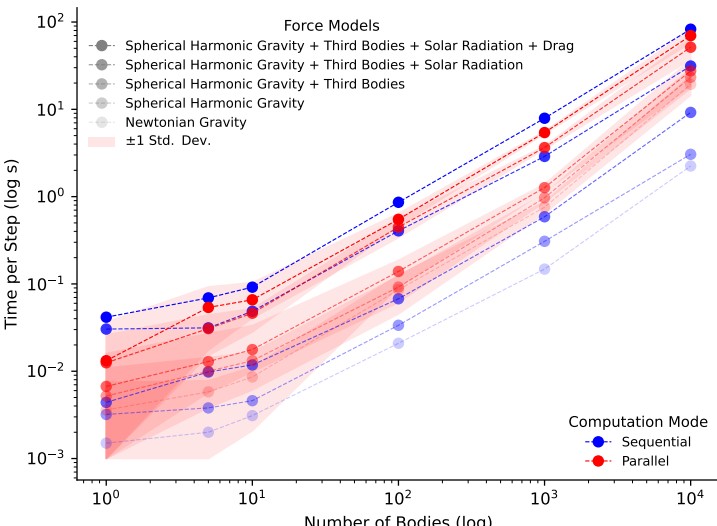

Figure 11: Time per step (in seconds) when using sequential and parallel computations, by the number of bodies and active forces. Each point represents an average 100 propagation steps of 10 seconds. Shaded regions indicate ±1 standard deviation.

Because of these results, OrbitZoo allows the user to change the computation mode upon environment creation (boolean value) according to its needs (through the JSON input file).

### D.4    Challenges and Future Improvements

As mentioned earlier, each propagator is currently treated independently, with duplicated instances for similar bodies. Future work regarding scalability may be related to automatically eliminating this redundancy for body propagators, as forces may be similar for different bodies, therefore benifitting from using the same force instances. This effectively reduces memory usage without the need to manually change these settings.

As real-world satellite constellations continue to grow in size and complexity, further research into scalability becomes increasingly important. Ensuring that RL-based control systems can scale effectively with the number of satellites, while maintaining performance and reliability, is a critical step toward practical deployment in real space missions.

## E    Experiments

To evaluate OrbitZoo's capabilities, we designed a series of missions utilizing algorithms ranging from foundational to state-of-the-art (as discussed in Section E.1). The first mission (Section E.2) compares the OrbitZoo environment to D. Kolosa's implementation for a general orbit change maneuver using DDPG, highlighting how the tools and flexibility provided by OrbitZoo can support the development of both existing and novel, more complex missions – without the technicalities typically present in orbital dynamics. Then, Herrera's [31] propagation method is compared to the one used in OrbitZoo, which is used to develop a station-keeping mission in LEO (Section E.3).

Next, two missions (Sections E.4 and E.5) focus on orbital transfers (OTs) using PPO, where agents learn strategies to reach static and dynamic targets, respectively, while considering real-world factors such as perturbative forces and mass loss due to fuel consumption.

We then present a collision avoidance maneuver (CAM) mission (Section E.6), in which a satellite learns to minimize the Probability of Collision with debris in the days leading up to a predicted close approach, while maintaining its initial orbit. This showcases how OrbitZoo can also be employed in missions where uncertainty plays a central role. In this mission, we directly compare the performance of strategies learned using DDQN and PPO, highlighting how discrete RL algorithms perform relative to continuous ones in CAM missions.

Finally, we introduce a multi-agent reinforcement learning (MARL) mission (Section E.7) set in Geostationary Earth Orbit (GEO), where four satellites learn to distribute themselves evenly along the orbit without losing altitude. For this scenario, we compare independent learning and federated learning approaches using the PPO algorithm, demonstrating how knowledge sharing among agents in cooperative settings can be beneficial.

For each experiment, we first provide a brief definition, highlighting the key aspects of the mission. We then describe the environment setup, detailing the objectives, environment characteristics, observation and action spaces, and reward functions. Finally, we present the results, assessing the generalization capabilities of each policy, followed by a discussion of the challenges encountered and potential improvements.

All experiments are fully supported by the accompanying code, which is available at the following repository: `https://github.com/orbitzoo/orbit_zoo`.

### E.1 Learning Algorithms and Architectures

### E.1.1 PPO

The chosen algorithm for a large part of the missions was PPO, since it is a state-of-the-art RL algorithm in environments with continuous action spaces, such as in the experiments carried out in this paper. The implementation was inspired by [8], which offers a minimal PPO setup for discrete and continuous action spaces. However, several modifications were introduced to enhance learning performance, as recommended in [3]. These include the use of generalized advantage estimation (GAE) instead of Monte Carlo estimation for calculating advantages, recalculating advantages at each epoch, and employing batch-based training. Other improvements include changes in the network architecture (with input normalization, tanh activation functions, and smaller initial weights), together with hyperparameter tuning and performance adjustments (such as calculation of expected returns only at training time).

The overall structure of the actor and critic networks used throughout the experiments is similar, with two hidden layers and Tanh activation functions, as represented in Table 5. No extensive research was made to find optimal hyperparameters.

Table 5: Network Structure of PPO Actor and Critic.

| Layer | Actor Network (Input: `state_dim`) | Critic Network (Input: `state_dim`) |
|---|---|---|
| **Input** | `BatchNorm1d(state_dim)` | `BatchNorm1d(state_dim)` |
| **Hidden Layer 1** | `Linear(state_dim_actor, 500), Tanh` | `Linear(state_dim_critic, 500), Tanh` |
| **Hidden Layer 2** | `Linear(500, 450), Tanh` | `Linear(500, 450), Tanh` |
| **Output** | `Linear(450, action_dim), Tanh` | `Linear(450, 1)` |

### E.1.2 DDPG

DDPG was employed for the mission described in Section E.2, aiming to replicate the work presented in [40] within a custom environment. Our implementation builds upon [67], incorporating the Ornstein-Uhlenbeck noise process [79] for exploration, and enhanced with prioritized experience replay [65], adapted from the implementation in [66].

The architectures of both the actor and critic networks consist of two hidden layers with Tanh activation functions, as detailed in Table 6. No exhaustive hyperparameter tuning was performed. Unlike PPO, the DDPG critic network estimates the Q-value, evaluating the expected return of the specific action taken by the actor in a given state, hence receiving the state-action pair.

Table 6: Network Structure of DDPG Actor and Critic.

| Layer | Actor Network (Input: `state_dim`) | Critic Network (Input: `state_dim, action_dim`) |
|---|---|---|
| **Input** | `BatchNorm1d(state_dim)` | `BatchNorm1d(state_dim_critic)` |
| **Hidden Layer 1** | `Linear(state_dim, 512), Tanh` | `Linear(state_dim_critic, 512 + action_dim), Tanh` |
| **Hidden Layer 2** | `Linear(512, 256), Tanh` | `Linear(512 + action_dim , 256), Tanh` |
| **Output** | `Linear(256, action_dim), Tanh` | `Linear(256, 1)` |

### E.1.3 DDQN

DDQN was employed as an alternative to PPO for the collision avoidance task (see Section E.6), following a similar approach to that in [13]. The implementation is based on the open-source project [18], which builds upon the original DQN algorithm introduced in [52]. In our implementation, we also incorporate prioritized experience replay [65] and a target Q-value network for increased stability.

The Q-network architecture consists of two hidden layers with Tanh activation functions, as detailed in Table 7. No extensive hyperparameter tuning was conducted to optimize performance.

Table 7: Network Structure of DDQN Q-value network.

| Layer | Q-Value Network (Input: state_dim) |
|---|---|
| **Input** | BatchNorm1d(state_dim) |
| **Hidden Layer 1** | Linear(state_dim, 512), Tanh |
| **Hidden Layer 2** | Linear(512, 256), Tanh |
| **Output** | Linear(256, action_dim) |

## E.2 OrbitZoo vs. SOTA: Kolosa Comparison

### E.2.1 Definition

Kolosa [40] developed a mission in Medium Earth Orbit (MEO) where a spacecraft learns to perform a general orbit change maneuver using DDPG. In this section, we replicate Kolosa's mission within OrbitZoo.

### E.2.2 Environment Setup

**Objective and Environment Characteristics.** We design an environment inspired by Kolosa's setup. The scenario involves a spacecraft with a dry mass of 500 kg and an initial fuel mass of 150 kg, equipped with a thruster of specific impulse $I_{\text{sp}} = 3100$ s. The initial orbit is defined by the equinoctial elements $(a, e_x, e_y, h_x, h_y, M) = (5500 + R_E \text{ km}, 0.153, 0.128, 0.041, 0.015, 10°)$, with the goal of reaching a target orbit characterized by $(a, e_x, e_y, h_x, h_y) = (6300 + R_E \text{ km}, 0.154, 0.171, 0.042, 0.019)$ – representing a general orbital transfer, where $R_E = 6378$ is the radius of Earth. The dynamics follow Newtonian gravitational attraction. Each episode spans 692 steps, with a time interval of 500 s per step (approximately four days total).

**Action Space.** Departing from Kolosa's Cartesian thrust representation, we employ a polar thrust parameterization, as described in Section B.2. The action at each time step is defined as $a_t = (T, \theta, \phi)$, constrained by $(T_{\text{max}}, \theta_{\text{max}}, \phi_{\text{max}}) = (0.6, \pi, 2\pi)$.

**Observation Space.** Building upon Kolosa's observation space, we include an additional realistic and time-varying parameter: fuel mass, which decreases as thrust is applied. The observation at time $t$ is defined as $o_t = (s_t, M_t, f_t)$, where $s_t = (a, e_x, e_y, h_x, h_y)$ are the equinoctial elements, $M_t$ is the mean anomaly, and $f_t$ is the remaining fuel.

**Reward Function.** The reward function also follows the same formulation, computed based on the deviation between the current and target equinoctial elements. Specifically, for each element $e$, the difference $\Delta e$ is scaled by a corresponding weight $\alpha_e$, resulting in:

$$r_t = -(\alpha_a \Delta a + \alpha_{e_x} \Delta e_x + \alpha_{e_y} \Delta e_y + \alpha h_x \Delta h_x + \alpha h_y \Delta h_y), \tag{24}$$

where

$$\Delta e = \begin{cases} \frac{\sqrt{(\hat{e}-e)^2}}{\hat{e}} & \text{if } e = a \\ \sqrt{(\hat{e}-e)^2} & \text{if } e \in \{e_x, e_y, h_x, h_y\} \end{cases}.$$

Table 8: OrbitZoo vs. Kolosa: Training hyperparameters.

| Parameter | Value |
|---|---|
| Actor learning rate | 0.00001 |
| Critic learning rate | 0.0001 |
| Epochs | 1 |
| Discount factor ($\gamma$) | 0.99 |
| $\tau$ | 0.01 |
| $\mu$ | 0 |
| $\sigma$ | 0.2 |
| $\theta$ | 0.15 |
| $\Delta$ t | 0.01 |
| Memory Capacity | 10 000 |
| Initial Standard Deviation | 0.5 |
| Batch Size | 256 |

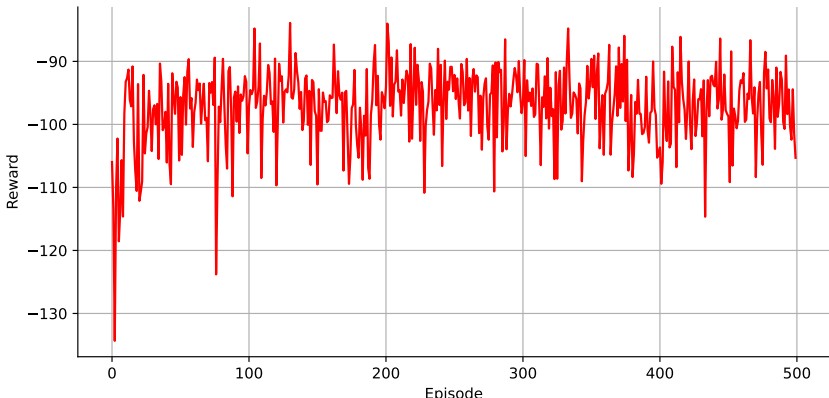

Figure 12: OrbitZoo vs. Kolosa: Cumulative reward per episode using the implemented DDPG.

### E.2.3 Results

The weights used during training were kept identical to those in Kolosa:

$$(\alpha_a, \alpha_{e_x}, \alpha_{e_y}, \alpha_{h_x}, \alpha_{h_y}) = (10^0, 10^0, 10^0, 10^1, 10^1).$$

The DDPG training hyperparameters are summarized in Table 8. The training performance using our DDPG implementation (see Section E.1.2) is presented in Figure 12.

When comparing our training performance with Kolosa's, we observe that our agent initially achieves a higher reward score but ultimately converges to a lower final score. This difference is likely due to architectural variations between the implementations. Nonetheless, the agent exhibits stable learning dynamics, with performance stagnating after approximately 200 episodes. Analyzing the evolution of the agent's orbital elements (Figure 13), we find that the agent correctly learns the objective, as evidenced by the consistent reduction in error across all elements except $e_y$. However, similarly to Kolosa, the agent is unable to converge all orbital elements within the predefined tolerance and time constraints.

Moreover, OrbitZoo offers a distinctive feature: a real-time visualization interface that allows users to observe the agent's maneuvers as they occur in space. This capability provides valuable insight into the learning dynamics and behavior of the agent, as illustrated in Figure 14.

This mission demonstrates that OrbitZoo can successfully reproduce similar RL missions with ease, while also offering additional configurations and features that facilitate the development and testing of RL agents.

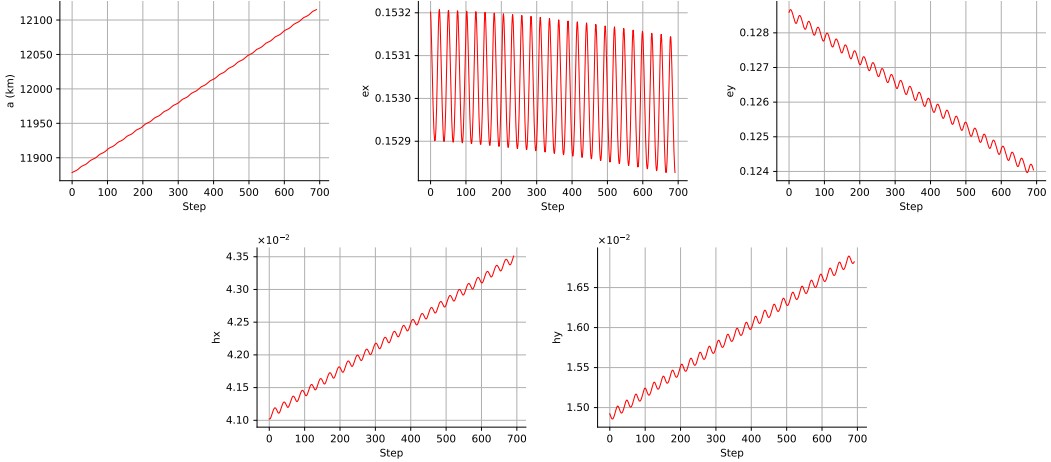

Figure 13: Orbital elements of the maneuver per step.

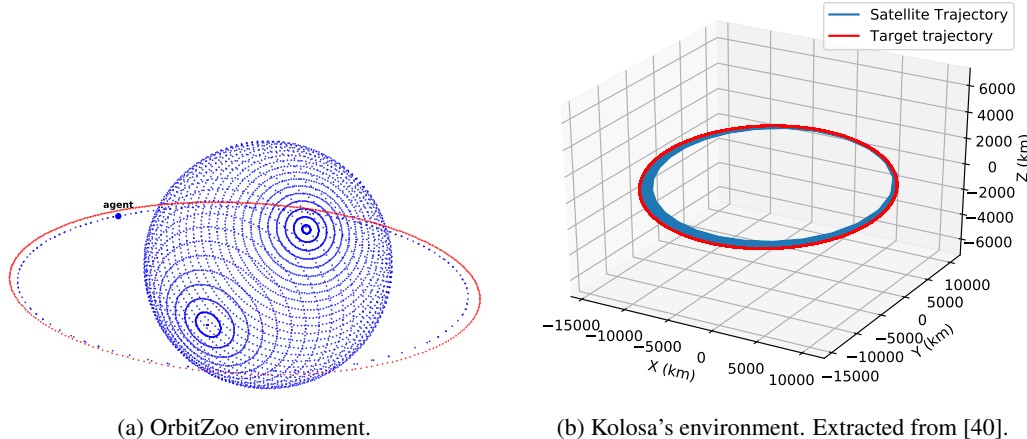

(a) OrbitZoo environment.
(b) Kolosa's environment. Extracted from [40].

Figure 14: Visual comparison of the maneuver in OrbitZoo and on Kolosa's environment.

### E.2.4 Challenges and Future Improvements

Since our DDPG implementation differs from the original, the agent learns a different strategy, resulting in diverging training outcomes. This highlights the need for further hyperparameter tuning to align the behavior more closely with expected results. Future work may focus on replicating additional environments and benchmark results, aiming to evaluate and enhance the generalization capabilities of OrbitZoo across a wider range of mission scenarios.

### E.3 OrbitZoo vs. SOTA: Herrera Comparison

To compare OrbitZoo's capabilities with those of existing environments that offer publicly available code, we extend our analysis by implementing the mission proposed by Herrera in [31], consisting of a station-keeping mission in LEO.

### E.3.1 Definition

Herrera developed a reinforcement learning environment for station-keeping in Low Earth Orbit (LEO). While Keplerian orbits describe a well-defined elliptical path around a central body, real-world satellites are influenced by perturbative forces beyond gravity, causing them to drift from their nominal orbits over time. Station-keeping missions aim to apply strategic control – through attitude adjustments or thrust maneuvers – to maximize the duration a satellite stays within its designated

orbit. This task becomes especially challenging and critical in LEO due to the significant impact of atmospheric drag.

### E.3.2 Environment Setup

**Objective and Environment Characteristics.** In this mission, Herrera employed PPO to train a satellite to control its thrust for station-keeping. The environment's dynamics were manually implemented using a 4th-order Yoshida integrator, which computes the satellite's next position and velocity based on gravitational and drag forces. The satellite began on a circular orbit at an altitude of 550 km above Earth's surface, with the objective of maintaining that altitude for as long as possible without deviating by more than 1 meter.

Herrera also clearly defined the episode termination conditions: (1) each episode has a maximum duration of 800 steps, with each step representing 1 second; (2) if the satellite does not perform any maneuvers, it exceeds the threshold after approximately 200 steps due to natural drift; and (3) the satellite is provided with enough fuel to apply maximum thrust for up to 125 steps. For a mission requiring this level of precision, the differences in dynamic modeling are critical. As stated by Herrera: "*The 4th Order Yoshida integrator is used as the default integrator as there is a balance of performance and error. RK4, while generally having less error, is not a semantic integrator and requires the calculation of the acceleration 4 times while the Yoshida integrator only requires 3.*". Given the difference in integration methods – Herrera employing the faster but less precise Yoshida integrator, and OrbitZoo using the more accurate Dormand-Prince integrator (a RK4 integrator with adaptive time steps) – we conduct a direct comparison between the values produced by each environment over the course of a full episode, starting from the same initial position and velocity. As shown in Figure 15, the difference in position after approximately 13 minutes is around 25 meters, while the difference in velocity is around 0.08 meters per second. With this comparison, we note that the one meter threshold initially defined for episode termination is extremely strict under realistic dynamic conditions. The code needed to run this comparison is provided in the supplementary material.

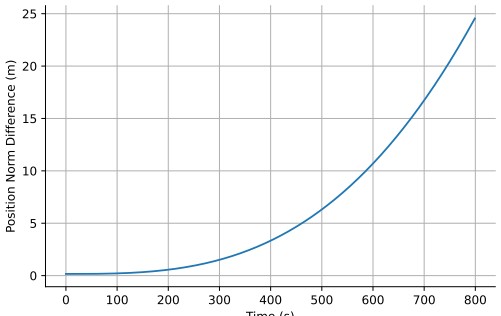
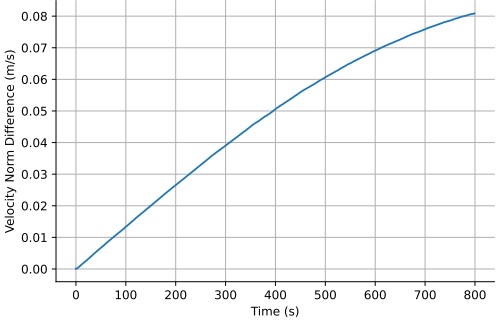

(a) Euclidean distance between Herrera's and Orbit-Zoo's propagated position, per second.

(b) Euclidean distance between Herrera's and Orbit-Zoo's propagated velocity, per second.

Figure 15: Comparison between Herrera's and OrbitZoo's natural propagation: position and velocity differences over time.

Following Herrera's setup, we designed a similar mission in which the agent has an initial mass of 100 kg, including 75 kg of fuel. The satellite is modeled as a perfect sphere with a radius of 16.8 meters, a drag coefficient of 2.123 and is equipped with a thruster characterized by a specific impulse of $I_{\mathrm{sp}} = 0.0067$ s. While this configuration is not physically realistic, it was chosen to maintain consistency with Herrera's termination conditions (2) and (3). Notably, in OrbitZoo, atmospheric drag is not solely determined by the satellite's current state (as in Herrera's implementation) but also incorporates historical data, allowing drag to vary dynamically based on the current moment.

**Action Space.** To simplify the problem, Herrera limits the action space to the orbital plane, and considers a polar thrust representation $(T, \theta)$. However, the action the agent performs is not directly the thrust being applied, but the variation applied to the current thrust, creating a smoother and more realistic maneuver: $a_t = (\Delta T, \Delta \theta)$. While the maximum thrust is char-

Table 9: OrbitZoo vs. Herrera: Training hyperparameters.

| Parameter | Value |
|---|---|
| Actor learning rate | 0.0001 |
| Critic learning rate | 0.001 |
| GAE $\lambda$ | 0.95 |
| Epochs | 5 |
| Discount factor ($\gamma$) | 0.99 |
| Clip ($\epsilon$) | 0.03 |
| Initial Standard Deviation | 0.5 |
| Standard Deviation Decay Rate | 10000 steps |
| Standard Deviation Decay Amount | 0.05 |
| Minimum Standard Deviation | 0.05 |
| Experiences for Training | 800 |
| Batch Size | 64 |

acterized by $(T_{\max}, \theta_{\max}) = (0.04, 2\pi)$, the action space is limited to a fraction of that change: $(\Delta T_{\max}, \Delta \theta_{\max}) = (T_{\max}/50, \theta_{\max}/6)$.

**Observation Space.** Similarly to Herrera, we use Cartesian coordinates to represent the current position $r \in \mathbb{R}^2$ and velocity $\dot{r} \in \mathbb{R}^2$ of the satellite. Additionally, we also define the distance to the nominal altitude, $r_{\text{target}} = |\|r\| - r_{\text{nominal}}|$ and nominal velocity, $\dot{r}_{\text{target}} = |\|\dot{r}\| - \dot{r}_{\text{nominal}}|$, where

$$r_{\text{nominal}} = 550 \text{ km} \qquad \dot{r}_{\text{nominal}} = \sqrt{\frac{\mu_E}{r_{\text{nominal}}}},$$

with $\mu_E$ representing the gravitational parameter of Earth. The observation $o_t \in \mathbb{R}^8$ consists of: $o_t = (r/r_{\text{nominal}}, \dot{r}/\dot{r}_{\text{nominal}}, r_{\text{target}}, \dot{r}_{\text{target}}, \theta, T)$.

**Reward Function.** In order to encourage the agent to both not run out of fuel and stay within the orbital termination threshold, the reward function is similar to Herrera's:

$$r_t = \begin{cases} 0 & \text{if } r_{\text{target}} > 1\text{m} \ \lor \ f = 0 \\ \frac{t}{800} + 0.5 & \text{otherwise} \end{cases}, \tag{25}$$

where $f$ is the current available fuel in kg, and $t$ is the current step within the episode.

### E.3.3 Results

Similarly to Herrera, we employed PPO to trained the agent, with the algorithm hyperparameters being shown in Table 9. If the agent can consistently exceed the baseline of 200 steps – representing the duration a satellite remains within tolerance without maneuvers – it demonstrates successful station-keeping. Figure 16 illustrates that, despite the narrow 1-meter tolerance from the nominal orbit (together with the more realistic perturbations), the agent effectively learned to maintain station-keeping by surpassing those 200 steps consistently.

### E.3.4 Challenges and Future Improvements

The agent did not achieve the same level of performance as in Herrera's environment, primarily due to differences in the dynamics, as demonstrated earlier. Additionally, the PPO implementations differed – one using TensorFlow and the other PyTorch – with variations in hyperparameters.

This attempt to replicate Herrera's environment suggests that OrbitZoo could benefit from using faster propagators. While these may be less precise, they are suitable for RL missions where extreme precision is not critical. As Herrera points out, continuous but deterministic algorithms (such as DDPG and TD3) may be better suited for this mission due to the precision required in controlling thrust maneuvers – something that is challenging to achieve with PPO.

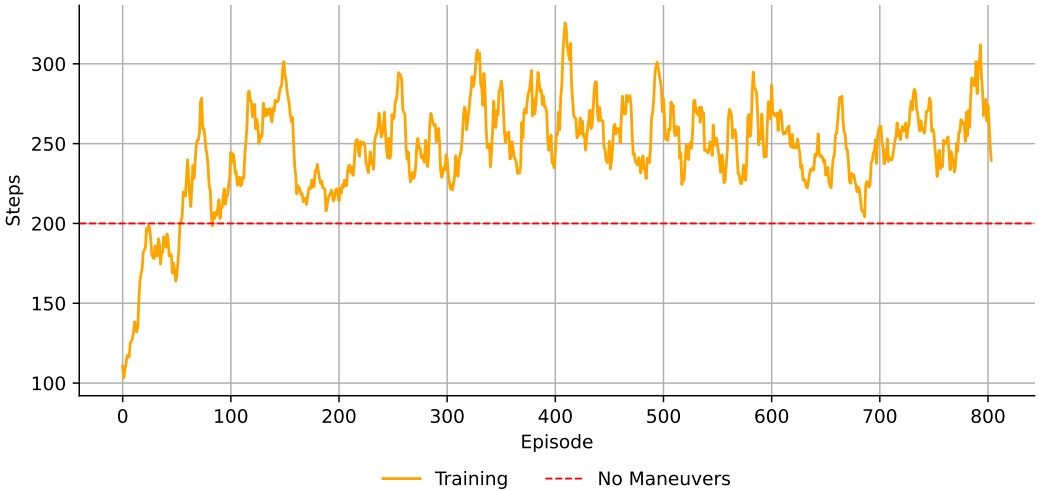

Figure 16: Steps per episode, averaged by a sliding window of 10 episodes. If no maneuvers are executed, the satellite exits the allowed tolerance range by step 200.

## E.4 Hohmann Maneuver

The Hohmann transfer experiment [33] was chosen due to its well-established relevance in orbital mechanics and its suitability as a benchmark for evaluating reinforcement learning frameworks in this domain. Furthermore, this maneuver is rarely addressed using reinforcement learning, highlighting the importance of investigating it. As one of the most fuel-efficient orbital transfer maneuvers, it provides a clear and analytically solvable problem that allows for direct comparison between RL-derived solutions and theoretical optima. Moreover, the Hohmann transfer incorporates key aspects of orbital dynamics, such as thrust application, trajectory optimization, and state transitions, making it an ideal testbed to validate the realism and effectiveness of OrbitZoo's high-fidelity environment.

### E.4.1 Definition

Consider a spacecraft on a nearly circular orbit, where its semi-major axis approximately corresponds to the distance of the spacecraft to the primary focus ($R$). If this spacecraft has the objective of ending up on an orbit with an increased distance of $R'$ while maintaining all other elements constant, the Hohmann transfer maneuver can achieve this in a highly fuel-efficient manner by applying two very specific impulsive thrusts in the along-track direction ($\Delta V$ and $\Delta V'$) [80]. The first establishes the transfer orbit (with a semi-major axis $R < a_H < R'$), and the second adjusts the elements to match the target orbit. When considering instantaneous impulses and conservation of energy, these changes in velocity can be easily calculated:

$$\Delta V = \sqrt{\frac{\mu}{R}}\left(\sqrt{\frac{2R'}{R+R'}} - 1\right) \qquad \Delta V' = \sqrt{\frac{\mu}{R'}}\left(1 - \sqrt{\frac{2R}{R+R'}}\right), \tag{26}$$

where $\mu$ is the standard gravitational parameter of Earth ($\mu = GM \approx 3.986 \text{ m}^3\text{s}^{-2}$), and $R$ and $R'$ are the radii of the departure and arrival orbit, respectively. According to the Tsiolkovsky rocket equation [78], a body with total mass $m_0$ and $m_0 - m_f$ of fuel mass, and with thrusting capabilities that have a specific impulse $I_{sp}$, can only perform the Hohmann maneuver if $\Delta V + \Delta V' \leq I_{sp}g_0 \ln\frac{m_0}{m_f}$, where $g_0$ is standard gravity. In a real scenario, the force ($F_1$) that is required to change the velocity by $\Delta V$ depends on the mass of the body and the time that we are applying that force ($\Delta t$):

$$F_1 = \frac{\Delta V \times m_0}{\Delta t}. \tag{27}$$

After this maneuver, the body loses some mass ($\dot{m}$) that is inversely proportional to the specific impulse of the thrust:

$$\dot{m} = \frac{F_1}{I_{sp}g_0}. \tag{28}$$

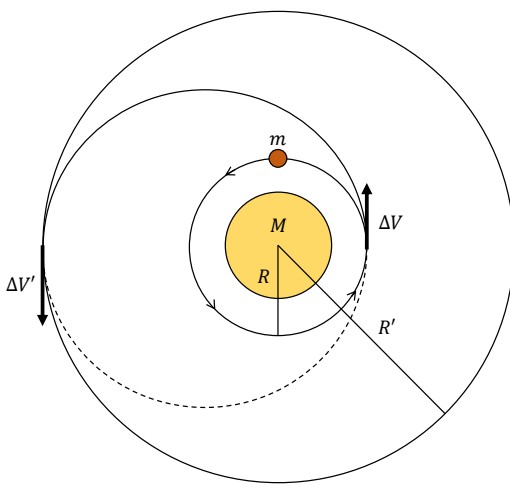

Figure 17: Representation of the Hohmann Maneuver. A spacecraft is initially in an orbit with semi-major axis $R$. It applies a thrust $\Delta V$ which positions it on the eccentric orbit. Later, it performs a second thrust $\Delta V'$ which positions it on the intended orbit, with a semi-major axis of $R'$.

Although small, this difference in mass will impact the force ($F_2$) that should be applied to change the velocity by $\Delta V'$:

$$F_2 = \frac{\Delta V' \times (m_0 - \dot{m})}{\Delta t}.$$

(29)

Finally, the time between $\Delta V$ and $\Delta V'$ (transfer time, $t_H$) is given by Kepler's third law:

$$t_H = \frac{1}{2}\sqrt{\frac{4\pi^2 a_H^3}{\mu}},$$

(30)

where usually it is assumed that $a_H = (R + R')/2$. A representation is shown in Figure 17.

### E.4.2 Environment Setup

**Objective and Environment Characteristics.** We define a spacecraft with a mass of $m_f = 200$ kg and 50 kg of fuel, starting on a near-circular orbit with equinoctial elements $(a, e_x, e_y, h_x, h_y) = (2000 + R_E \text{ km}, 0.007, 0.006, 0.041, 0.015)$ and ending up on $(a, e_x, e_y, h_x, h_y) = (2030 + R_E \text{ km}, 0.007, 0.006, 0.041, 0.015)$. Therefore, the objective is to raise the semi-major axis by 30 km. The spacecraft is equipped with a thruster that provides a specific impulse of $I_{sp} = 310$ s and is capable of performing thrusts in every direction.

According to the values provided, using the Hohmann transfer, the optimal maneuver requires $\Delta V = \Delta V' = 6.16$ m/s and a transfer time of $t_H = 3826.1$ s. Therefore, we define that each episode contains 1000 steps of 5 seconds, giving extra time for the agents to reach the target. Instead of assuming instantaneous burns, which are unrealistic, the thrust is applied for the entire step (5 seconds). For this interval, the optimal forces are $F_1 = 308$ N and $F_2 = 307.9$ N in the along-track direction.

For simplicity, we consider a Newtonian attraction model with no perturbations, and the agents to always start in the same anomaly with little uncertainty. The agent completes its objective if it reaches the target orbit within a tolerance for every equinoctial element: $(\pm 100 \text{ m}, \pm 0.005, \pm 0.005, \pm 0.001, \pm 0.001)$.

**Action Space.** To account for the requirement that thrust should be applied only at specific moments during the episode rather than continuously, an additional component is included in the action space: $a_t = (T, \theta, \phi, \delta)$. Here, $\delta$ represents the decision to apply thrust, where values close to 1 or 0 indicate a high certainty to apply or retain thrust, respectively. The action space is limited by $(T_{\max}, \theta_{\max}, \phi_{\max}, \delta_{\max}) = (500, \pi, 2\pi, 1)$.

Table 10: Hohmann maneuver: Training hyperparameters.

| Parameter | Value |
|---|---|
| Actor learning rate | 0.0001 |
| Critic learning rate | 0.001 |
| GAE $\lambda$ | 0.95 |
| Epochs | 5 |
| Discount factor ($\gamma$) | 0.99 |
| Clip ($\epsilon$) | 0.1 |
| Initial Standard Deviation | 0.5 |
| Standard Deviation Decay Rate | 40000 steps |
| Standard Deviation Decay Amount | 0.05 |
| Minimum Standard Deviation | 0.05 |
| Experiences for Training | 4096 |
| Batch Size | 64 |

**Observation Space.** The observation $o_t$ is defined as a vector comprising the current equinoctial elements $s_t = (a_t, e_{xt}, e_{yt}, h_{xt}, h_{yt})$, the mean anomaly $M_t$, and the remaining fuel $f_t$: $o_t = (s_t, M_t, f_t) \in \mathbb{R}^7$.

**Reward Function.** The reward function is designed to encourage the agent to apply large but accurate thrusts. First, the absolute difference between the current orbit at time $t$ and the target orbit, $\hat{s}$, is defined as $\Delta s_t = |s_t - \hat{s}|$. Next, the scalar progress is computed as $P_t = w^T \cdot (\Delta s_{t-1} - \Delta s_t)/s_t$, where $w$ is a vector of constant weights assigned to each orbital element. The reward function is then formulated as:

$$r_t = \mathbb{1}_{\delta > 0.5} \left( \alpha_1 \frac{T}{T_{\max}} P_t - \alpha_2 \frac{\theta}{\theta_{\max}} \right), \tag{31}$$

where $\mathbb{1}_{\delta > 0.5}$ is an indicator function that activates the reward only when the thrust decision $\delta$ exceeds 0.5, and $\alpha_1$ and $\alpha_2$ are the importance given to the progress, and actions in the along-track direction, respectively.

### E.4.3 Results

The agent training hyperparameters are shown in Table 10. Two experiments were conducted to analyze the agent's behavior, with two different reward functions as shown in Table 11. However, there was no extensive research to find optimal coefficients.

Table 11: Values of $\alpha_1$ and $\alpha_2$ used in each experiment to analyze the agent's behavior. The first experiment does not penalize thrust deviations from the along-track direction, while the second experiment includes a penalty for such deviations.

| Experiment | $\alpha_1$ | $\alpha_2$ |
|---|---|---|
| Experiment 1 | 1 | 0 |
| Experiment 2 | 1 | 0.5 |

For training, the weights used to measure the progress $P_t$ were $w = (10^3, 10^0, 10^0, 10^1, 10^1, 10^{-3})$, without extense research to find optimal parameters. The agent was trained until it successfully adopted a strategy to achieve the target orbit within the specified tolerance. Figure 18a shows the reward achieved on each experiment by episode, illustrating that Experiment 1 took more time to reach a good policy, explained by the lack of signaling regarding the along-track direction ($\alpha_2$). The fuel, as shown in Figure 18b, shows that the agent learned to save fuel in both experiments.

For evaluation, in the second experiment, the generalization of the agent was tested by switching from a Newtonian attraction model to a spherical harmonic gravity, and using third body forces (Sun and Moon), SRP, and drag force.

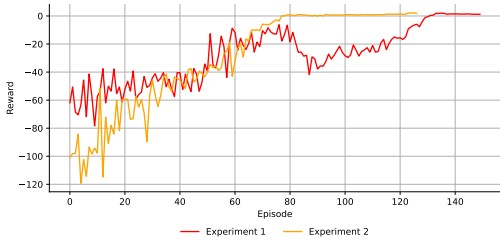 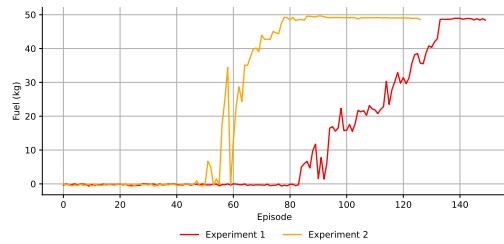

(a) Cumulative Reward for Experiment 1 and Experiment 2.

(b) Final fuel for Experiment 1 and Experiment 2.

Figure 18: Hohmann Maneuver: Cumulative reward and fuel consumption across episodes.

Figure 19 illustrates that in both experiments, the agent achieved the target semi-major axis while also correcting some other orbital elements. However, in Experiment 1, the agent failed to bring the $h_y$ component within tolerance. In contrast, the results of the second experiment show a closer approximation to the optimal maneuver. Notably, when using more realistic dynamics by adding perturbative forces, the agent is also able to reach the target orbit, even with a constant change in orbital parameters created by these perturbations.

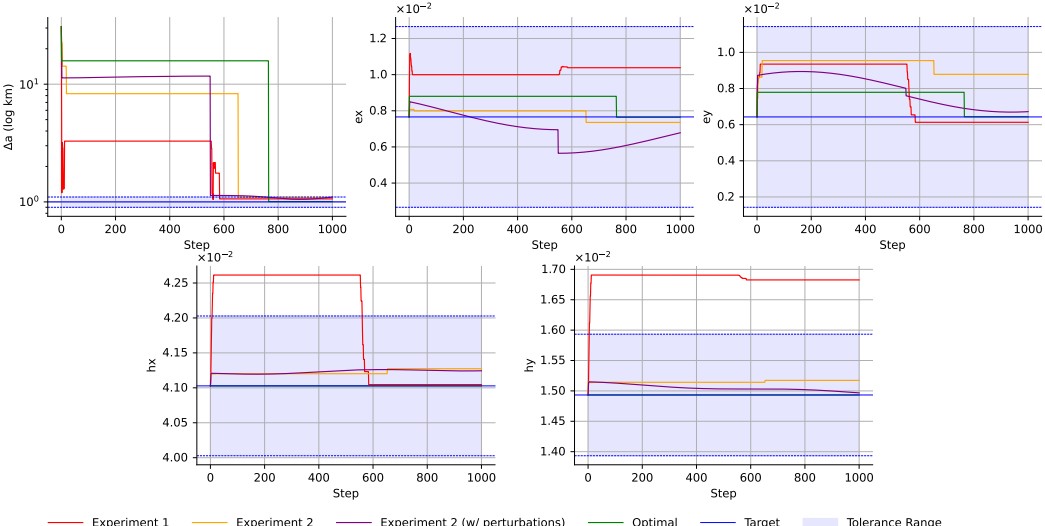

Figure 19: Evolution of equinoctial elements ($\Delta a, e_x, e_y, h_x, h_y$) in the Hohmann maneuver for the first experiment (red), second experiment (yellow), second experiment with realistic perturbations (purple) and optimal maneuver (green). $\Delta a$ corresponds to the absolute difference between the target and current semi-major axis.

Nevertheless, both experiments demonstrate that the agent incorrectly learned to apply thrusts that alter the inclination, as shown in Figure 20, with thrusts being applied in the radial ($r$) and cross-track ($w$) directions. Finally, Figure 21 illustrates that the agent's decision policy $\delta$ is nearly optimal, with thrust activations occurring both at the beginning of the episode and near $t_H$.

An additional analysis related to the error between the optimal control approach and five successful attempts was performed when using the agent trained in Experiment 1 and Experiment 2 with different levels of perturbations. In Table 12, we measure the average fuel error (in kg) and standard deviation of these attempts to the optimal approach under simplified dynamics. Results suggest that both experiments are able to achieve near optimal control while using, on average, no more than 1 kg of additional fuel, which is a small portion of all the available fuel.

Additionally, we compare the altitude error (in km) of those attempts by step, as seen in Table 13. The steps shown in the table were selected based on the intervals where agents perform thrusts (as

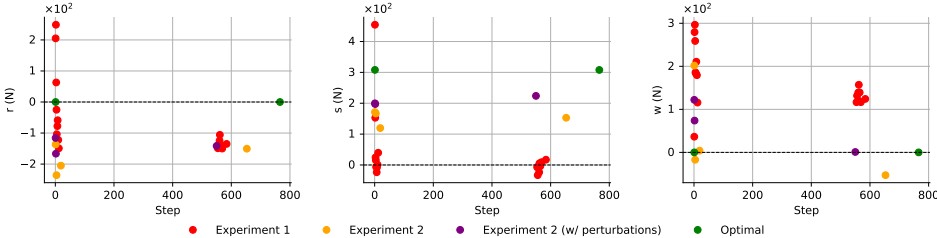

Figure 20: Comparison of the applied thrust $(r, s, w)$, in Newtons, for the first experiment (red), second experiment (yellow), second experiment with realistic perturbations (purple) and optimal maneuver (green).

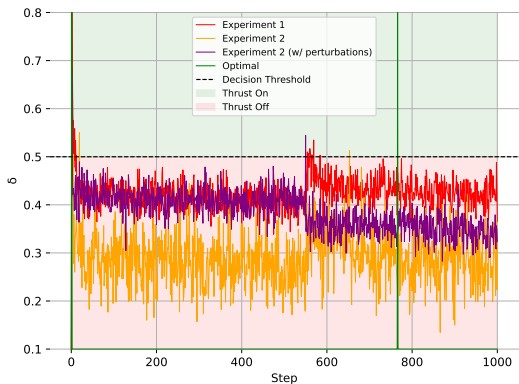

Figure 21: Comparison of the agents decisions ($\delta$) to apply thrust during a full episode.

seen in Figure 21). These results suggest that after the first optimal thrust (step 0), Experiment 2 generally better approximates the optimal altitude compared to Experiment 1, but with more variance. After the second optimal thrust (step 765), while Experiment 1 contains a smaller error in terms of altitude until the end of the episode, Experiment 2 contains a lot more variance, suggesting that in this case the agent arrives at the target orbit (given in equinoctial elements) through different trajectories, indicating a better understanding of the dynamics than Experiment 1. These results also suggest that shaping the reward to give more importance to the semi-major axis may be needed, since the agents where trained to minimize the error of all equinoctial elements.

Table 12: Average fuel difference (in kg) and standard deviation between each experiment and the optimal maneuver under simplified dynamics (Newtonian attraction), using five successful attempts.

| Exp 1 No Perturbations | Exp 1 Perturbed | Exp 2 No Perturbations | Exp 2 Perturbed |
|---|---|---|---|
| $0.34_{(0.02)}$ | $0.36_{(0.01)}$ | $0.85_{(0.31)}$ | $0.86_{(0.16)}$ |

Overall, this experiment demonstrates that even when trained without perturbations, the policy successfully generalizes to real dynamics, adapting to complex gravity fields, third-body forces, SRP, and atmospheric drag. This is particularly significant in LEO, where gravity and drag are the dominant influences on a satellite's trajectory.

Additionally, the impact of providing a more informative reward function is evident, as Experiment 2 more quickly learns to avoid performing maneuvers outside the orbital plane.

### E.4.4  Challenges and Future Improvements

Despite the agent's ability to generalize to realistic dynamics, it still encounters difficulties reaching the target orbit consistently, as actions in PPO are stochastic by nature, whereas the Hohmann maneuver requires precise execution. Future improvements for this mission include employing rein-

Table 13: Average altitude difference (in km) and standard deviation between each experiment and the optimal maneuver under simplified dynamics (Newtonian attraction), using five successful attempts.

| Step | Exp 1 No Perturbations | Exp 1 Perturbed | Exp 2 No Perturbations | Exp 2 Perturbed |
|------|------------------------|-----------------|------------------------|-----------------|
| 0 | $0.00_{(0.00)}$ | $0.00_{(0.00)}$ | $0.00_{(0.00)}$ | $0.00_{(0.00)}$ |
| 1 | $0.01_{(0.00)}$ | $0.01_{(0.00)}$ | $0.01_{(0.00)}$ | $0.01_{(0.00)}$ |
| 2 | $0.03_{(0.00)}$ | $0.03_{(0.00)}$ | $0.02_{(0.02)}$ | $0.02_{(0.01)}$ |
| 3 | $0.07_{(0.00)}$ | $0.06_{(0.01)}$ | $0.04_{(0.04)}$ | $0.04_{(0.01)}$ |
| 4 | $0.11_{(0.01)}$ | $0.10_{(0.01)}$ | $0.06_{(0.06)}$ | $0.06_{(0.01)}$ |
| | | ... | | |
| 553 | $2.57_{(1.51)}$ | $10.13_{(1.66)}$ | $2.43_{(2.07)}$ | $8.10_{(3.60)}$ |
| 554 | $2.61_{(1.51)}$ | $10.13_{(1.66)}$ | $2.35_{(2.07)}$ | $8.07_{(3.58)}$ |
| 555 | $2.65_{(1.51)}$ | $10.12_{(1.67)}$ | $2.28_{(2.07)}$ | $8.04_{(3.55)}$ |
| | | ... | | |
| 765 | $10.73_{(1.30)}$ | $5.10_{(1.12)}$ | $17.49_{(2.97)}$ | $5.66_{(4.63)}$ |
| 766 | $10.78_{(1.29)}$ | $5.05_{(1.12)}$ | $17.59_{(2.97)}$ | $5.73_{(4.63)}$ |
| 767 | $10.83_{(1.29)}$ | $5.00_{(1.11)}$ | $17.69_{(2.98)}$ | $5.80_{(4.63)}$ |
| | | ... | | |
| 998 | $16.30_{(0.96)}$ | $3.74_{(0.75)}$ | $30.74_{(3.53)}$ | $13.33_{(6.08)}$ |
| 999 | $16.30_{(0.96)}$ | $3.76_{(0.75)}$ | $30.74_{(3.53)}$ | $13.32_{(6.09)}$ |

Table 14: Chase Target: Training hyperparameters.

| Parameter | Value |
|-----------|-------|
| Actor learning rate | 0.00001 |
| Critic learning rate | 0.0001 |
| GAE $\lambda$ | 0.95 |
| Epochs | 5 |
| Discount factor ($\gamma$) | 0.99 |
| Clip ($\epsilon$) | 0.1 |
| Initial Standard Deviation | 0.4 |
| Standard Deviation Decay Rate | 10000 steps |
| Standard Deviation Decay Amount | 0.05 |
| Minimum Standard Deviation | 0.05 |
| Experiences for Training | 4096 |
| Batch Size | 64 |

forcement learning algorithms specifically designed for discrete actions and exploring hyperparameter tuning, since thrusts were not always applied at optimal moments.

## E.5 Chase Target

### E.5.1 Definition

Orbital transfer missions typically involve placing a satellite into a specific stationary orbit, as seen in the Hohmann transfer, or performing station-keeping maneuvers to maintain a nominal orbit. However, targets in such missions can also be nonstationary, as in scenarios where the objective is to rendezvous with moving bodies such as debris or active satellites. Many missions focus on approaching dynamic targets, like comets, particularly when the goal is scientific observation and study.

### E.5.2 Environment Setup

**Objective and Environment Characteristics.** In this experiment, we have an agent/satellite, called follower, that starts on an orbit characterized by the following Keplerian elements $(a_F, e_F, i_F, \omega_F, \Omega_F, M_F) = (10000 + R_E \text{ km}, 0.1, 5.0°, 10.0°, 10.0°, 10.0°)$. Additionally, there is a non-maneuverable body, called target, that starts on an orbit with a much larger altitude and less inclination than the follower: $(a_L, e_L, i_L, \omega_L, \Omega_L, M_L) = (40000 +$

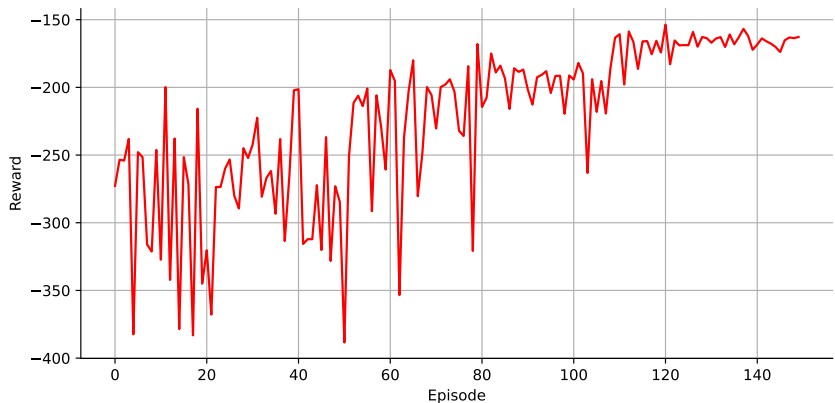

Figure 22: Chase Target: Cumulative reward per episode.

$R_E$ km, $0.001, 5.0°, 10.0°.10.0°, 10.0°$). To avoid singularities, we use equinoctial parameters to represent the orbits of the follower $(a_F, e_{x_F}, e_{y_F}, h_{x_F}, h_{y_F})$ and the leader $(a_L, e_{x_L}, e_{y_L}, h_{x_L}, h_{y_L})$. The objective is for the follower to approximate the leader.

The satellite is characterized by being spherical with a radius of 5 m, a mass of 650 kg (from which 150 kg are fuel), and has a propulsion system containing a specific impulse of $I_{sp} = 3000$ s. Finally, the dynamics for training are composed of Newtonian attraction without additional perturbations, and each episode contains 2000 steps of 500 seconds.

**Action Space.** The satellite thrust system is capable of performing thrusts in any direction with a maximum magnitude of 30 N, that is, limited by $(T_{\max}, \theta_{\max}, \phi_{\max}) = (30, \pi, 2\pi)$.

**Observation Space.** Since the orbital elements of the leader remain static throughout the episode (besides the anomaly $M_L$), the observation space of the agent becomes smaller, as it does not need to know those elements: $o_t = (a_F, e_{x_F}, e_{y_F}, h_{x_F}, h_{y_F}, M_F, M_L, f_t)$, where $f_t$ represents the current fuel.

**Reward Function.** The reward function aims to minimize the error between each equinoctial element, including the anomaly. We first define the difference between each element $e$ as:

$$\Delta e = \begin{cases} \frac{\sqrt{(e_L - e_F)^2}}{e_L} & \text{if } e = a \\ \sqrt{(e_L - e_F)^2} & \text{if } e \in \{e_x, e_y, h_x, h_y\} \\ \text{atan2}(\sin(e_L - e_F), \cos(e_L - e_F)) & \text{if } e = M \end{cases} \quad (32)$$

The final reward function weights each difference by a factor $\alpha_e$:

$$r_t = -\left(\alpha_a|\Delta a| + \alpha_{e_x}|\Delta e_x| + \alpha_{e_y}|\Delta e_y| + \alpha_{h_x}|\Delta h_x| + \alpha_{h_y}|\Delta h_y| + \alpha_M|\Delta M|\right). \quad (33)$$

The error for the semi-major axis is normalized by the target element (leader) due of its large scale, and the error in the anomaly needs to account for the periodical nature of the element, where errors vary between $[0, \pi]$.

### E.5.3 Results

For training, the following weights were used, without extensive research for optimal parameters: $(\alpha_a, \alpha_{e_x}, \alpha_{e_y}, \alpha_{h_x}, \alpha_{h_y}, \alpha_M) = (10^0, 10^{-3}, 10^{-3}, 10^{-2}, 10^{-2}, 10^{-6})$. The evolution of training is shown in Figure 22.

To assess the performance and generalization ability of the follower, we conducted three experiments incorporating all perturbative forces available in OrbitZoo, including harmonic gravity fields, third-body forces (Sun and Moon), SRP, and atmospheric drag.

As detailed in Table 15, the first experiment evaluates the model under the same initial orbital elements used during training but with all forces applied. In the second experiment, the agent starts in an orbit

Table 15: Chase Target: Initial Keplerian elements of the follower for the different experiments.

| Experiment | a | e | i | $\omega$ | $\Omega$ | M |
|---|---|---|---|---|---|---|
| Experiment 1 | $10000 + R_E$ km | 0.1 | 5.0° | 10.0° | 10.0° | 10.0° |
| Experiment 2 | $15000 + R_E$ km | 0.5 | 8.0° | 10.0° | 10.0° | 90.0° |
| Experiment 3 | $5000 + R_E$ km | 0.001 | 5.0° | 10.0° | 180.0° | 180.0° |

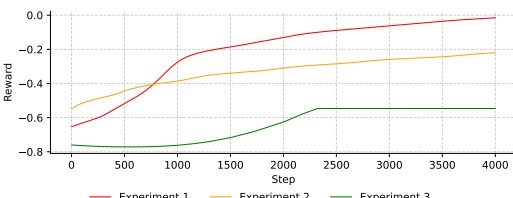

Figure 23: Reward per step, for each experiment. A reward of 0.0 indicates that the follower contains exactly the same orbital elements as the target, including anomaly.

with a larger semi-major axis, higher eccentricity and inclination, and a different initial anomaly. Finally, the third experiment tests the agent in a lower orbit (smaller semi-major axis), with reduced eccentricity and a new initial anomaly. While Experiment 1 assesses the agent's generalization to different perturbative forces, Experiment 2 evaluates its adaptability when starting in an orbit with significantly different characteristics. Lastly, Experiment 3 tests the agent's ability to escape Earth's stronger gravitational potential by applying larger thrusts, which requires greater fuel consumption.

The most straightforward way to assess the agent's performance is by tracking the immediate reward over a full episode. Since the reward represents an error, the agent's objective is to minimize it. As shown in Figure 23, the agent effectively reduces the error in all experiments, demonstrating that it has learned a successful policy for reaching the target. In Experiment 3, the reward remains constant because the agent exhausts all available fuel, maintaining the same orbit until the episode ends. This outcome is expected, as escaping Earth's gravitational pull requires significantly more force when starting at an altitude of 5000 km compared to 10000 km.

To gain a deeper understanding of the agent's policy and its impact on the reward, we examine the differences in each orbital element, as shown in Figure 24. Across all experiments, the agent prioritizes increasing the semi-major axis while keeping the remaining elements as close as possible to the target.

This behavior is particularly evident in Experiment 3, where the agent initially struggles to increase the semi-major axis because it simultaneously attempts to minimize deviations in other elements while gaining altitude. In contrast, Experiment 1 is the only case where the anomaly closely matches that of the leader, whereas in the other experiments, the agent places greater emphasis on minimizing all elements. This strategy aligns with the weight distribution in the reward function, reflecting the relative importance assigned to each orbital parameter.

This experiment demonstrates that the agent learns an effective policy capable of generalizing thrust maneuvers to initial conditions it did not encounter during training, while also adapting to realistic orbital dynamics with non-conservative forces.

### E.5.4 Challenges and Future Improvements

Despite the agent demonstrating good generalization capabilities, it fails to perform maneuvers effectively when the target orbit's inclination differs significantly from those encountered during training. Additionally, fuel consumption is not explicitly accounted for in the reward function, which negatively impacts the agent's long-term strategy, often resulting in rapid fuel depletion.

Future improvements to this mission could include introducing additional penalties for fuel consumption in the reward function, training the agent with initial random orbits, and further exploring the limits of the agent's strategy.

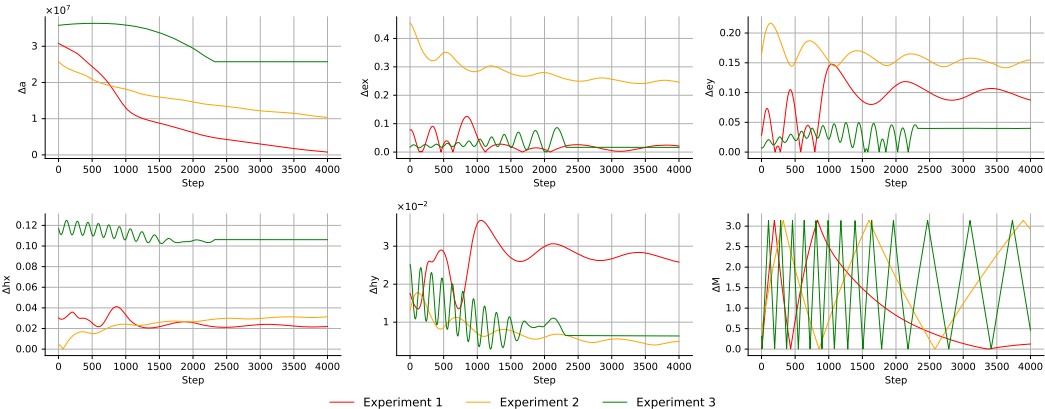

Figure 24: Absolute difference between the follower and leader orbital elements per step, for each experiment.

## E.6 Collision Avoidance

### E.6.1 Definition

Collision avoidance missions often rely on probabilities, particularly the Probability of Collision (PoC) [2] between two objects. These include a maneuverable body, referred to as the target with radius $R_T$, and a non-maneuverable body, referred to as the chaser with radius $R_C$. As the propagation of these bodies is simulated, inherent uncertainties in their current positions and velocities grow over time. These uncertainties form the basis for calculating PoC and generating CDMs.

The PoC assumes that the trajectories of the bodies follow linear motion during the very short period of possible collision, neglects uncertainties in their current velocities, and considers that position uncertainties remain constant over the short timeframe [80]. By defining the relative position ($r = r_T - r_C$) and relative velocity ($\dot{r} = \dot{r}_T - \dot{r}_C$) vectors, the exact Time of Closest Approach (TCA) within the step can be calculated as:

$$\text{TCA} = -\frac{r^T \dot{r}}{\dot{r}^T \dot{r}}. \tag{34}$$

A simplified coordinate system, known as the RTN frame, is then introduced, with the chaser positioned at the origin. The unit vectors defining the RTN frame are computed as follows:

$$\hat{R} = \frac{r_C}{\|r_C\|} \qquad \hat{N} = \frac{r_C \times \dot{r}_C}{\|r_C \times \dot{r}_C\|} \qquad \hat{T} = \hat{N} \times \hat{R}, \tag{35}$$

where $\hat{R}$, $\hat{T}$ and $\hat{N}$ represent the radial, transverse, and normal directions, respectively. These unit vectors form the columns of the $3 \times 3$ transformation matrix $Q = [\hat{R}, \hat{T}, \hat{N}]$, which is used to transform the original relative position, velocity, and position covariances of the target and chaser ($\Sigma_T$ and $\Sigma_C$) into the RTN reference frame:

$$r_{\text{RTN}} = Qr \qquad \dot{r}_{\text{RTN}} = Q\dot{r} \qquad \Sigma_{\text{T,RTN}} = Q\Sigma_T Q^T \qquad \Sigma_{\text{C,RTN}} = Q\Sigma_C Q^T. \tag{36}$$

Next, an orthogonal coordinate system is established based on the relative vectors:

$$\hat{I} = \frac{\dot{r}_{\text{RTN}}}{\|\dot{r}_{\text{RTN}}\|} \qquad \hat{J} = \frac{\dot{r}_{\text{T,RTN}} \times \dot{r}_{\text{C,RTN}}}{\|\dot{r}_{\text{T,RTN}} \times \dot{r}_{\text{C,RTN}}\|} \qquad \hat{K} = \hat{I} \times \hat{J}, \tag{37}$$

where $\hat{I}$, $\hat{J}$ and $\hat{K}$ are the unit vectors forming the orthogonal system. Similarly to the orbital plane encountered in a Keplerian orbit (as explained in B.1), given the assumption of negligible velocity uncertainty, only the $\hat{J}$ and $\hat{K}$ axes (defining the conjunction plane) are relevant for the analysis. This simplifies the problem to two dimensions by introducing a projection matrix $C = [\hat{J}, \hat{K}]$.

Under the assumption that uncertainties follow a Gaussian distribution, the distribution's parameters are given by $\mu = C^T r_{\text{RTN}}$ and $\Sigma = C(\Sigma_{\text{T,RTN}} + \Sigma_{\text{C,RTN}})C^T$. The PoC is then calculated as the

integral over a circle centered at the origin with radius $R = R_T + R_C$:

$$\text{PoC} = \frac{1}{2\pi|\Sigma|^{\frac{1}{2}}} \int_{-R}^{R} \int_{-\sqrt{R^2-y^2}}^{\sqrt{R^2-y^2}} e^{-\frac{1}{2}\delta^T \Sigma^{-1} \delta} dz dy, \tag{38}$$

where

$$\delta = \begin{bmatrix} y \\ z \end{bmatrix} - \mu$$

and $|\Sigma|$ represents the determinant of matrix $\Sigma$.

In this approach, we use a faster computation method compared to the traditional one, as outlined in [25]. According to NASA, a PoC value greater than $10^{-4}$ indicates a high collision risk, requiring maneuvers to address it. However, SpaceX places its threshold at $10^{-6}$, which is the one used in this experiment.

### E.6.2 Environment Setup

**Objective and Environment Characteristics.** We consider a scenario involving two spherical and isotropic bodies: a satellite (target) – with $m_0 = 250$ kg, 50 kg of fuel, $I_{sp} = 3100$ s and $R_T = 10$ meters – and a drifter (chaser) – with $R_C = 5$ meters. The real propagation of the bodies follows a model incorporating Newtonian gravitational attraction and atmospheric drag. However, the satellite is equipped with a system capable of simulating the propagation of both bodies (and their associated uncertainties) using a simplified model based solely on Newtonian gravitational attraction, neglecting drag effects. As expected, this simulator introduces inaccuracies since it does not fully represent the real propagation dynamics. Nonetheless, it provides valuable estimates of key parameters, including the PoC, the TCA, and the miss distance, at any given moment.

The setup of the initial conditions for each episode is critical, as the satellite must begin two days prior to the initial TCA detected by its system, with a PoC exceeding $10^{-6}$. To achieve this, the initial positions of both bodies are set at the same expected location, but with opposing velocities and small uncertainties. At the start of each episode, the states of both bodies are sampled from their respective expected positions and uncertainties, followed by propagating their trajectories backward in time. The objective is for the target to learn a strategy that effectively minimizes the PoC while maintaining a trajectory close to its nominal orbit.

In practice, both bodies are considered to start with the same Keplerian elements $(a, e, i, \omega, \Omega, M) = (2000 + R_E \text{ km}, 0.01, 5.0°, 20.0°, 20.0°, 10.0°)$. However, they start with opposite velocity vectors. There is also an uncertainty of 0.1 m and 0.1 m/s when sampling the initial Cartesian position and velocity of both bodies, respectively.

At each step, the satellite simulates the dynamics of both bodies until the end of the episode, considering their current states and uncertainties. The simulation outputs the TCA, miss distance, and PoC. Each step lasts 15 minutes, with termination occurring 10 steps after the initial TCA. Thrust is applied only during the first 10 seconds of each step.

**Action Space.** As in the Hohmann maneuver mission (E.4) mission, apart from the polar representation of the thrust, the action space also contains the decision ($\delta$) to perform the given thrust: $a_t = (T, \theta, \phi, \delta)$. This action space is limited by $(T_{\max}, \theta_{\max}, \phi_{\max}, \delta_{\max}) = (5, \pi, 2\pi, 1)$.

**Observation Space.** The observation $o_t = (s_t, M_t, s'_t, M'_t, f_t, P_t) \in \mathbb{R}^{14}$ consists of the current equinoctial elements of the satellite ($s_t$) and the drifter ($s'_t$), both bodies mean anomaly ($M_t$ and $M'_t$), together with the current fuel mass ($f_t$) and PoC ($P_t$).

**Reward Function.** The reward function is built in a way to not apply thrusts when there's a small PoC, and apply thrusts that do not deviate too much from the satellite's nominal orbit ($\hat{s}$) when it is higher than $10^{-6}$, while reducing the PoC:

$$r_t = \begin{cases} -\alpha_1 \mathbb{1}_{\delta > 0.5} & , P_{t-1} < 10^{-6} \\ -(\Delta s_t + \alpha_2 \mathbb{1}_{P_t > 10^{-6}}) & , P_{t-1} \geq 10^{-6} \end{cases}, \tag{39}$$

where $\Delta s_t = w^T(s_t - \hat{s})$, with $w^T$ representing a vector of weights, $P_{t-1}$ and $P_t$ are respectively the PoC before and after the action, and $\alpha_1$ and $\alpha_2$ are weights given to each parameter.

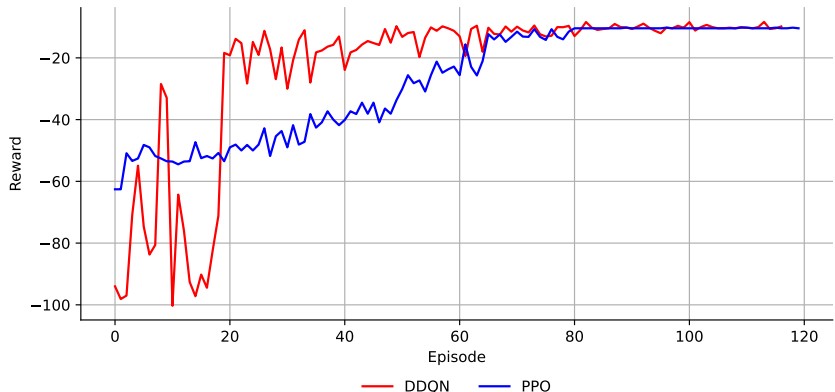

Figure 25: Collision Avoidance: Cumulative reward per episode using DDQN and PPO.

### E.6.3 Results

The weights used to measure the deviation from the nominal orbit ($\Delta s_t$) were set to

$$w = (10^1, 10^{-2}, 10^{-2}, 10^{-1}, 10^{-1}),$$

and the weights assigned to each term in the reward function were $(\alpha_1, \alpha_2) = (10^0, 10^{-1})$. No extensive tuning was performed to find optimal parameter values.

For collision avoidance maneuvers, algorithms that work with discrete actions, such as DQN, work well, as shown by [13]. Because of this, we used both the DDQN (with the implementation shown in section E.1.3) and PPO (with the implementation shown in section E.1.1) algorithms. The training hyperparameters for both algorithms are shown in Table 16.

Table 16: Collision Avoidance: Training hyperparameters for PPO and DDQN.

| PPO | | DDQN | |
|---|---|---|---|
| **Parameter** | **Value** | **Parameter** | **Value** |
| Actor learning rate | 0.0001 | Learning rate | 0.00005 |
| Critic learning rate | 0.001 | Epochs | 1 |
| GAE $\lambda$ | 0.95 | Discount factor ($\gamma$) | 0.95 |
| Epochs | 5 | $\tau$ | 0.001 |
| Discount factor ($\gamma$) | 0.95 | Memory Capacity | 10 000 |
| Clip ($\epsilon$) | 0.5 | Initial $\epsilon$ | 0.5 |
| Initial Standard Deviation | 0.2 | $\epsilon$ Decay Rate | 1000 steps |
| Standard Deviation Decay Rate | 5000 steps | $\epsilon$ Decay Amount | 0.05 |
| Standard Deviation Decay Amount | 0.05 | Minimum $\epsilon$ | 0.05 |
| Minimum Standard Deviation | 0.05 | Batch Size | 256 |
| Experiences for Training | 256 | Update Target Frequency | 10 Online updates |
| Batch Size | 64 | | |

Since DDQN only handles discrete actions, we chose to consider the maximum application of thrust in each of the directions in a 3D space, together with the non-action, as shown in Table 17.

The training progression for both algorithms is illustrated in Figure 25, where PPO demonstrates more stable convergence compared to DQN. To evaluate the agent's performance, we examine its decision-making behavior through key metrics such as the PoC, miss distance, and TCA. Furthermore, to assess the agent's generalization capabilities, we activate all perturbative forces available in OrbitZoo, including higher-order gravitational harmonics, third-body perturbations from the Sun, Moon and Jupiter, SRP, and atmospheric drag. Importantly, the internal satellite simulator dynamics remain unchanged, preserving the realistic limitation that the agent operates under incomplete information about the environment.

Table 17: Possible actions in DDQN.

| Action | Thrust $(T, \theta, \phi, \delta)$ | Direction |
|---|---|---|
| 0 | $(5, 0, 0, 1)$ | Forward |
| 1 | $(5, \pi/2, 0, 1)$ | Left (radial out) |
| 2 | $(5, \pi, 0, 1)$ | Behind |
| 3 | $(5, \pi/2, \pi, 1)$ | Right (radial in) |
| 4 | $(5, \pi/2, \pi/2, 1)$ | Up |
| 5 | $(5, \pi/2, 3\pi/2, 1)$ | Down |
| 6 | $(0, 0, 0, 0)$ | Nothing |

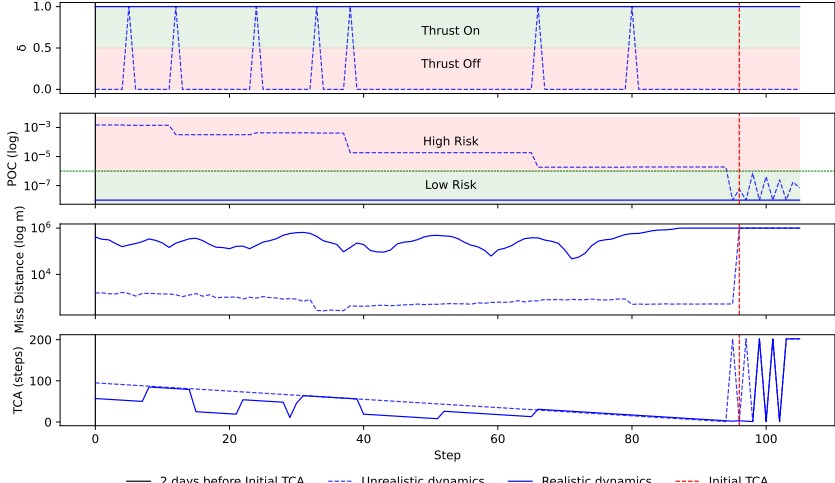

Figure 26: Collision Avoidance using DDQN: Agent decision to apply thrust ($\delta > 0.5$), Probability of Collision (PoC), Miss Distance, and Time of Closest Approach (TCA) by step.

The DDQN results (Figure 26) show that, under unrealistic dynamics, the agent effectively reduces the collision risk to a low level before the Time of Closest Approach (TCA) by occasionally activating thrust. Under realistic dynamics, although the agent experiences a consistently low collision risk throughout the episode, it applies thrust almost continuously. A detailed examination of the orbital elements in Figure 27 reveals that, under the unrealistic dynamics encountered during training, the agent performs extremely well by maintaining the nominal orbit. In contrast, under realistic dynamics, the agent reduces its semi-major axis by approximately 40 km while keeping the eccentricity components close to those of the nominal orbit.

In contrast to DDQN, the results from PPO (Figure 28) indicate that the agent learns a strategy of applying thrust early in the episode – precisely when the collision risk is highest. This early intervention effectively reduces the risk for the remainder of the episode. Notably, the agent's

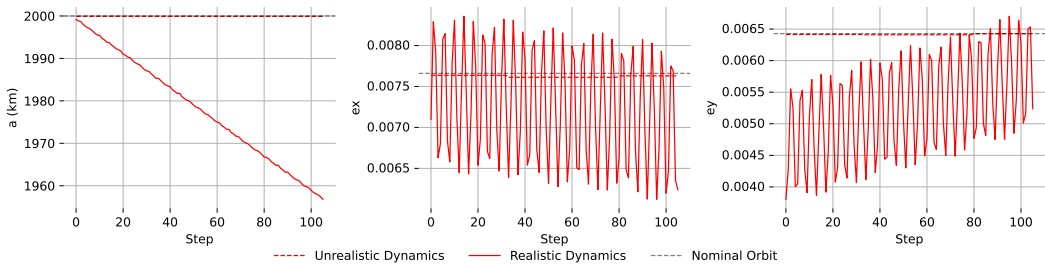

Figure 27: Collision Avoidance using DDQN: Evolution of the semi-major axis (a) and eccentricity components ($e_x$ and $e_y$) of the orbit by step, using unrealistic and realistic dynamics.

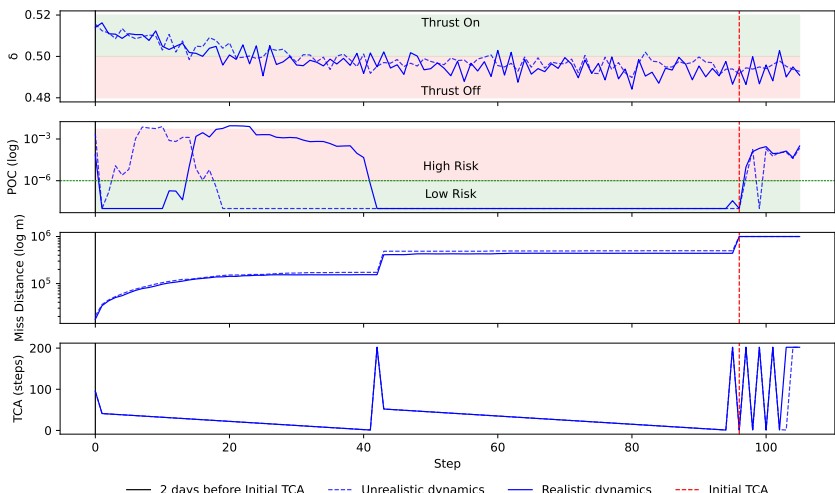

Figure 28: Collision Avoidance using PPO: Agent decision to apply thrust ($\delta > 0.5$), Probability of Collision (PoC), Miss Distance, and Time of Closest Approach (TCA) by step.

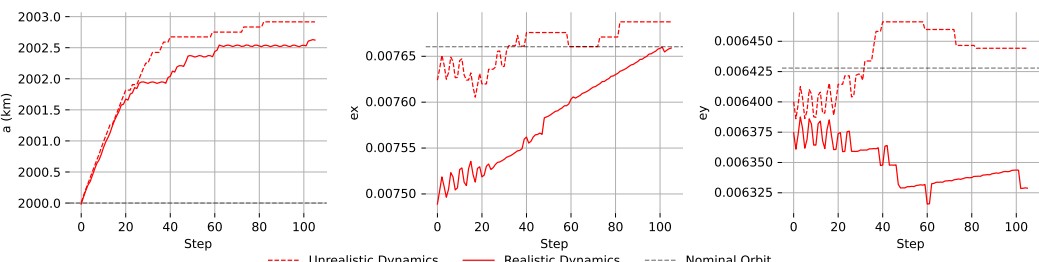

Figure 29: Collision Avoidance using PPO: Evolution of the semi-major axis (a) and eccentricity components ($e_x$ and $e_y$) of the orbit by step, using unrealistic and realistic dynamics.

behavior closely resembles what would be expected under realistic orbital dynamics, demonstrating the effectiveness of the learned strategy. A closer examination of the orbital elements in Figure 29 shows that the agent increases the semi-major axis by approximately 3 km while maintaining the eccentricity components near those of the nominal orbit.

From these experiments, we conclude that although both agents learn effective policies under the unrealistic dynamics used during training, PPO demonstrates superior performance under realistic dynamics. This improvement is largely attributed to the continuous action space in PPO, which provides greater flexibility in executing maneuvers.

### E.6.4 Challenges and Future Improvements

While both algorithms demonstrate some ability to generalize to realistic dynamics, they struggle with the inherent instability of the satellite simulation, particularly the sharp spikes in probability of collision (PoC) near the Time of Closest Approach (TCA). Additionally, the agents have difficulty generalizing their strategies when initialized on orbits that differ from those encountered during training. Although DDQN is unable to fully generalize to realistic dynamics, it shows potential by maintaining consistent eccentricity components, suggesting a partial understanding of the objective.

Future improvements could include training with a randomized time horizon for maneuver completion, initializing agents from a broader range of orbits, and enhancing the observation space to include more informative features. These modifications would reduce uncertainty and support more effective decision-making. Additionally, providing more detailed information to each body's internal simulation could further enable robust strategy learning. The DDQN architecture may also benefit

from the integration of recurrent networks, as demonstrated by [13], which are capable of capturing temporal dependencies critical for decision-making in dynamic environments.

### E.7 GEO Constellation

#### E.7.1 Definition

Unlike LEO and MEO, which occupy defined altitude ranges, the Geostationary Earth Orbit (GEO) is a specific orbit located 35786 km above Earth's equator. This orbit is significant because any object within it maintains an orbital period equal to Earth's rotational period, making it appear stationary to observers on Earth.

There are various possible formations for constellations of satellites in geostationary orbit (GEO). In this work, we focus on a mission scenario where all agents learn to maintain equal angular separations along the orbit. Specifically, with $n$ satellites, each pair of adjacent satellites should be separated by an anomaly difference of $2\pi/n$ rad.

Given that the satellites share similar observations and are cooperating, we assume that agents can exchange knowledge after training, as discussed in Section C.5. In the context of PPO, this enables agents to share their critic networks after local training by aggregating and averaging model parameters – an operation that could be performed, for example, by a ground station. By forming a global critic, each agent gains a more comprehensive understanding of the environment's overall state, even though it has only partial observations locally.

#### E.7.2 Environment Setup

**Objective and Environment Characteristics.** We define an environment with $n = 4$ agents in GEO, characterized by a semi-major axis of $a_{\text{GEO}} = 35786 + R_E = 42164$ km, where $R_E$ is Earth's equatorial radius, in a near perfect circular orbit. Each agent is a satellite with $m_0 = 250$ kg, 50 kg of which is fuel, and equipped with a thrust with $I_{sp} = 3100$ s.

Each episode contains 500 steps with step sizes of 360 s (approximately 2 revolutions/days), and the dynamics are purely based on Newtonian attraction. At the start of each episode ($t = 0$), agents are initialized with random mean anomalies ($M_0^i, i = \{1, 2, ..., n\}$), representing their positions along the orbit. The objective for the agents is to learn to distribute themselves uniformly around the orbit, maintaining angular separations of $2\pi/n$ radians (90 degrees for $n = 4$).

**Action Space.** Since GEO lies in a single orbital plane near the equator, the problem's dimensionality can be reduced. The action space is simplified to two dimensions: $a_t = (T, \theta)$. This configuration allows agents to perform maneuvers affecting the semi-major axis and eccentricity components of their equinoctial elements while leaving inclination unchanged. The actions are limited to $(T_{\max}, \theta_{\max}) = (5, 2\pi)$.

**Observation Space.** The observation space is also reduced. Each agent observation $o_t = (a_t, e_{xt}, e_{yt}, f_t, M_t^1, M_t^2, M_t^3, M_t^4)$ receives its current semi-major axis, eccentricity components, remaining fuel, and the anomalies of all agents in the environment.

**Reward Function.** To calculate the collective penalty related to the difference in anomalies ($P_M$), we first normalize the anomalies to be within $[0, 2\pi]$, followed by:

$$P_M = \frac{1}{C_2^n} \sum_{i=1}^{n-1} \sum_{j=i+1}^{n} \max\left(0, \frac{2\pi/n - \Delta_{ij}}{2\pi/n}\right), \tag{40}$$

where

$$\Delta_{ij} = \min\left(|M_i - M_j|, 2\pi - |M_i - M_j|\right).$$

The reward function of each agent penalizes differences in altitude (norm of Cartesian position $r$), magnitude of thrust ($T$), and close anomalies:

$$r_t = -(\alpha_1 |a_{\text{GEO}} - \|r\|| + \alpha_2 T + \alpha_3 P_M), \tag{41}$$

where $\alpha_1$, $\alpha_2$ and $\alpha_3$ work as customizable weights. This reward function is extensible to an arbitrary number of agents.

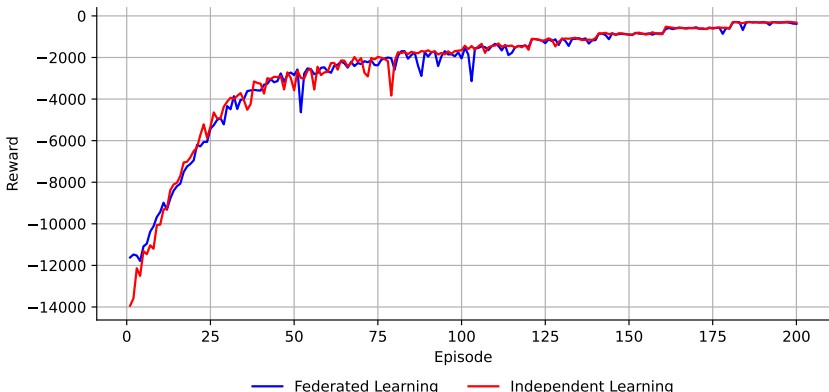

Figure 30: GEO Constellation: Average Cumulative Reward per episode, using independent and federated learning.

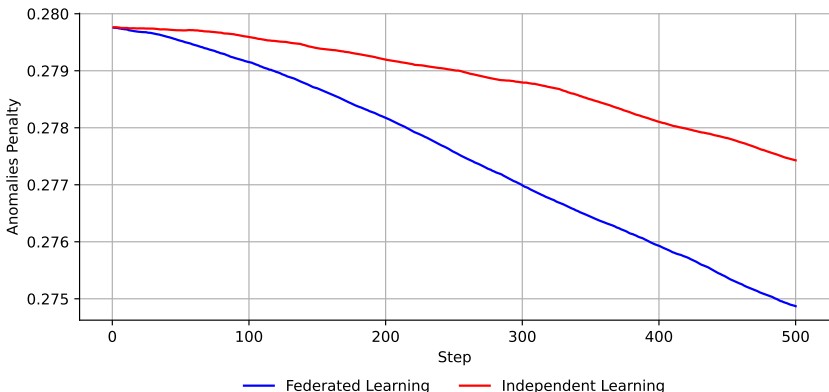

Figure 31: Average Anomaly Penalty ($P_M$) per step over thirty Monte Carlo simulations with realistic dynamics, comparing independent and federated learning. In these simulations, initial anomalies are random but equal for both federated and independent constellations.

### E.7.3 Results

The agents were trained both independently (without communication) and with a centralized critic using the Federated Averaging (FedAvg) algorithm, with the training hyperparameters provided in Table 18.

For training, a combination of parameters that ended up performing rather well was $(\alpha_1, \alpha_2, \alpha_3) = (10^{-8}, 10^1, 10^{-2})$. The training evolution using these parameters for both experiments is shown at Figure 30.

For evaluation, we tested the generalization capability of the policies by introducing realistic dynamics, which include harmonic gravity fields, third body forces, SRP, and drag. To evaluate the performance of both constellations, we conducted Monte Carlo simulations analyzing the evolution of the anomaly penalty, as illustrated in Figure 31.

The results show that agents are capable of reducing the collective anomaly penalty in both scenarios. However, the rate and effectiveness of this reduction depend on the initial constellation configuration. Federated learning demonstrates a faster decrease in the anomaly penalty, highlighting its potential advantage over independent learning. Through the OrbitZoo interface, shown in Figure 2, it is evident that the agents are successfully learning the intended objective, albeit with some imperfections and rather slowly.

In conclusion, although the agents were initially trained independently without sharing information, they successfully learned to maintain equal spacing around the GEO orbit – even under realistic

Table 18: GEO Constellation: Training hyperparameters.

| Parameter | Value |
| --- | --- |
| Actor learning rate | 0.00001 |
| Critic learning rate | 0.0001 |
| GAE $\lambda$ | 0.95 |
| Epochs | 3 |
| Discount factor ($\gamma$) | 0.99 |
| Clip ($\epsilon$) | 0.2 |
| Initial Standard Deviation | 0.5 |
| Standard Deviation Decay Rate | 10000 steps |
| Standard Deviation Decay Amount | 0.05 |
| Minimum Standard Deviation | 0.05 |
| Experiences for Training | 1024 |
| Batch Size | 64 |

dynamics not encountered during training. The federated approach, however, demonstrated clear advantages, enabling more stable training and a better understanding of the global objective in a cooperative setting.

### E.7.4 Challenges and Future Improvements

In some initial configurations – particularly those where agents begin well separated – the anomaly penalty fails to decrease significantly. This limitation appears in both the independent and federated constellations. Addressing it may require tuning hyperparameters more effectively, including adjustments to the reward function or exploring alternative RL architectures. Another challenge stems from the current reward setup: the agents must balance reducing the anomaly penalty with maintaining the nominal altitude, which slows the rate of improvement.

Future work includes implementing vertical federated learning (VFL) using split learning, as the FedAvg approach can become unstable due to its simplistic averaging mechanism. Another key direction is scaling to larger constellations to better reflect realistic operational scenarios.

# F Exploratory Data Analysis of Starlink open data

Starlink's ephemeris files are uploaded three times a day, every 8 hours, with each file containing three days of propagated data. This means that consecutive files have overlapping predictions, with the first 8 hours of each file being considered the most accurate. By analyzing the overlapping predictions, we can evaluate Starlink's own propagation error.

Figures 32 through 35 compare the predictions of a three-day propagation starting on November 27th at around 5:00 AM and ending three days later. The aggregation of 10 consecutive ephemeris files allows for the residual analysis of the initial three-day prediction (from the first ephemeris file) and the subsequent predictions made in the following 9 ephemeris files. This analysis shows that Starlink's data is not error free with prediction errors increase significantly after the 48-hour mark. This is due to a change in propagator by the ephemeris provider. Due to the high computational cost, it shifts to a simpler model that only accounts for the oblateness of the Earth [44].

Figures 36 through 39 show the evolution of the three positional vectors for four different satellites. Since the plotted position is derived from a combination of ephemeris files, using only the most accurate segments from each file, a discontinuity occurs at the transitions between files when one ends and another begins. As expected, the position is oscillatory in nature, reaching its maximum value when the satellite is at apogee (the farthest point from Earth along the major axis of an elliptical orbit) and its minimum value when the satellite is at perigee (the closest point to Earth along the same axis). Figures 36 to 43 illustrate the satellites' positional evolution and orbital trajectories over three days.

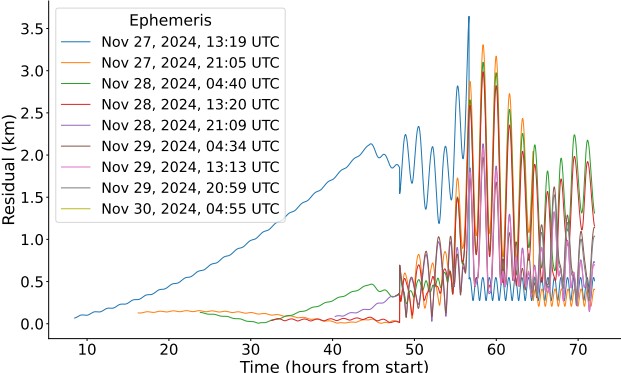

Figure 32: Comparison of predictions between the first ephemeris file, starting on November 27, 2024, at 04:50 UTC, with the subsequent nine ephemeris files to form a complete three-day prediction for the satellite with NORAD ID 44744. The residuals represent the distance between the corresponding points in each comparison.

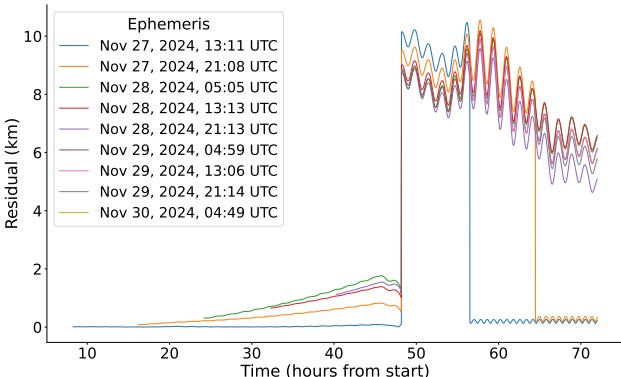

Figure 33: Comparison of predictions between the first ephemeris file, starting on November 27, 2024, at 04:54 UTC, with the subsequent nine ephemeris files to form a complete three-day prediction for the satellite with NORAD ID 44748. The residuals represent the distance between the corresponding points in each comparison.

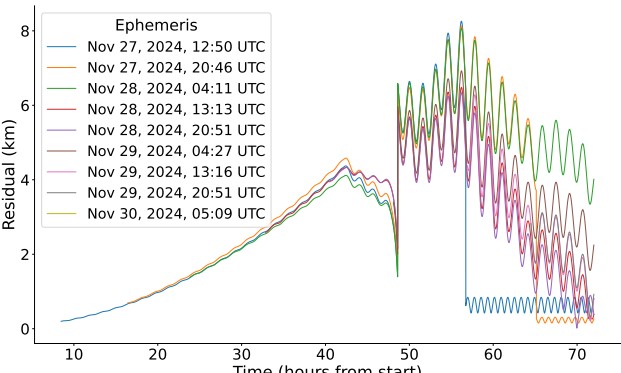

Figure 34: Comparison of predictions between the first ephemeris file, starting on November 27, 2024, at 04:21 UTC, with the subsequent nine ephemeris files to form a complete three-day prediction for the satellite with NORAD ID 44753. The residuals represent the distance between the corresponding points in each comparison.

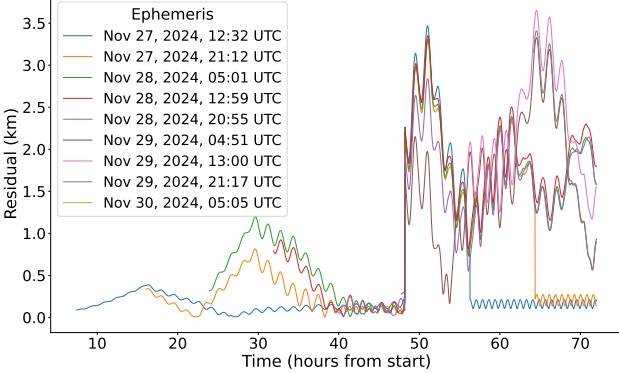

Figure 35: Comparison of predictions between the first ephemeris file, starting on November 27, 2024, at 05:05 UTC, with the subsequent nine ephemeris files to form a complete three-day prediction for the satellite with NORAD ID 44921. The residuals represent the distance between the corresponding points in each comparison.

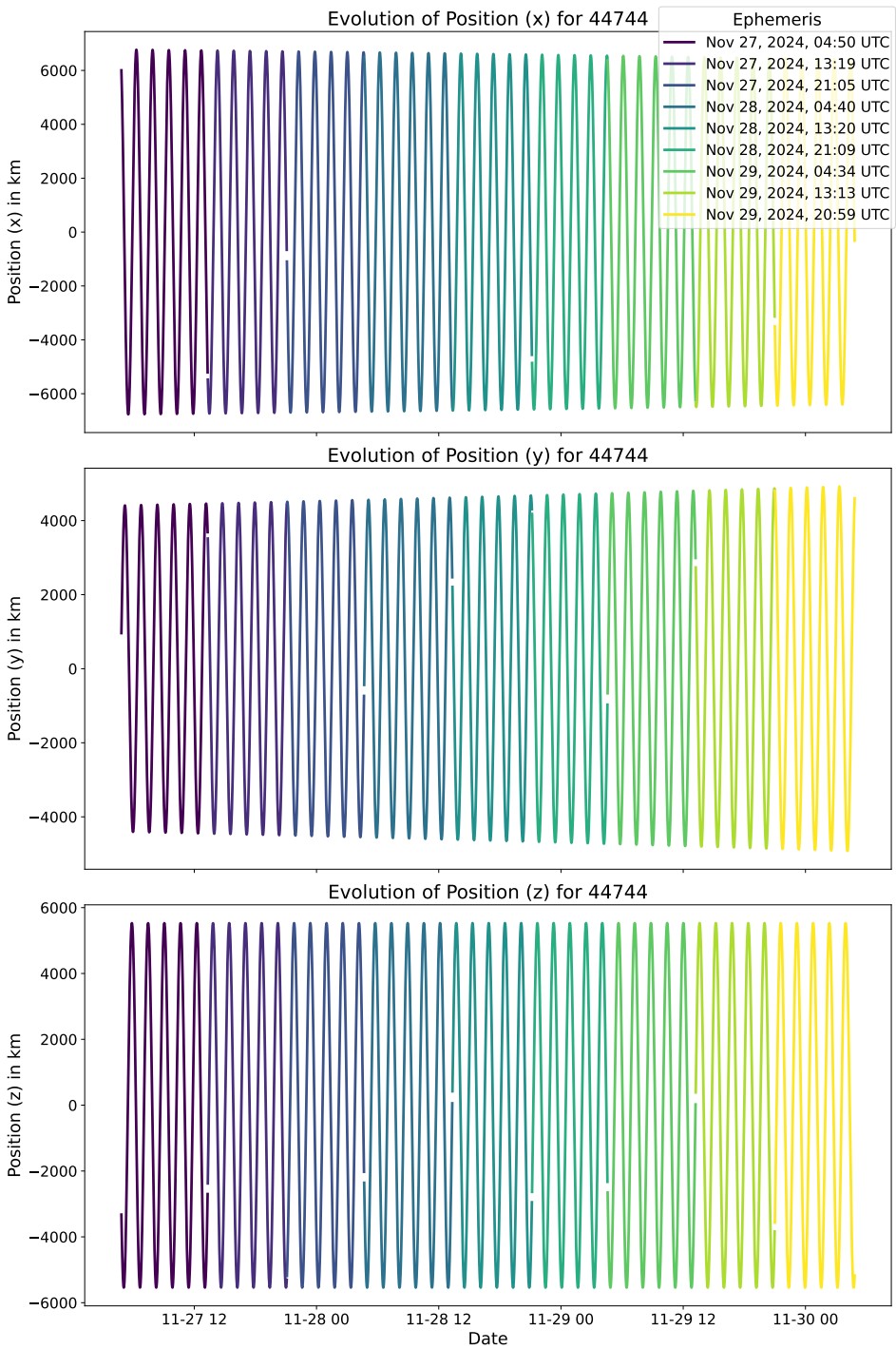

Figure 36: Visualization of the satellite's x, y, and z positional evolution over three days for NORAD ID 44744. Since the plotted position is derived from a combination of ephemeris files, using only the most accurate segments from each file, a discontinuity occurs at the transitions between files when one ends, and another begins.

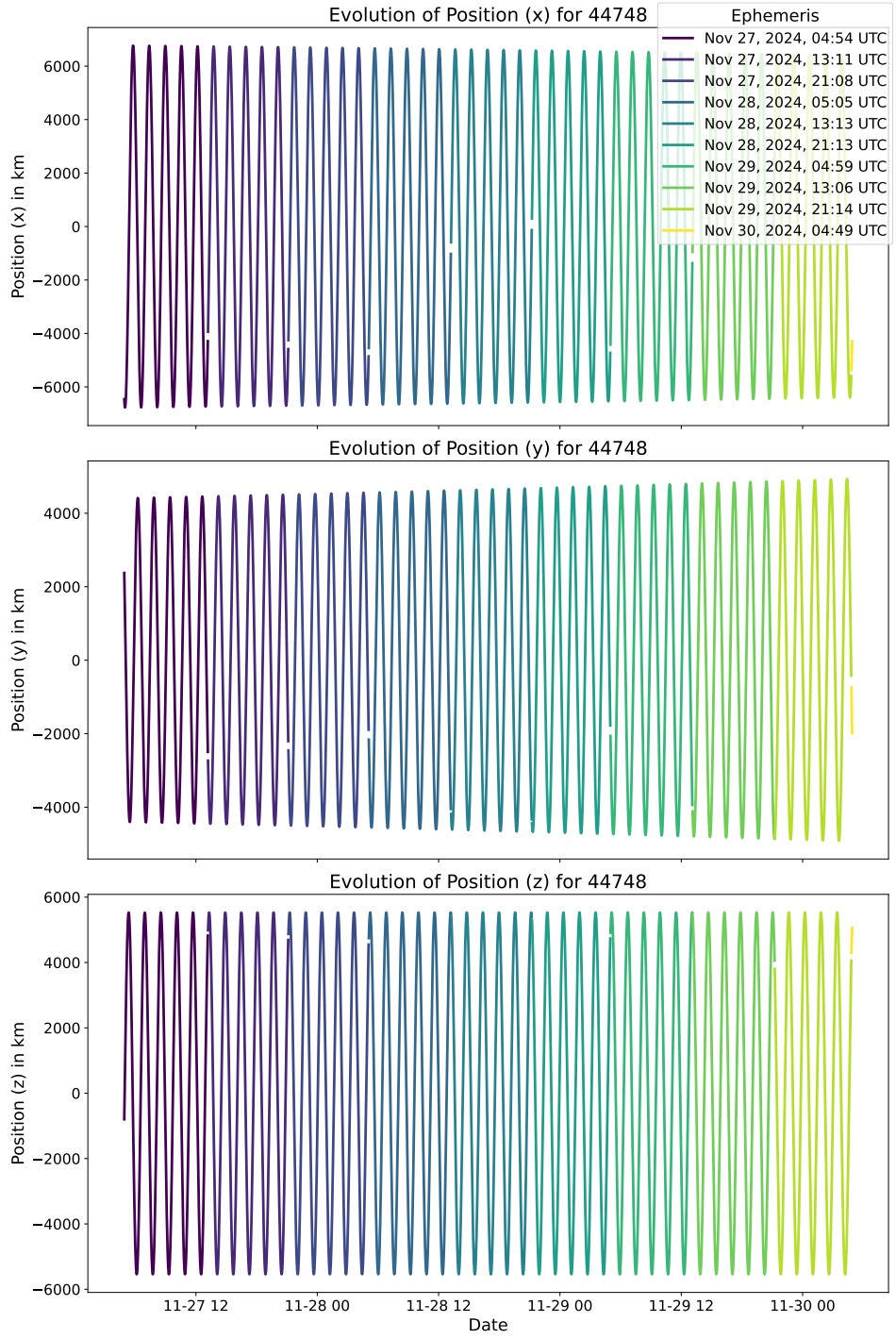

Figure 37: Visualization of the satellite's x, y, and z positional evolution over three days for NORAD ID 44748. Since the plotted position is derived from a combination of ephemeris files, using only the most accurate segments from each file, a discontinuity occurs at the transitions between files when one ends, and another begins.

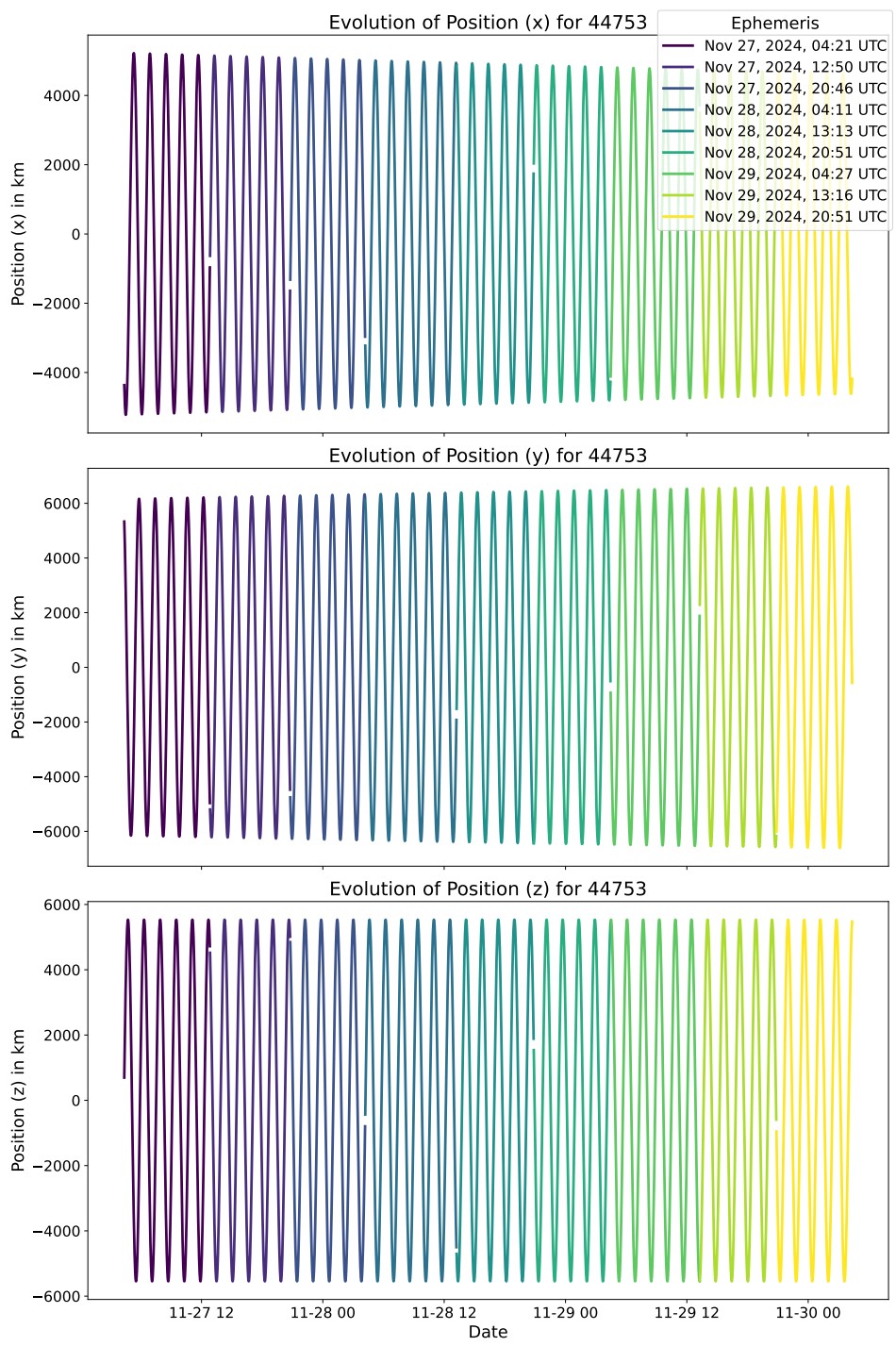

Figure 38: Visualization of the satellite's x, y, and z positional evolution over three days for NORAD ID 44753. Since the plotted position is derived from a combination of ephemeris files, using only the most accurate segments from each file, a discontinuity occurs at the transitions between files when one ends, and another begins.

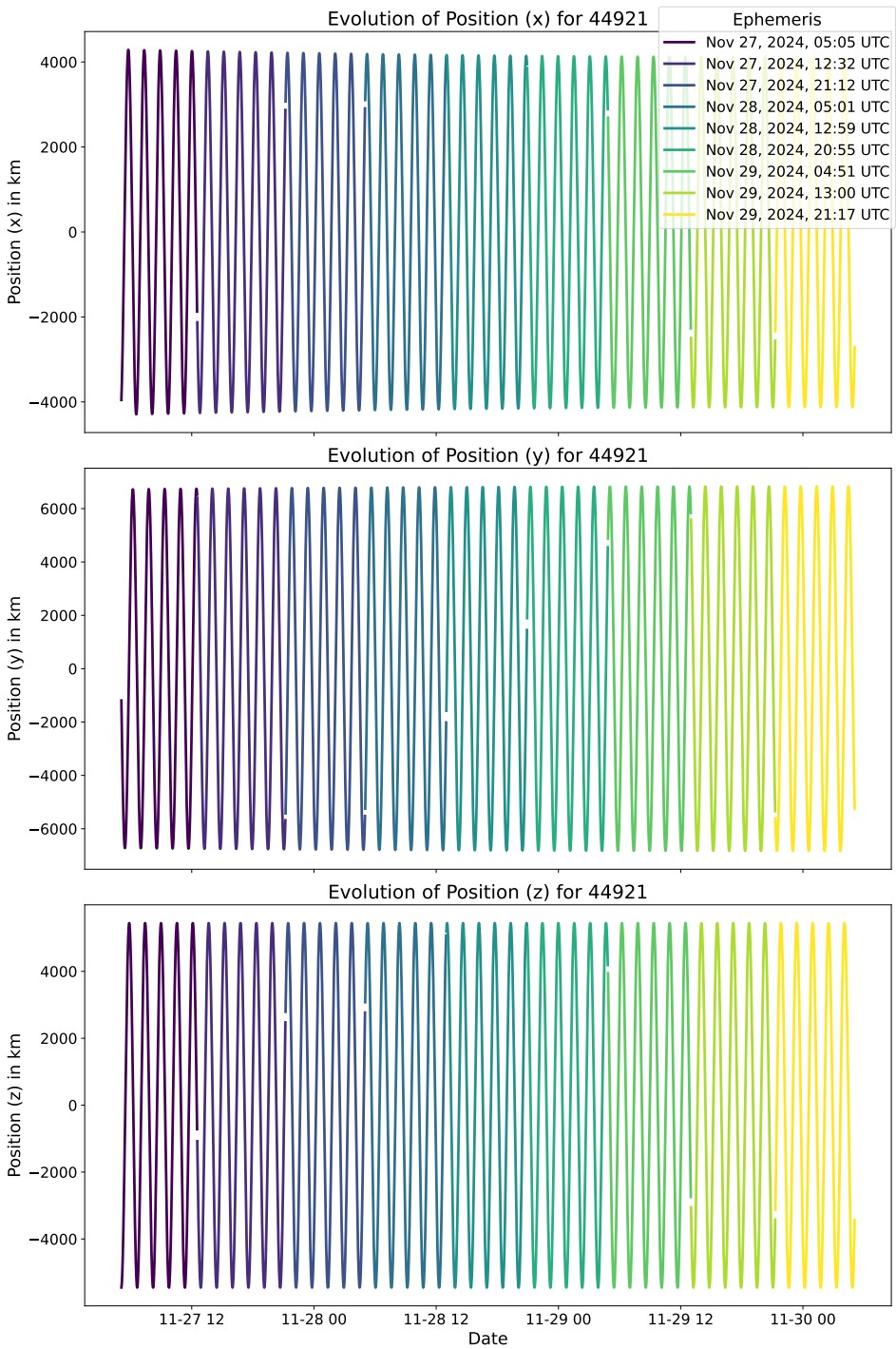

Figure 39: Visualization of the satellite's x, y, and z positional evolution over three days for NORAD ID 44921. Since the plotted position is derived from a combination of ephemeris files, using only the most accurate segments from each file, a discontinuity occurs at the transitions between files when one ends, and another begins.

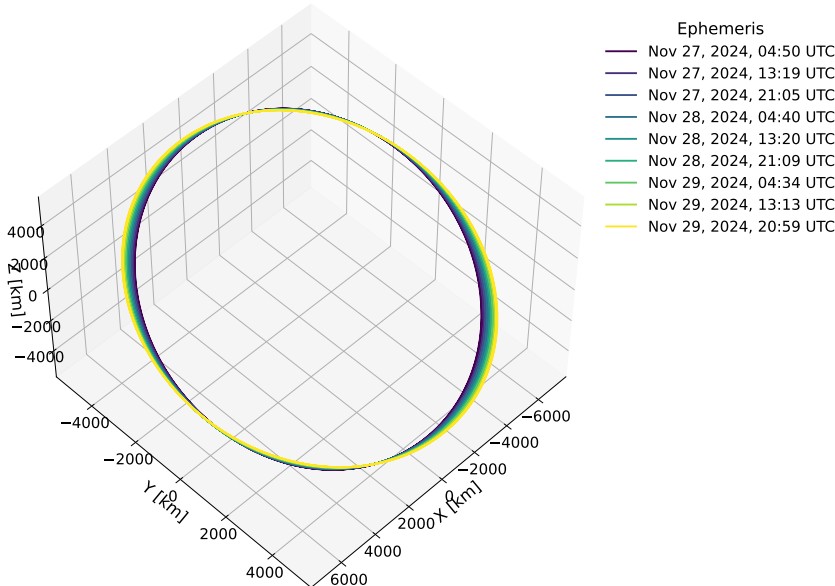

Figure 40: Visualization of the satellite's orbit evolution over three days for satellite with NORAD ID 44744. To display the most accurate three-day trajectory, a combination of consecutive ephemeris files was used, selecting only the most accurate segment from each ephemeris file, roughly the first 8 hours of each file.

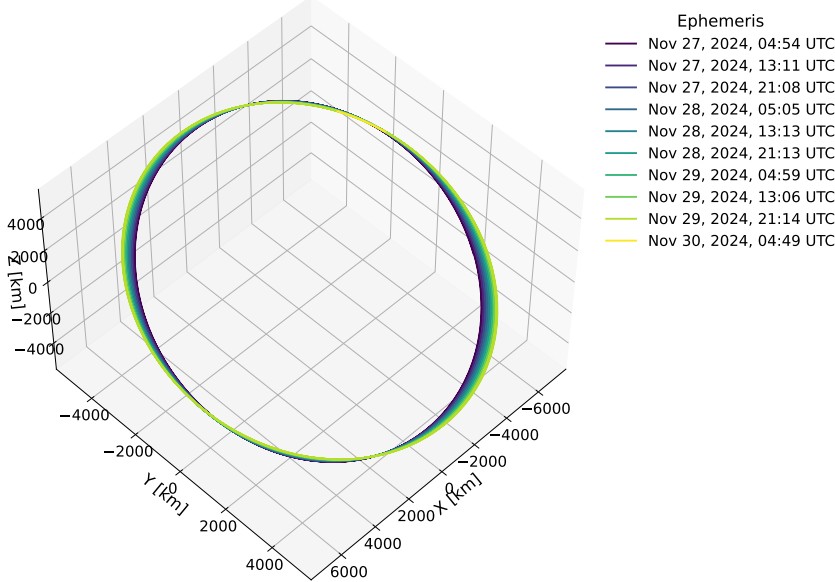

Figure 41: Visualization of the satellite's orbit evolution over three days for satellite with NORAD ID 44748. To display the most accurate three-day trajectory, a combination of consecutive ephemeris files was used, selecting only the most accurate segment from each ephemeris file, roughly the first 8 hours of each file.

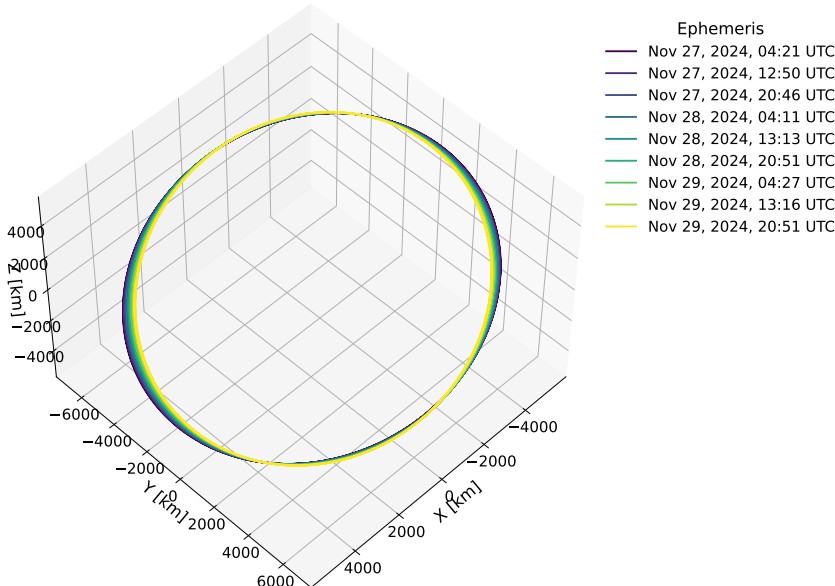

Figure 42: Visualization of the satellite's orbit evolution over three days for satellite with NORAD ID 44758. To display the most accurate three-day trajectory, a combination of consecutive ephemeris files was used, selecting only the most accurate segment from each ephemeris file, roughly the first 8 hours of each file.

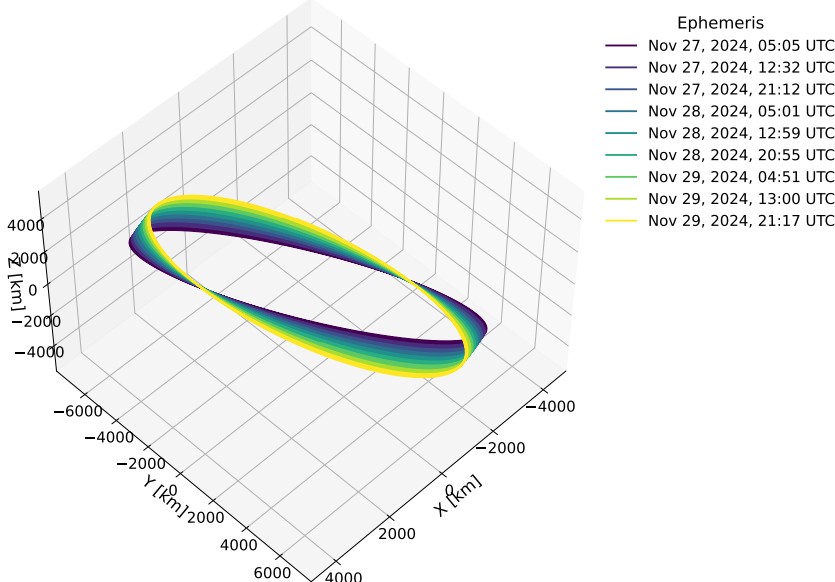

Figure 43: Visualization of the satellite's orbit evolution over three days for satellite with NORAD ID 44921. To display the most accurate three-day trajectory, a combination of consecutive ephemeris files was used, selecting only the most accurate segment from each ephemeris file, roughly the first 8 hours of each file.

# G  Comparison of Orekit Propagation and Ephemerides

To validate Orekit's propagator, physical parameters were optimized using Bayesian optimization. The parameters included:

- Drag coefficient ($C_D$).
- Reflection coefficient ($C_R$).
- Satellite radius.
- Third-body gravitational forces (enabled/disabled).

The optimization minimized the Root Mean Square Error (RMSE) between Orekit's propagated trajectory and the ephemeris data over 1000 steps (16.6 hours). The RMSE was calculated as:

$$\text{RMSE} = \sqrt{\frac{1}{N} \sum_{t=1}^{N} \| r_{\text{ephemeris}}(t) - r_{\text{Orekit}}(t) \|^2}. \tag{42}$$

## G.1  Figures of Residuals and Orbits

Figures 49 to 52 present the propagated orbits compared to the ephemeris data. Oscillatory residuals were observed in the $x$, $y$, and $z$ components, in Figure 48, increasing over time due to unmodeled perturbative forces and numerical integration errors.

## G.2  Residuals Analysis

In this section we present further analysis to validate the precision of the OrbitZoo propagation against ephemeris data from Starlink. Residuals between real orbital data were computed by comparing overlapping predictions from ephemeris files. The residual at each time step was calculated as the Euclidean norm of the difference between the position vectors:

$$\text{Residual}(t) = \| r_{\text{ephemeris}}(t) - r_{\text{Orekit}}(t) \|. \tag{43}$$

The following scenarios were evaluated to understand the discrepancies between predictions:

- **Scenario 1: Full Propagation vs. Initial Ephemeris File.** Residuals were computed for the entire 1000-step propagation against the first ephemeris file.
- **Scenario 2: Second Half Propagation vs. Second Ephemeris File.** Residuals were computed for the second half of the propagation (500 steps) against the second ephemeris file.
- **Scenario 3: Overlapping Region Between Consecutive Files.** Residuals were computed for the overlapping period (approximately 8 hours) between two consecutive ephemeris files.

Figures 44 to 47 illustrate the residuals for these scenarios. The overlapping region provides the most reliable assessment of the Starlink propagator's accuracy, as it is based on the most recent ephemeris data. These figures demonstrate the high accuracy of Orekit's propagator when compared to Starlink's ephemeris predictions, particularly in the overlapping regions.

As mentioned earlier, the first 8 hours of these ephemeris files are the most relevant as they contain the most up-to-date information at the time of their release. To assess the accuracy of our predictions using Orekit, we must also evaluate the accuracy of Starlink's own predictions. If Starlink's error is significant, potentially due to unforeseen events, our propagation may also exhibit a large error as a result.

A propagation of 16.6 hours (1000 steps) was performed using Orekit, Figures 44 through 47 display the residuals of this propagation for four different satellites.

The first case shown in the figures corresponds to the residuals between the propagated data and the first ephemeris file. This case is labeled as " Orekit vs Ephemeris 1" because it uses the initial

ephemeris file for comparison. The propagation was initiated with the first position of this first ephemeris file.

The second case shows the residuals between the same propagated trajectory and the second ephemeris file, labeled as "ephemeris 2." This second file contains more up-to-date information compared to the first ephemeris file, as it was released 8 hours later.

Finally, the third case examines the residuals between the two consecutive ephemeris files (ephemeris 1 and ephemeris 2). These residuals represent Starlink's propagator's error, as the discrepancies arise solely from Starlink's own predictions over the overlapping time frame. If the residual from this third case is higher than the error in the first case, it means that Orekit's propagation outperformed Starlink's.

As a final remark, multiple factors beyond unknown satellite characteristics can contribute to higher RMSE values.

First, limited information on how Starlink produces its ephemeris data can lead to inconsistencies when attempting to approximate the satellites' trajectories. Researchers [44] suggest that Starlink may use multiple propagators and even switch among them mid-propagation, introducing further uncertainty into a single ephemeris file. Consequently, reconstructing each satellite's real orbital trajectory from Starlink's ephemeris data may vary across different satellites.

Second, our propagation relies exclusively on the first recorded position and velocity from each Starlink ephemeris file, under the assumption that this initial state has the least accumulated error. If that first state is inaccurate, our predicted trajectory will be skewed, causing higher RMSE values.

Lastly, Starlink's propagation benefits from detailed space weather information. As a result, some of our trajectory predictions may happen to align more closely with real conditions than others, contributing to variability in RMSE values. This same reasoning extends to the previously mentioned satellite characteristics, as well as other forces acting on the satellite.

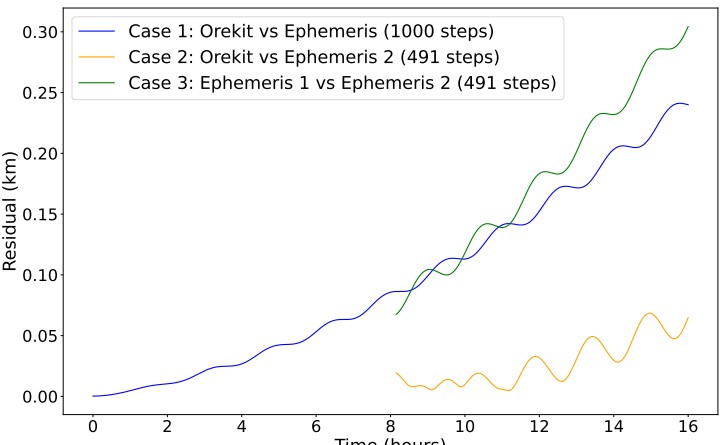

Figure 44: Visualization of the residuals in three different cases for NORAD ID 44744. The residuals represent the distance between the corresponding points in each comparison. Case 1 (in blue) shows the residuals between the ephemeris data from the first ephemeris file and the Orekit propagation over the entire 1000 steps. Case 2 (in yellow) shows the residuals between the second ephemeris file and the Orekit propagation over the later 491 steps. Case 3 (in green) shows the residuals between the two consecutive ephemeris files during their overlapping region within the 1000 steps. The first and second ephemeris files provide position data for different time spans, with the intersection capturing the transition between them.

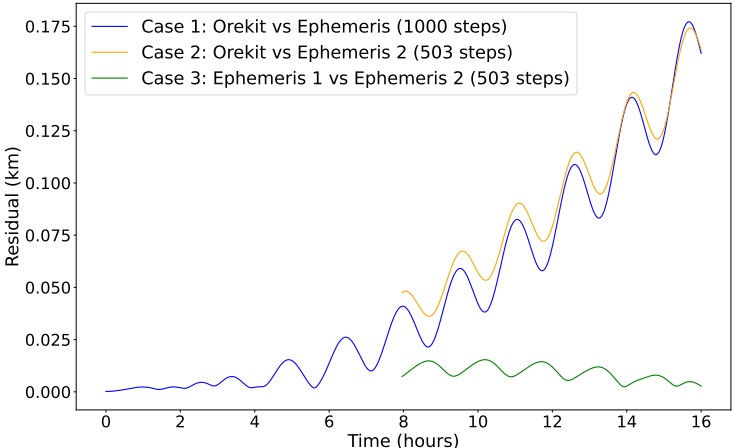

Figure 45: Visualization of the residuals in three different cases for NORAD ID 44753. The residuals represent the distance between the corresponding points in each comparison. Case 1 (in blue) shows the residuals between the ephemeris data from the first ephemeris file and the Orekit propagation over the entire 1000 steps. Case 2 (in yellow) shows the residuals between the second ephemeris file and the Orekit propagation over the later 503 steps. Case 3 (in green) shows the residuals between the two consecutive ephemeris files during their overlapping region within the 1000 steps. The first and second ephemeris files provide position data for different time spans, with the intersection capturing the transition between them.

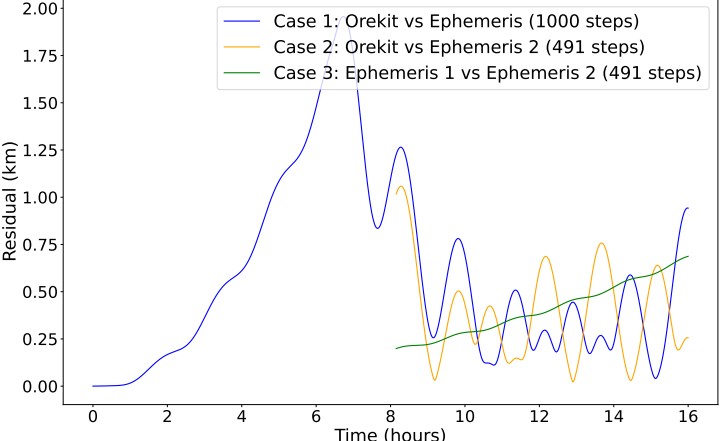

Figure 46: Visualization of the residuals in three different cases for NORAD ID 44923. The residuals represent the distance between the corresponding points in each comparison. Case 1 (in blue) shows the residuals between the ephemeris data from the first ephemeris file and the Orekit propagation over the entire 1000 steps. Case 2 (in yellow) shows the residuals between the second ephemeris file and the Orekit propagation over the later 491 steps. Case 3 (in green) shows the residuals between the two consecutive ephemeris files during their overlapping region within the 1000 steps. The first and second ephemeris files provide position data for different time spans, with the intersection capturing the transition between them.

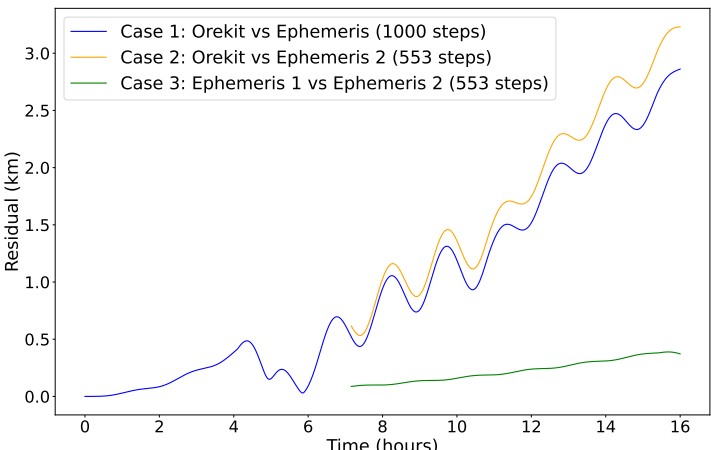

Figure 47: Visualization of the residuals in three different cases for NORAD ID 44921. The residuals represent the distance between the corresponding points in each comparison. Case 1 (in blue) shows the residuals between the ephemeris data from the first ephemeris file and the Orekit propagation over the entire 1000 steps. Case 2 (in yellow) shows the residuals between the second ephemeris file and the Orekit propagation over the later 553 steps. Case 3 (in green) shows the residuals between the two consecutive ephemeris files during their overlapping region within the 1000 steps. The first and second ephemeris files provide position data for different time spans, with the intersection capturing the transition between them.

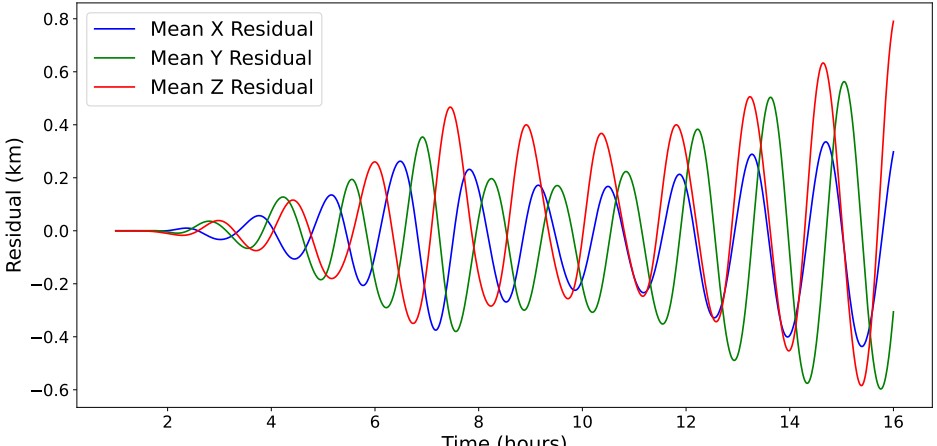

Figure 48: Visualization of the mean position residuals between the ephemeris data and Orekit propagation over 1000 steps for the satellites with NORAD ID 44744, 44748, 44753 and 44921.

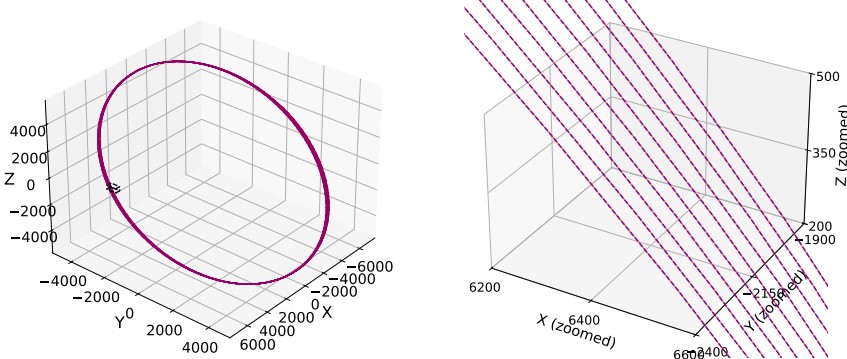

Figure 49: Comparison of the propagated orbits and the ephemeris data, for NORAD ID 44744, illustrating the overlap between the two. The cube in the left plot indicates the approximated region that is magnified in the right plot for a closer view. The Orekit propagation is represented by a solid red line, while the ephemeris data is shown as a dashed blue line.

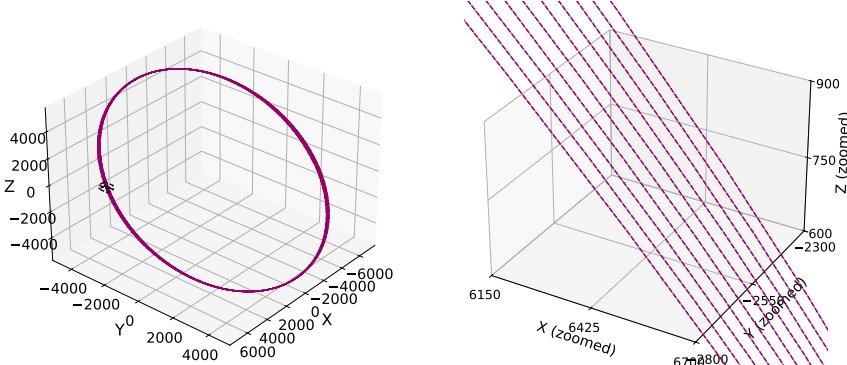

Figure 50: Comparison of the propagated orbits and the ephemeris data, for NORAD ID 44748, illustrating the overlap between the two. The cube in the left plot indicates the approximated region that is magnified in the right plot for a closer view. The Orekit propagation is represented by a solid red line, while the ephemeris data is shown as a dashed blue line.

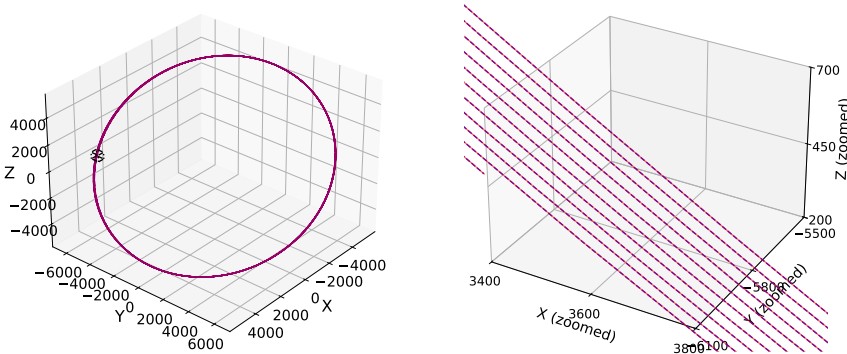

Figure 51: Comparison of the propagated orbits and the ephemeris data, for NORAD ID 44753, illustrating the overlap between the two. The cube in the left plot indicates the region approximated that is magnified in the right plot for a closer view. The Orekit propagation is represented by a solid red line, while the ephemeris data is shown as a dashed blue line.

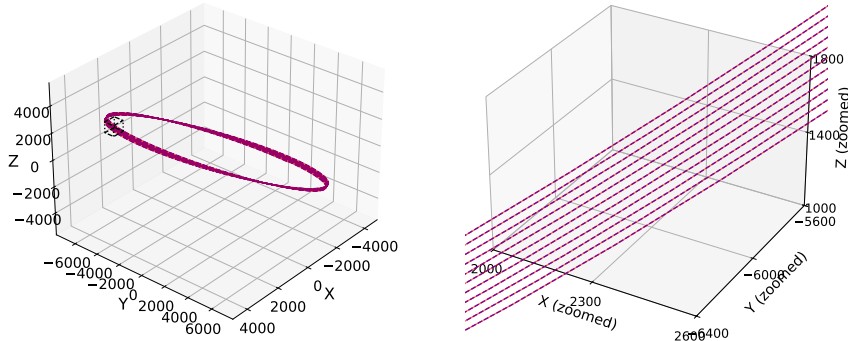

Figure 52: Comparison of the propagated orbits and the ephemeris data, for NORAD ID 44921 illustrating the overlap between the two. The cube in the left plot indicates the region approximated that is magnified in the right plot for a closer view. The Orekit propagation is represented by a solid red line, while the ephemeris data is shown as a dashed blue line.

### G.3 Applicability Beyond Starlink

Since no RL environments offering quantitative propagation analysis were found, the evaluation was extended beyond Starlink data. The Root Mean Squared Error, expressed in meters, was compared over 16 hours of propagation between OrbitZoo (via Orekit) and real satellite data from multiple sources, including OneWeb, as shown in Table 53. Additional datasets were obtained from the EUROLAS Data Center, encompassing organizations such as the Deutsches Geodätisches Forschungsinstitut (DGFI), the Joint Center for Earth Systems Technology (JCET), and the NERC Space Geodesy Facility (NSGF).

| Source | Satellite ID | RMSE (m) |
|--------|--------------|----------|
| OneWeb | 19 | 3945.9 |
| OneWeb | 1 | 2540.5 |
| OneWeb | 20 | 554.0 |
| OneWeb | 28 | 3360.3 |
| OneWeb | 14 | 2007.7 |
| OneWeb | 16 | 1581.9 |
| OneWeb | 18 | 5745.6 |
| OneWeb | 22 | 4797.2 |
| OneWeb | 23 | 3067.1 |
| OneWeb | 24 | 247.5 |
| OneWeb | 42 | 1050.8 |
| OneWeb | 39 | 1552.8 |
| OneWeb | 30 | 5653.4 |
| OneWeb | 21 | 4228.9 |
| OneWeb | 17 | 3648.1 |
| OneWeb | 15 | 1526.9 |
| OneWeb | 46 | 1095.0 |
| OneWeb | 48 | 4606.5 |
| OneWeb | 20 | 702.8 |
| OneWeb | 49 | 327.1 |
| OneWeb | 57 | 410.4 |
| DGFI | ajisai | 1801.2 |
| DGFI | lageos1 | 4509.7 |
| DGFI | lageos2 | 3531.3 |
| JCET | etalon1 | 2096.1 |
| JCET | etalon2 | 4624.4 |
| NSGF | lares | 1553.1 |
| NSGF | lares2 | 5006.4 |
| NSGF | larets | 1169.8 |
| NSGF | starlette | 1573.2 |
| NSGF | stella | 826.6 |

Figure 53: Orbital propagation errors (RMSE) between the Orekit propagation and reference satellites from different organizations.

For some of the tested satellites in Table 53, OrbitZoo maintains errors below 1 km after 16 hours of propagation. We note that satellites outside the OneWeb constellation experience corrective forces (e.g., empirical accelerations) not modeled by Orekit, which may explain some discrepancies.

## H Broader Impacts

This work advances RL research in high-dimensional, discrete and continuous control domains relevant to space operations, incorporating perturbative forces, model uncertainties, and ephemeris-based validation. Notably, **OrbitZoo** includes support for federated learning, enabling decentralized training across multiple agents or environments while preserving data locality, an important step toward scalable and privacy-aware coordination in space missions.

**Potential positive impacts** include increased autonomy and efficiency in satellite operations, reduced reliance on ground-based control, and more resilient responses to unexpected events like debris conjunctions. OrbitZoo can accelerate the development of safe, adaptive control strategies for real-world missions, and democratize access to space-focused RL research by offering an open, extensible platform.

A central challenge for deployment is the sim-to-real transfer – the reliable application of policies trained in simulation to real spacecraft. Closing this gap requires models that capture orbital perturbations, actuator limits, and sensor noise while remaining computationally efficient for on-board use. Real-time operation further demands low-latency inference and synchronization with existing flight software. Integrating RL-based control with established attitude and orbit control systems, telemetry pipelines, and ground operations is crucial for achieving certifiable autonomy.

**Potential negative impacts** include over-reliance on black-box models that may behave unpredictably in safety-critical situations. The deployment of learned policies without robust interpretability or certification frameworks could lead to unintended behaviors or mission failures. Additionally, advances in autonomous maneuvering might be misused in military or competitive commercial contexts without adequate regulatory oversight.

Our experiments demonstrate that policies trained in **OrbitZoo** can closely replicate real-world satellite behavior, including Starlink trajectories, positioning the platform as a benchmark for trustworthy and autonomous space systems.

