# OpenReview forum: "OrbitZoo: Real Orbital Systems Challenges for Reinforcement Learning"
_NeurIPS.cc/2025/Conference — NeurIPS 2025 poster_

### Official Review · Reviewer_8aNh · 2025-06-18

**Clarity:** 1
**Significance:** 2
**Originality:** 1
**Rating:** 4
**Confidence:** 3

**Summary:**

This paper presents OrbitZoo, a multi-agent RL environment to control
satellites in earth's orbit. OrbitZoo adheres to the PettingZoo API for the
multi-agent RL and uses Orekit for the flight dynamics.

The paper discusses general challenges of multi-agent RL for orbital dynamics,
presents OrbitZoo as a framework in this domain, and gives a few experiments
(or rather demonstrations).

It is unclear what the research question was that is to be tackled by this
paper. However, it mentions three contributions: (1) generation of
high-fidelity orbital data (which is not done by OrbitZoo itself), (2) that it
follows PettingZoo for a standardized API and (3) that it can be configured in
a flexible way (e.g., arbitrary number of bodies, etc.) and features a 3D
visualization.

In a sense, this paper is a technical report on the design the OrbitZoo
framework rather than a research paper.

**Questions:**

In §5.1 you say you have trained "the agent". What agent? How trained? How does
your framework lead to improvements compared to what baseline or state of the
art?

In l.213 you construct a covariance matrix Σ. It is probably the diagonal
matrix formed by the sigmas mentioned before, but do not let the reader guess.
So what is it? You explain that this is used to sample position and velocity
upon resetting the environment, but when is this again becoming relevant later
in the paper? It seems like a random details.

**Ethical Concerns:**

["NO or VERY MINOR ethics concerns only"]

**Final Justification:**

My remaining critique has very much been addressed in extensive discussions, including other reviewers. The remaining reasons why I am not going to accept is that still the application domain is a certain niche.

**Limitations:**

yes

**Paper Formatting Concerns:**

l.151: "in A" → "in appendix A"?

**Quality:**

2

**Strengths And Weaknesses:**

## Significance

It remains unclear in what sense this constitutes a novel, scientific
contribution to the field of RL, rather than an engineering artifact (which
might be very useful). The treatments in the paper are rather located in the
field of software engineering (e.g., architectural design, JSON configuration,
UML-like diagrams, description of modules) and orbital mechanics simulation
(e.g., representations of coordinates).

This also explains why the works in Table 1, to which this paper compares, are
or centered around venues in the aforementioned venues and corresponding Master
and PhD theses.

However, the actual multi-agent RL problem setting of controlling a multitude
of satellites is interesting.

## Clarity

It is unclear how OrbitZoo per se solves the key challenges mentioned in §3, as
discussed in §4. Saying that OrbitZoo uses Orekit is not a scientific
contribution, but a report.

Consequently, it is also unclear how the experiments in §5 are experiments
addressing a research hypothesis or a research question posed by the paper.
When, for instance, the Hohmann Maneuver experiments runs satisfactory, how or
in what sense is this attributed to the proposed framework OrbitZoo rather than
the components it orchestrates? How does it compare to alternative methods? Why
would your solution improve upon what by what mechanism? To sum up, the
experiments are rather demonstrations than experiments: They are not explaining
or revealing insight to explicit questions concerning the prior discussion of
your framework.

## Quality

This also becomes apparent in §5 (discussion). Demonstrating that RL policies
were trained in OrbitZoo is sold as a result of OrbitZoo rather than the
underlying RL algorithms. The use of continuous action spaces is mentioned as
an insight to improve performance, but this is not really part of the treatment
in this paper, which is about a framework. The final conclusion, that OrbitZoo
would pave the way to trustworthy AI-driven autonomy in future space missions
is, well, blunt overselling.

## Originality

It is unclear in what terms an originality of this work is given.

## More comments

In §3.3 the exploration not only concerns the action space, but the joint
state-action space.

In §3.4 you should explain what "constellation" and "propagation" means, i.e.,
add context.

In §3.3, your argument that deterministic policies and Gaussian noise for
exploration may be suboptimal sounds hand wavy.

In §3.5 you phrase "realistic simulation environments, such as OrbitZoo" is
imprecise: It should be "as _in_ OrbitZoo", as this is actually done by Orekit,
right? In fact, say that this is provided _by_ OrbitZoo is an argument on thin
ice.

In §4 you phrase "to address the challenges". Name the challenges, e.g., C1 to
C4, and explicitly explain _how your framework_ addresses each C1 to C4. Also,
what is "vanilla OrbitZoo"; it indicates that there is a vanilla version and an
extended/patched version that could do more? This is not the case, but you
attempt to sell the merits of the third-party components as merits of the
OrbitZoo framework.

In §4 you speak of a "Python environment". Note that "environment" is now very
ambiguous as it could also be a (Python virtual) environment or a (RL)
environment (in the sense of Barto Sutton) or a (simulation) environment (as
discussed earlier). Be more precise.

---

> ### Author Rebuttal · Authors · 2025-07-30
>
> We sincerely thank you for your thoughtful review and for **highlighting the potential of multi-agent reinforcement learning in satellite control**. We especially appreciate your recognition that OrbitZoo could be **a valuable engineering contribution**, which is central to our goal.
>
> While we acknowledge the Datasets and Benchmarks (D&B) track, **our primary aim is to support exploration of fundamental RL challenges in orbital dynamics**, such as partial observability, coordination, communication, sparse rewards, fidelity-efficiency tradeoffs, and sim-to-real transfer. We believe this practical utility lays the groundwork for future scientific advances.
>
> We agree that some sections of the paper could be clearer, and we have made changes to address several of your concerns. First, we outline those changes. Then, we respond to your specific questions, followed by a discussion of the identified weaknesses and remaining comments.
> ___
> ## Changes
> ### Change 1
> L151 (Section 3.2) - We fix the typo at the end of the sentence: “(…) are explored in appendix A.”.
> ### Change 2
> L173 (Section 3.3) - We now address the challenge of exploration in Section 3.2, highlighting the importance of a modular framework that supports various RL algorithms—not limited to continuous action spaces. We also specifically discuss joint state and action spaces in cooperative and adversarial multi-agent RL settings.
> ### Change 3
> L179 (Section 3.4) - We explicitly define a constellation as a group of satellites cooperating to achieve a partial or complete collective objective, and refer readers to the propagation details in Appendix A.3 for further understanding.
> ### Change 4
> L188 (Section 3.5) - We change the initial sentence to “Reinforcement learning environments with external components that provide high-fidelity dynamics, such as OrbitZoo, (…)”.
> ### Change 5
> L194 (Section 4) - We revised the first sentence to: “OrbitZoo is a flexible Python RL environment designed to address (…)”. Challenges are referred to Section 3.2.
> ### Change 6
> L213 (Section 4.1) - We rewrite the sentence as “(…) which are internally used to construct a diagonal covariance matrix $\Sigma$ with these uncertainties as its diagonal entries.”.
> ### Change 7
> L246 (Section 4.3) - We updated paragraphs to follow the new structure, moving “Realistic Thrust Modeling” under “Bridging the Reality Gap” and addressing scalability and computational trade-offs in “Multi-Agent Coordination.” We also highlight OrbitZoo’s capability to simulate FRL.
> ### Change 8
> L277 (Section 5.1) - In the “Setup” paragraph, we briefly characterize what is the agent (a satellite with thrusting capabilities that interacts with the environment) and its strategy to apply thrusts (RL algorithm).
> ___
> ## Questions
> > Q1. In 5.1 you say you have trained "the agent". What agent? How trained? How does your framework lead to improvements compared to what baseline or state of the art?
>
> The agent refers to **a thrust-capable satellite trained with PPO to reach a target orbit**. Due to limited context, **we made [Change 8](#change-8)**. As described in Appendix D.3, the agent learns to reach the target within a small tolerance. **The framework does not guarantee improvement over a baseline**, as results depend on factors like the **RL algorithm**, **dynamics realism**, **implementation**, and available **validation tools**. In the Hohmann maneuver, classical control offers a fuel-optimal solution under simplified, ideal assumptions. However, **RL agents adapt to perturbations and continuous thrust**, reaching the target under more realistic settings and **outperforming classical methods when ideal assumptions no longer hold**. We compare errors between the optimal maneuver (simplified dynamics) and five successful RL runs (Experiments 1 and 2, Table 9), across varying realism levels. The table reports the mean and standard deviation of fuel error (in kg):
> Exp 1 No Perturbations|Exp 1 Perturbed|Exp 2 No Perturbations|Exp 2 Perturbed
> -|-|-|-
> $0.34_{(0.02)}$|$0.36_{(0.01)}$|$0.85_{(0.31)}$|$0.86_{(0.16)}$
>
> Additionally, we compare the altitude error (in km) of those attempts by step:
> Step|Exp 1 No Perturbations|Exp 1 Perturbed|Exp 2 No Perturbations|Exp 2 Perturbed
> -|-|-|-|-
> **0**|$0.00_{(0.00)}$|$0.00_{(0.00)}$ | $0.00_{(0.00)}$|$0.00_{(0.00)}$
> 1|$0.01_{(0.00)}$|$0.01_{(0.00)}$ $0.01_{(0.01)}$|$0.01_{(0.00)}$
> 2 |$0.03_{(0.00)}$|$0.03_{(0.00)}$|$0.02_{(0.02)}$|$0.02_{(0.01)}$
> 3 |$0.07_{(0.00)}$|$0.06_{(0.01)}$|$0.04_{(0.04)}$|$0.04_{(0.01)}$
> 4 |$0.11_{(0.01)}$|$0.10_{(0.01)}$|$0.06_{(0.06)}$|$0.06_{(0.01)}$
> ...|...|...|...|...
> 553|$2.57_{(1.51)}$|$10.13_{(1.66)}$|$2.43_{(2.07)}$|$8.10_{(3.60)}$
> 554|$2.61_{(1.51)}$|$10.13_{(1.66)}$|$2.35_{(2.07)}$|$8.07_{(3.58)}$
> 555|$2.65_{(1.51)}$|$10.12_{(1.67)}$|$2.28_{(2.07)}$|$8.04_{(3.55)}$
> ...|...|...|...|...
> **765**|$10.73_{(1.30)}$|$5.10_{(1.12)}$|$17.49_{(2.97)}$|$5.66_{(4.63)}$
> 766|$10.78_{(1.29)}$|$5.05_{(1.12)}$|$17.59_{(2.97)}$|$5.73_{(4.63)}$
> 767|$10.83_{(1.29)}$|$5.00_{(1.11)}$|$17.69_{(2.98)}$|$5.80_{(4.63)}$
> ...|...|...|...|...
> 998|$16.30_{(0.96)}$|$3.74_{(0.75)}$|$30.74_{(3.53)}$|$13.33_{(6.08)}$
> 999|$16.30_{(0.96)}$|$3.76_{(0.75)}$|$30.74_{(3.53)}$|$13.32_{(6.09)}$
>
> Both experiments reach the target orbit using less than 1 kg of additional fuel on average. This last table highlights thrust steps (see Figure 19), with bold marking optimal thrusts. Experiment 1 yields lower final altitude error, while Experiment 2 shows greater trajectory variance—suggesting a deeper understanding of the dynamics. These findings also indicate that shaping the reward to emphasize the semi-major axis could improve performance. **This analysis has been added to the revised version**.
> > Q2. In l.213 you construct a covariance matrix Σ. It is probably the diagonal matrix formed by the sigmas mentioned before, but do not let the reader guess. So what is it? You explain that this is used to sample position and velocity upon resetting the environment, but when is this again becoming relevant later in the paper?
>
> Precisely. The covariance matrix $\Sigma$ uses uncertainties as diagonal entries. We thank the reviewer for pointing this out, which led to [Change 5](#change-5). Sampling from this multivariate normal defines the body’s initial state, hence the importance of mentioning it. In the collision avoidance mission (Appendix D.5), this sets the miss distance at time of closest approach (TCA), necessary for the agent to experience slightly different scenarios during training.
> ___
> ## Strengths and Weaknesses
> ### Significance
> As mentioned earlier, OrbitZoo was designed to **advance RL research in orbital dynamics** by offering a **standardized and accessible framework**.
> ### Clarity and Quality
> As addressed in your first question, **the results depend on external components beyond OrbitZoo itself**, which may have impacted clarity in some sections.
> ### Originality
> While somewhat subjective, **OrbitZoo offers a unique interactive 3D interface** for visualizing multi-body dynamics and mission strategies — **a distinctive validation feature absent in other environments**. Additionally, **Orekit developers have expressed interest in the OrbitZoo framework**.
> ### Additional Comments
> > C1. In 3.3 the exploration not only concerns the action space, but the joint state-action space.
>
> Now we address joint state and action spaces in **Section 3.2**, as [Change 2](#change-2) states.
> > C2. In 3.4 you should explain what "constellation" and "propagation" means, i.e., add context.
>
> **We clarify the terms *constellation* and *propagation*** in that context, reflecting [Change 3](#change-3).
> > C3. In 3.3, your argument that deterministic policies and Gaussian noise for exploration may be suboptimal sounds hand wavy.
>
> As noted in [Change 2](#change-2), **we now highlight the importance of a modular framework** **supporting various RL algorithms**, not just those with continuous actions.
> > C4. In 3.5 you phrase "realistic simulation environments, such as OrbitZoo" is imprecise: It should be "as in OrbitZoo", as this is actually done by Orekit, right?
>
> Yes, **propagation is handled by Orekit, not OrbitZoo**, as noted throughout the paper (e.g., Figure 1, L11, L53, L60, L121, L248). Orekit propagates bodies but **does not provide a standardized RL environment**. It lacks methods for applying thrusts based on immediate policies, though it supports pre-defined thrusts via event detectors. OrbitZoo builds on this by dynamically adding/removing thrust during propagation. **[Change 4](#change-4) reinforces this distinction**.
> > C5. In 4 you phrase "to address the challenges". Name the challenges, e.g., C1 to C4, and explicitly explain how your framework addresses each C1 to C4. Also, what is "vanilla OrbitZoo"; it indicates that there is a vanilla version and an extended/patched version that could do more?
>
> **[Change 5](#change-5) and [Change 7](#change-7) address the challenges you've raised.**
> *Vanilla* OrbitZoo refers to the environment in Figure 1, **without a built-in RL implementation**, allowing flexibility for different mission types. It supports multi-body setups, thrust, perturbations, and various propagation settings, enabling control strategy design, data generation, or dynamics analysis. OrbitZoo is easily extensible for RL or MARL missions by integrating custom algorithms, rewards, training routines, and communication protocols, as shown.
> **We do not attempt, *in any way*, to sell the merits of third-party components**, as clarified earlier.
> > C6. In 4 you speak of a "Python environment". Note that "environment" is now very ambiguous as it could also be a (Python virtual) environment or a (RL) environment (in the sense of Barto Sutton) or a (simulation) environment (as discussed earlier).
>
> We thank you for your suggestion. **We have rephrased that sentence to be more precise**, as seen in [Change 5](#change-5).
> ___
> ## Paper Formatting Concerns
> [Change 1](#change-1) aims to correct the typo.

---

> > ### Comment · Reviewer_8aNh · 2025-08-01
> >
> > Thank you for your work in addressing and fixing various points raised in the
> > review.
> >
> > We can largely group the elements of this submission in two groups:
> >
> >   1. The introduction of the OrbitZoo environment in the domain of multi-agent
> >      satellite control. This is largely about software engineering and orbital
> >      mechanics.
> >
> >   2. Content that is related to RL per se, but in terms of original
> >      contribution actually disconnected to the contribution of this paper,
> >      which is about OrbitZoo, but rather provided by already existing
> >      engineering components and scientific contributions.
> >
> > What would make this paper fit for the RL track of NeurIPS would be part (2),
> > but there is no scientific contribution to RL in its essence, but it is solely
> > the motivation and application domain of this framework.
> >
> > An interesting multi-agent RL environment is surely beneficial for RL research.
> > But what would make a researcher for multi-agent RL switch or not to switch to
> > your framework, which cannot be provided by other multi-agent RL application
> > domains? The lack of motivation and comparison in this direction also shows
> > that this apper is not so much about RL but about orbital dynamics per se.
> >
> > While I see a merit in the work of the authors, I agree with the authors when
> > they say that the Datasets and Benchmarks (D&B) track would have been more
> > fitting. In particular, the call for papers of this track would have implied
> > more stringent changes in the submission guidelines for this type of
> > submission, e.g., required benchmark code submission.
> >
> > To sum up, I feel confirmed that this paper should be submitted to a more
> > fitting venue.

---

> ### Author Response · Authors · 2025-08-01
>
> Thank you for the follow-up and your thoughtful distinctions!
>
>
>
> **Clarifying Track Fit**
>
>
>
> We respectfully clarify a misunderstanding: we did **not** state that the Datasets and Benchmarks (D&B) track would be more appropriate. We deliberately submitted to the main track because OrbitZoo is not a fixed benchmark suite but a modular environment that enables the exploration of core RL challenges such as multi-agent coordination, partial observability, reward sparsity, and sim-to-real transfer—all of which are central to modern RL research. **We emphasize that works like "PettingZoo: A Standard API for Multi-Agent Reinforcement Learning" were published in the NeurIPS main track in 2021, the same year the Datasets and Benchmarks (D&B) track was introduced. This reinforces that RL environment contributions of broad relevance to ML research are recognized as suitable for the main track.**
>
>
>
> **Relevance to RL Community**
>
>
>
> While OrbitZoo builds on realistic orbital mechanics, its novelty lies in **supporting new RL research directions**. Like OpenAI Gym, DMLab, or D4RL, its contribution is in enabling **new problem formulations**, not proposing new algorithms. OrbitZoo’s fidelity, configurability, and PettingZoo compliance make it a unique, high-impact addition to the MARL ecosystem.
>
>
>
> **Why Use OrbitZoo?**
>
>
> Researchers seeking realistic, high-stakes coordination settings currently lack suitable environments. OrbitZoo offers long-horizon dynamics, grounded validation, and scalable multi-agent interaction—features underrepresented in traditional MARL settings. In addition to realistic dynamics, OrbitZoo introduces **novel domain-specific, physically grounded reward functions for tasks such as target chasing, orbit transfers, and collision avoidance**. These reward formulations are designed to reflect real mission objectives and constraints, going beyond simplistic distance- or success-based shaping often found in other contributions. This makes OrbitZoo and this paper a relevant contribution for developing and evaluating RL methods in complex, safety-critical domains, like space.
>
>
> **Reflecting on Revisions**
>
> We sincerely appreciate the reviewer’s detailed critique, which prompted us to substantially revise both the paper and our framing. In response, we made **numerous textual and structural improvements** (summarized above), clarified terminology, r**efined the articulation of challenges and contributions**, and provided **new experimental analysis** (e.g., policy behavior under perturbations). We hope these clarifications demonstrate our commitment to making OrbitZoo a scientifically useful and rigorously presented contribution for the RL community.
>
>
>
>
> We hope this clarifies our contribution and reinforces the relevance of OrbitZoo to the NeurIPS main track. We're happy to further discuss any remaining concerns.

---

> > ### Comment · Reviewer_8aNh · 2025-08-05
> > **Summary**
> >
> > **C1. The paper presents itself mainly as a technical report.**
> >
> > Has also been phrased by reviewer paFx. The paper discussed the visualization,
> > json files, architectural aspects, and so on.
> >
> > I agree with reviewer paFx that the paper addresses a very specialized problem
> > setting (orbital dynamics), which makes it difficult to raise general interest
> > in the RL community at NeurIPS. The works in Table 1 (Comparison of RL
> > environments for orbital dynamics) in the paper illustrate this by the venues
> > where they have been published.
> >
> >
> > **C2. The contribution of the paper to NeurIPS is unclear.**
> >
> > The abstract mainly discusses orbital dynamics as such and that existing RL
> > environments are custom-built, so they would introduce OrbitZoo built on
> > "high-fidelity industry standard libraries". The framework is then evaluated
> > against a Starlink dataset. This is essentially the abstract's position.
> >
> > The conclusion section then shifts focus to the demonstration that a RL policy
> > could be trained within OrbitZoo and it would reach good performance, so the
> > authors claim, this makes OrbitZoo an interesting RL benchmark. And OrbitZoo
> > would pave the way to trustworthy AI-driven autonomy in future space missions.
> >
> > What is now the contribution for the NeurIPS community?
> >
> > As the authors mentioned in a response to a review 4WXn and myself, it is not
> > their "intent to achieve state-of-the-art results", but "showcase the large
> > range of missions that OrbitZoo supports". It should be an "RL research enabler".
> >
> > Well, then, to judge whether OrbitZoo is an interesting RL environment, where
> > RL research could be done, we would like to see how the current
> > state-of-the-art RL methods would perform and where the RL research gaps
> > exactly lie in. That is, we would like the paper to identify research gaps, so
> > foster and enable RL research on this RL environment. But this is not the scope
> > of this paper, as the authors responded.
> >
> >
> > **C3. Cannot check framework and data generation as contribution.**
> >
> > When we look at the contributions mentioned in the introduction then three
> > categories are mentioned: (i) data generation), (ii) RL, and (iii) customizable
> > framework with visualization.
> >
> > The contribution (ii) RL is questionable, because this paper does not provide
> > any RL research per se, but only would like to make RL research in orbital
> > dynamics easier, see above.
> >
> > Then we are at (i) data generation and (iii) the framework. This would rather
> > fit to the datasets and benchmarks track. There is a good reason for why this
> > track exists: It has different submission criteria.
> >
> > In particular, it requires to send in code. This totally makes sense for this
> > submission, since the main contribution is the framework. To judge this
> > framework, and whether it would be able to fulfill the main promise of this
> > submission – to foster MARL research –, we would need to see the code. However,
> > in a response to reviewer 4WXn, it became clear that a public repository is not
> > readily available but they "intend to provide" it "in the near future". This
> > seems implausible, given that this is the very work of this submission.
> >
> >
> > **C4. Distinguish between framework, simulation and RL.**
> >
> > OrbitZoo is a framework, and it orchestrates components. The components are not
> > a contribution of the authors. For instance, one core component is Orekit,
> > which does the physics simulation.
> >
> > For every experiment, validation and demonstration, we need to exactly ask,
> > what the underlying question is that is addressed, and where in the overall
> > system this question is actually raised. For instance, an experiment that
> > validates the simulation fidelity of Orekit then this is not a statement about
> > OrbitZoo. Or if we have an experiment about the performance achieved by
> > a single RL algorithm then this is a statement about the RL algorithm, not
> > OrbitZoo. I miss clarity on this level in the experimental section.
> >
> >
> > **Summary.** I do not challenge the statement of the authors that OrbitZoo is
> > useful. It might be very useful, for a very specific application domain to
> > a very specific group of people. I highly doubt that this submission is fit for
> > a NeurIPS paper or could attract relevant interest of MARL researchers at the
> > conference, because the paper does not address their needs or requirements.
> >
> > The quality of the original submission was limited, but the authors reported to
> > have fixed many details in the discussion and the precision of language. At the
> > same time, we appear to disagree on some fundamental positions. So I am not
> > sure whether, as one example, the critique on naming the challenges C1-C4 in the
> > paper is addressed by the framework, has been finally really resolved.
> >
> > I appreciate a lot that the authors engaged with the reviewers. I think there
> > is a merit in this work, but it would gain a lot by a clear positioning and
> > a submission to a more fitting venue. And maybe you can publish the code first
> > and a technical paper that systematically discusses on how to foster MARL on
> > this framework later on.

---

> ### Author Response · Authors · 2025-08-05
>
> **Thank you for your thorough and thoughtful follow-up.** We appreciate your open reflections and your acknowledgment of the improvements made to clarity, language, and detail during the rebuttal phase. We are also grateful for your recognition that OrbitZoo may hold value for the community and your emphasis on the importance of clear positioning.
>
> ---
>
> ### C1 & C2: Framing and Contribution to NeurIPS, in the epicenter of autonomy research
>
> We respectfully affirm that OrbitZoo contributes to NeurIPS by enabling research on foundational RL challenges that are increasingly relevant in the emerging era of **autonomous systems in the physical world**. As RL progresses toward real-world deployment, there is a pressing need to move **beyond synthetic toy environments and engage with scenarios involving multi-agent coordination under physical constraints, partial observability, limited actuation (e.g., fuel), safety-critical decision making, and sim-to-real transfer.**
>
> OrbitZoo provides a modular, high-fidelity environment to support this transition. While grounded in orbital dynamics, its abstractions and challenge structures (e.g., limited sensing, coordination over communication constraints, adversarial agents) are representative of **broader autonomy domains such as air traffic management, underwater exploration, and disaster response.** This aligns with NeurIPS precedent, such as the 2021 main-track paper on MARL for voltage control in power networks, which—like OrbitZoo ---proposed a domain-specific Dec-POMDP environment without introducing new algorithms.
>
> Rather than proposing a new algorithm, OrbitZoo enables the RL community to rigorously study these challenges in a **reproducible and physically grounded setting**. This research-enabling role is consistent with the publication of prior NeurIPS environments such as PettingZoo, which advanced the field by opening new problem classes, not by benchmarking or algorithmic novelty alone.
>
> ### C3: Code Availability Clarification
>
> We would like to respectfully correct a repeated misunderstanding. Contrary to the reviewer's claim, the **complete OrbitZoo codebase is already available with the submission.** It is included as:
>
> - **A supplementary ZIP archive containing all code for the experiments and environment** (this page below the abstract and checklist confirmation), and
>
> - An **anonymized GitHub repository link**, clearly stated in the main paper (line 836).
>
> All experiments, figures, and demonstrations are **fully reproducible using this material**. We understand the fast pace of reviewing, but given the centrality of this point, we kindly ask reviewer 8aNh, the AC and other reviewers to consider that this concern may be based on a misunderstanding of what was already included at submission time.
>
> ### C4: Attribution and Experimental Clarity
>
> We agree it is crucial to distinguish between the contributions of OrbitZoo and those of its underlying components (e.g., Orekit). We’ve taken care in the paper and figures (e.g., Figure 1, §3.5, §4.3) to credit the use of Orekit for propagation while clearly articulating how OrbitZoo integrates real-time thrust control, RL interfaces, and task definitions atop it. We also updated the experimental section to focus on **what OrbitZoo enables** (e.g., learning under perturbations, continuous control under uncertainty), making it absolutely clear  what is attributable to algorithmic results and what is attributable to the framework.
>
>
> ---
> **We sincerely appreciate your engagement throughout this process.** While we may differ on track fit, we believe OrbitZoo’s capacity to expose underexplored RL challenges --- and its alignment with prior main-track infrastructure papers --- makes it a valuable contribution to the NeurIPS community. We are grateful for the feedback, which has already strengthened the work and will guide its further evolution.

---

> > ### Comment · Reviewer_8aNh · 2025-08-05
> >
> > I am happy to confirm that C3 is resolved.
> >
> > On the other points I still disagree. Anyhow, the authors changed my mind in the sense that a strong reject as assessment is not justified anymore from my point of view and I will improve my score.
> >
> > Thank you for your enduring engagement.

---

> > > ### Author Response · Authors · 2025-08-05
> > > **Acknowledgment**
> > >
> > > We sincerely thank you for your reconsideration and for updating your assessment!
> > >
> > > We deeply appreciate your thoughtful engagement throughout the process and the opportunity to improve our work through this exchange.

---

### Official Review · Reviewer_4WXn · 2025-06-30

**Clarity:** 3
**Significance:** 3
**Originality:** 3
**Rating:** 5
**Confidence:** 2

**Summary:**

This paper proposes a new multi-agent RL environment for orbital dynamics called OrbitalZoo. OrbitalZoo can support data generation/simulation, RL algorithm research and visualization, etc. It addresses some key challenges in this area, including Bridging the Reality Gap, Realistic Thrust Modeling, Multi-Agent Coordination, and Interactive Visualization for Debugging and Analysis. Experimental results including single agent, multi-agent as well as real world data validation show OrbitZoo as a reliable benchmark for RL-based autonomy in space operations.

**Questions:**

1. What's the plan for open source?
2. What is the baseline algorithm? Compared w/ RL algorithms, is RL much better than baseline algorithm?
3. Is the MARL support general to all RL algorithms? Or does it only work for PPO?

**Ethical Concerns:**

["NO or VERY MINOR ethics concerns only"]

**Final Justification:**

Very solid work and authors have addressed all my concerns.

**Limitations:**

yes

**Quality:**

3

**Strengths And Weaknesses:**

Strengths:
1. The paper is well written.
2. It seems OrbitZoo supports a variety of RL algorithms and experiments provide comparison among these algorithms for the same task.
3. Various of interesting tasks related to orbital dynamics are included in this benchmark, such as chase target, collision avoidance, Hohmann Maneuver, etc.

Weakness:
1. regarding "Bridging the Reality Gap." it seems it is only supported by the Starlink example, is there any other evidence to show OrbitZoo can bridge the reality gap better than other environments?
2. Is there any example of OrbitZoo's visualization capabilities?

---

> ### Author Rebuttal · Authors · 2025-07-30
>
> We thank you for your positive feedback. We are glad to hear that the **clarity of the writing**, the **variety of reinforcement learning algorithms supported**, **multi-agent capabilities**, and the **diversity of orbital tasks included in OrbitZoo** were well received. We appreciate your recognition of the **benchmark's breadth and relevance** to real-world orbital dynamics problems.
>
> We will first address your questions and related weaknesses, followed by a discussion of the remaining limitations and areas for improvement as thoroughly as possible.
> ___
> ## Questions
>
> > Q1. What's the plan for open source?
>
> **We already provide full access** to the OrbitZoo environment, trained models, and code needed to reproduce our experiments in both the **supplementary material** and on the **anonymized repository** (appendix D, in the end of the introduction). In the near future, we do **intend to provide a public repository** so that people can independently contribute in their areas of expertise (e.g., use cases, RL algorithms, propagation and integration, scalability, more methods for body interactions, interface components, etc.), building upon what was already developed.
>
> > Q2. What is the baseline algorithm? Compared w/ RL algorithms, is RL much better than baseline algorithm?
>
> Answering your first question, **we disagree with the assumption that there is a defined baseline throughout the paper**, since **there are comparisons that evaluate OrbitZoo with other environments** (Section D.2) not only in terms of **RL algorithms** (Kolosa’s environment), but also regarding **data fidelity** (Herrera’s environment), and more nuanced differences, such as the **actual code** needed to develop such missions and **easily evaluate** them. We highlight that **it is not our intent here to achieve state-of-the-art results** within each of these missions, but simply **showcase the large range of missions that OrbitZoo supports**, which can indeed be used to easily develop new and more robust algorithms, fundamentally working as a **RL research enabler**. Nonetheless (and concerning your second question), the Hohmann maneuver contains an optimal classical control approach (as shown in D.3.1) that can benefit from a more detailed comparison to the strategy learned by PPO. We have performed an additional analysis on the error between the optimal control approach and five succesfull attempts when using the agent trained in Experiment 1 and Experiment 2 (refer to Table 9) with different levels of perturbations. In the table below, we measure the average fuel error (in kg) and standard deviation of these attempts to the optimal approach under simplified dynamics:
>
> | Exp 1 No Perturbations | Exp 1 Perturbed | **Exp 2** No Perturbations | **Exp 2** Perturbed |
> | --- | --- | --- | --- |
> | $0.34_{(0.02)}$ | $0.36_{(0.01)}$ | $0.85_{(0.31)}$ | $0.86_{(0.16)}$ |
>
> Additionally, we compare the altitude error (in km) of those attempts by step:
>
> | **Step** | **Exp 1** No Perturbations | **Exp 1** Perturbed | **Exp 2** No Perturbations | **Exp 2** Perturbed |
> | --- | --- | --- | --- | --- |
> | **0** | $0.00_{(0.00)}$ | $0.00_{(0.00)}$ | $0.00_{(0.00)}$ | $0.00_{(0.00)}$ |
> | 1 | $0.01_{(0.00)}$ | $0.01_{(0.00)}$ | $0.01_{(0.01)}$ | $0.01_{(0.00)}$ |
> | 2 | $0.03_{(0.00)}$ | $0.03_{(0.00)}$ | $0.02_{(0.02)}$ | $0.02_{(0.01)}$ |
> | 3 | $0.07_{(0.00)}$ | $0.06_{(0.01)}$ | $0.04_{(0.04)}$ | $0.04_{(0.01)}$ |
> | 4 | $0.11_{(0.01)}$ | $0.10_{(0.01)}$ | $0.06_{(0.06)}$ | $0.06_{(0.01)}$ |
> | ... | ... | ... | ... | ... |
> | 553 | $2.57_{(1.51)}$ | $10.13_{(1.66)}$ | $2.43_{(2.07)}$ | $8.10_{(3.60)}$ |
> | 554 | $2.61_{(1.51)}$ | $10.13_{(1.66)}$ | $2.35_{(2.07)}$ | $8.07_{(3.58)}$ |
> | 555 | $2.65_{(1.51)}$ | $10.12_{(1.67)}$ | $2.28_{(2.07)}$ | $8.04_{(3.55)}$ |
> | ... | ... | ... | ... | ... |
> | **765** | $10.73_{(1.30)}$ | $5.10_{(1.12)}$ | $17.49_{(2.97)}$ | $5.66_{(4.63)}$ |
> | 766 | $10.78_{(1.29)}$ | $5.05_{(1.12)}$ | $17.59_{(2.97)}$ | $5.73_{(4.63)}$ |
> | 767 | $10.83_{(1.29)}$ | $5.00_{(1.11)}$ | $17.69_{(2.98)}$ | $5.80_{(4.63)}$ |
> | ... | ... | ... | ... | ... |
> | 998 | $16.30_{(0.96)}$ | $3.74_{(0.75)}$ | $30.74_{(3.53)}$ | $13.33_{(6.08)}$ |
> | 999 | $16.30_{(0.96)}$ | $3.76_{(0.75)}$ | $30.74_{(3.53)}$ | $13.32_{(6.09)}$ |
>
> Results from the first table suggest that **both experiments are able to achieve near optimal control while using, on average, no more than 1 kg of additional fuel**, which is a small portion of all the available fuel. The steps shown in second table were selected based on the intervals where agents perform thrusts (as seen in Figure 19). These results suggest that after the first optimal thrust (step 0), Experiment 2 generally better approximates the optimal altitude compared to Experiment 1, but with more variance. After the second optimal thrust (step 765), while Experiment 1 contains a smaller error in terms of altitude until the end of the episode, Experiment 2 contains a lot more variance, suggesting that in this case the agent arrives at the target orbit (given in equinoctial elements) through different trajectories, indicating a better understanding of the dynamics than Experiment 1. These results also suggest that shaping the reward to give more importance to the semi-major axis may be needed, since the agents where trained to minimize the error of all equinoctial elements. **We have added this analysis in the revised version**. We hope this addresses you question.
>
> > Q3. Is the MARL support general to all RL algorithms? Or does it only work for PPO?
>
> **MARL support within OrbitZoo is general for all RL algorithms, not just PPO**. This is possible due to the modular architecture shown in Figure 1, where the actual RL algorithm for each satellite is not strictly defined. In fact, when using more complex MARL algorithms that contain centralized training decentralized execution (CTDE) properties (which is common in missions with homogeneous agents), we find that **it is easy to even change the initial architecture that OrbitZoo provides for specific missions*. **Centralizing RL algorithms that contain shared parameters across satellites** (e.g., as seen in IPPO and MAPPO) **is straightforward in OrbitZoo**.
> ___
> ## Weaknesses
>
> > W1. Regarding "Bridging the Reality Gap." it seems it is only supported by the Starlink example, is there any other evidence to show OrbitZoo can bridge the reality gap better than other environments?
>
> Yes. Since we did not find existing RL environments providing quantitative propagation analysis, **we extended our evaluation beyond Starlink data**. We compared the 16-hour propagation Root Mean Squared Error (RMSE), in meters, between OrbitZoo (using Orekit) and real satellite data from multiple sources, including OneWeb. Additional datasets were obtained from the EUROLAS Data Center, representing organizations such as the Deutsches Geodätisches Forschungsinstitut (DGFI), the Joint Center for Earth Systems Technology (JCET), and the NERC Space Geodesy Facility (NSGF):
> Source|Satellite|Date|RMSE
> -|-|-|-
> OneWeb|19|2020-01-01|3945.9
> OneWeb|1|2020-01-01|2540.5
> OneWeb|20|2020-01-01|554.0
> OneWeb|28|2020-01-01|3360.3
> OneWeb|14|2022-01-01|2007.7
> OneWeb|16|2022-01-01|1581.9
> OneWeb|18|2022-01-01|5745.6
> OneWeb|22|2022-01-01|4797.2
> OneWeb|23|2022-01-01|3067.1
> OneWeb|24|2022-01-01|247.5
> OneWeb|42|2024-01-01|1050.8
> OneWeb|39|2024-01-01|1552.8
> OneWeb|30|2024-01-01|5653.4
> OneWeb|21|2024-01-01|4228.9
> OneWeb|17|2024-01-01|3648.1
> OneWeb|15|2024-01-01|1526.9
> OneWeb|46|2024-01-01|1095.0
> OneWeb|48|2024-01-01|4606.5
> OneWeb|20|2030-01-01|702.8
> OneWeb|49|2024-01-01|327.1
> OneWeb|57|2024-01-01|410.4
> DGFI|ajisai|2018-01-11|1801.2
> DGFI|lageos1|2025-06-07|4509.7
> DGFI|lageos2|2025-07-12|3531.3
> JCET|etalon1|2012-05-12|2096.1
> JCET|etalon2|2012-05-12|4624.4
> NSGF|lares|2025-07-26|1553.1
> NSGF|lares2|2025-07-26|5006.4
> NSGF|larets|2025-07-26|1169.8
> NSGF|starlette|2025-07-25|1573.2
> NSGF|stella|2025-07-26|826.6
>
> As shown, **OrbitZoo achieves errors under 1 km after 16 hours of propagation for several tested satellites**. However, satellites beyond the OneWeb constellation are subject to corrective forces (such as empirical accelerations) that Orekit does not model, which likely accounts for some of the observed discrepancies. **We have added this analysis to the revised version.**
>
> > W2. Is there any example of OrbitZoo's visualization capabilities?
>
> Yes. In section 4.2. (in the Interface paragraphs), **we provide four different video links that showcase how OrbitZoo can be used** to provide a clear analysis of systems of bodies and evaluate policies in different kinds of missions. Additionally, there is a link to the anonymized code repository in appendix D (in the end of the introduction) containing a **step-by-step example on how to render a system, together with an “*example_visualization.py*” file**.

---

### Official Review · Reviewer_w3KN · 2025-07-02

**Clarity:** 4
**Significance:** 3
**Originality:** 3
**Rating:** 5
**Confidence:** 4

**Summary:**

This paper introduces OrbitZoo, a novel multi-agent reinforcement learning (MARL) environment for satellite operations grounded in high-fidelity orbital dynamics. Built atop the Orekit library, OrbitZoo supports realistic modeling of gravitational, atmospheric, and third-body forces. It integrates with PettingZoo to provide MARL capabilities and supports interactive 3D visualization. The system is evaluated through tasks such as Hohmann transfers, constellation coordination, and validation against Starlink ephemerides.

**Questions:**

Can the authors provide more quantitative comparisons with existing RL environments or classical control approaches to establish baseline performance?

How scalable is OrbitZoo in multi-agent settings beyond 4–5 satellites in practice?

Could the interface support integration with onboard learning frameworks for spacecraft? Does the environment offer support for simulating noisy sensor models or communication constraints?

What measures are taken to ensure physical plausibility of agent-generated trajectories under unbounded thrust policies?

**Ethical Concerns:**

["NO or VERY MINOR ethics concerns only"]

**Final Justification:**

Discussion complete with the authors

**Limitations:**

The limitations are reasonably well-discussed. However, implications of applying such agents in real satellite systems under uncertainty or adversarial conditions could be elaborated further.

**Quality:**

3

**Strengths And Weaknesses:**

Strengths:
The integration of Orekit ensures industry-grade fidelity in orbital simulation, a clear advancement over simplified environments.
The inclusion of multi-agent support, real-world data validation, and realistic perturbations within a single open-source platform is a strong contribution. Enables reproducible, high-fidelity MARL experimentation in the space domain, with applications in satellite traffic management and autonomous operations. Overall well-written, with informative figures and detailed architectural breakdowns.

Weakness:
The RL training code is not part of the core environment, limiting direct reproducibility.
Experiments are illustrative but not benchmarked against baseline policies or environments quantitatively.
Real-time applications, potential safety concerns, and fail-safe guarantees are not addressed in depth.

---

> ### Author Rebuttal · Authors · 2025-07-30
>
> We deeply appreciate your thorough review, and the recognition of the **importance of integrating high-fidelity dynamics, support for multi-agent learning, and our focus on realism and reproducibility**. We are glad the overall **clarity and potential applications** came through, and we're grateful for your positive assessment.
>
> We will first address your questions, followed by additional concerns as effectively as possible.
> ___
> ## Questions
>
> > Q1. Can the authors provide more quantitative comparisons with existing RL environments or classical control approaches to establish baseline performance?
>
> Yes. Since we found no RL environments offering quantitative propagation analysis, **we extended our evaluation beyond Starlink data**. Specifically, we compared the Root Mean Squared Error (RMSE), in meters, over 16 hours of propagation between OrbitZoo (via Orekit) and real satellite data from various sources, including OneWeb. Additional data came from the EUROLAS Data Center, covering organizations such as the Deutsches Geodätisches Forschungsinstitut (DGFI), the Joint Center for Earth Systems Technology (JCET), and the NERC Space Geodesy Facility (NSGF).
> Source|Satellite|Date|RMSE
> -|-|-|-
> OneWeb|19|2020-01-01|3945.9
> OneWeb|1|2020-01-01|2540.5
> OneWeb|20|2020-01-01|554.0
> OneWeb|28|2020-01-01|3360.3
> OneWeb|14|2022-01-01|2007.7
> OneWeb|16|2022-01-01|1581.9
> OneWeb|18|2022-01-01|5745.6
> OneWeb|22|2022-01-01|4797.2
> OneWeb|23|2022-01-01|3067.1
> OneWeb|24|2022-01-01|247.5
> OneWeb|42|2024-01-01|1050.8
> OneWeb|39|2024-01-01|1552.8
> OneWeb|30|2024-01-01|5653.4
> OneWeb|21|2024-01-01|4228.9
> OneWeb|17|2024-01-01|3648.1
> OneWeb|15|2024-01-01|1526.9
> OneWeb|46|2024-01-01|1095.0
> OneWeb|48|2024-01-01|4606.5
> OneWeb|20|2030-01-01|702.8
> OneWeb|49|2024-01-01|327.1
> OneWeb|57|2024-01-01|410.4
> DGFI|ajisai|2018-01-11|1801.2
> DGFI|lageos1|2025-06-07|4509.7
> DGFI|lageos2|2025-07-12|3531.3
> JCET|etalon1|2012-05-12|2096.1
> JCET|etalon2|2012-05-12|4624.4
> NSGF|lares|2025-07-26|1553.1
> NSGF|lares2|2025-07-26|5006.4
> NSGF|larets|2025-07-26|1169.8
> NSGF|starlette|2025-07-25|1573.2
> NSGF|stella|2025-07-26|826.6
>
> For some tested satellites, OrbitZoo maintains errors below 1 km after 16 hours of propagation. We note that satellites outside the OneWeb constellation experience corrective forces (e.g., empirical accelerations) not modeled by Orekit, which may explain some discrepancies.
>
> In terms of comparing the RL algorithm with a baseline, we note that the classical control solution for the Hohmann maneuver (described in **Appendix D.3.1**) assumes **simplified dynamics and instantaneous thrusts**. Under **more realistic conditions**—such as continuous thrust and additional perturbations—**an analytical solution is no longer feasible**, and the classical approach fails to reach the target orbit.
>
> In contrast, **RL algorithms adapt to these realistic dynamics and can still achieve successful transfers**. To support this claim, we performed a quantitative analysis comparing the fuel usage error (in kg) between the optimal maneuver under simplified dynamics and five successful attempts using the agent trained in both Experiment 1 and Experiment 2 (differences discussed in Table 9), across different levels of realism. The table below reports the mean and standard deviation of the fuel error for each configuration:
>
> Exp 1 No Perturbations|Exp 1 Perturbed|Exp 2 No Perturbations|Exp 2 Perturbed
> -|-|-|-
> $0.34_{(0.02)}$|$0.36_{(0.01)}$|$0.85_{(0.31)}$|$0.86_{(0.16)}$
>
> These values suggest that **both experiments are able to achieve near optimal control while using, on average, no more than 1 kg of additional fuel**, which corresponds to only 1/50th of all the available fuel for maneuvers. Additionally, we compare the altitude error (in km) of those attempts by step:
>
> Step|Exp 1 No Perturbations|Exp 1 Perturbed|Exp 2 No Perturbations|Exp 2 Perturbed
> -|-|-|-|-
> **0**|$0.00_{(0.00)}$|$0.00_{(0.00)}$ | $0.00_{(0.00)}$|$0.00_{(0.00)}$
> 1|$0.01_{(0.00)}$|$0.01_{(0.00)}$|$0.01_{(0.01)}$|$0.01_{(0.00)}$
> 2 |$0.03_{(0.00)}$|$0.03_{(0.00)}$|$0.02_{(0.02)}$|$0.02_{(0.01)}$
> 3 |$0.07_{(0.00)}$|$0.06_{(0.01)}$|$0.04_{(0.04)}$|$0.04_{(0.01)}$
> 4 |$0.11_{(0.01)}$|$0.10_{(0.01)}$|$0.06_{(0.06)}$|$0.06_{(0.01)}$
> ...|...|...|...|...
> 553|$2.57_{(1.51)}$|$10.13_{(1.66)}$|$2.43_{(2.07)}$|$8.10_{(3.60)}$
> 554|$2.61_{(1.51)}$|$10.13_{(1.66)}$|$2.35_{(2.07)}$|$8.07_{(3.58)}$
> 555|$2.65_{(1.51)}$|$10.12_{(1.67)}$|$2.28_{(2.07)}$|$8.04_{(3.55)}$
> ...|...|...|...|...
> **765**|$10.73_{(1.30)}$|$5.10_{(1.12)}$|$17.49_{(2.97)}$|$5.66_{(4.63)}$
> 766|$10.78_{(1.29)}$|$5.05_{(1.12)}$|$17.59_{(2.97)}$|$5.73_{(4.63)}$
> 767|$10.83_{(1.29)}$|$5.00_{(1.11)}$|$17.69_{(2.98)}$|$5.80_{(4.63)}$
> ...|...|...|...|...
> 998|$16.30_{(0.96)}$|$3.74_{(0.75)}$|$30.74_{(3.53)}$|$13.33_{(6.08)}$
> 999|$16.30_{(0.96)}$|$3.76_{(0.75)}$|$30.74_{(3.53)}$|$13.32_{(6.09)}$
>
> The steps in the table correspond to intervals where agents apply thrust (see Figure 19). After the first optimal thrust (step 0), Experiment 2 typically achieves altitudes closer to the target than Experiment 1, though with higher variance. After the second thrust (step 765), Experiment 1 maintains lower altitude error, while Experiment 2 shows more diverse trajectories, suggesting a better grasp of orbital dynamics. These findings also indicate that modifying the reward to place greater emphasis on the semi-major axis could improve performance, as the current shaping balances errors across all equinoctial elements. **We have included this analysis in the revised version.**
>
> > Q2. How scalable is OrbitZoo in multi-agent settings beyond 4–5 satellites in practice?
>
> **OrbitZoo can easily simulate systems with hundreds or thousands of bodies**, with propagation time largely dependent on the desired level of realism. **We address scalability in Appendix C**, where we analyze how propagation time and memory usage scale with the number of bodies. Simulation time generally increases linearly as more bodies are added, assuming the number of active perturbations remains fixed. For large, high-fidelity systems, **OrbitZoo supports parallelization via Orekit**, which can reduce the time per simulation step by several seconds. Users can enable this feature through a JSON configuration option.
>
> > Q3. Could the interface support integration with onboard learning frameworks for spacecraft? Does the environment offer support for simulating noisy sensor models or communication constraints?
>
> Yes. While OrbitZoo is currently intended for ground-based simulation and training, **its modular design allows for adaptation to onboard learning frameworks**. The environment follows standard RL conventions, enabling trained agents to be exported and deployed on embedded systems. Although high-fidelity dynamics are handled by Orekit in the step function, this component can be replaced with simplified or onboard models. In future work, we plan to support reduced-fidelity dynamics with GPU acceleration, enabling fast propagation of large-scale systems for onboard inference or non-control MARL tasks.
>
> OrbitZoo does not yet include built-in models for noisy sensors, but **its architecture supports easily adding such functionality**—an important direction for improving realism. **For communication constraints, OrbitZoo already includes methods for simulating distance and visibility limitations**, including line-of-sight checks that account for Earth’s obstruction, enabling the modeling of scenarios such as communication not only with other satellites, but also with ground-stations by considering them as stationary bodies.
>
> > Q4. What measures are taken to ensure physical plausibility of agent-generated trajectories under unbounded thrust policies?
>
> While unbounded policies are dependent on the specific RL algorithm that is being used (e.g., PPO can have them, but DQN can not), and we assume that developers clip the actual thrust to the action space, **OrbitZoo is prepared to handle unbounded policies in several ways**. The *Satellite.change_thrust()* function expects to receive a thrust in polar parametrization (refer to appendix A.2) for each body with thrusting capabilities. **Magnitudes are validated to have a positive force** (if they are not, it is assumed the agent is not performing any thrust). Since the two remaining dimensions correspond to angles indicating the direction of thrust in a 3D space, any real value is accepted.
> If, by any chance, bodies intersect the central body or orbits become unbounded (hyperbolic trajectory) as a result of bad policies, **Orekit automatically throws an exception that can be handled by the developer**. Additionally, while Orekit does not inherently distinguish between fuel mass and dry mass for propagating bodies (as calculations are only based on wet/total mass), **we provide functions related to levels of fuel that may be used to control unbounded policies** by terminating episodes early (*Satellite.get_fuel()*, *Satellite.has_fuel()*).
> ___
> ## Limitations
>
> > L1. (...) implications of applying such agents in real satellite systems under uncertainty or adversarial conditions could be elaborated further.
>
> We agree that real-world deployment introduces significant challenges. In particular, uncertainty in sensing (as you mentioned), thrust delays, and unmodeled dynamics can degrade agent performance. However, **these uncertainties can all be simulated in OrbitZoo**. Additionally, **adversarial scenarios are naturally supported in OrbitZoo**. For instance, in the MARL experiment we developed, the mission does not perfectly follow a Dec-POMDP framework given that agents contain an adversarial component, where they need to find a balance between individual (orbit) and collective (anomalies) rewards. That is, at each time step they receive different rewards, introducing some level of competition. In the revised version, **we have expanded our discussion in Section 3.2** (L148) to highlight the issue and importance of uncertainty and adversarial training frameworks.

---

> ### Comment · Reviewer_w3KN · 2025-08-01
>
> Thank you to the authors for the comprehensive rebuttal. Your clarifications and additions addressed the majority of my concerns effectively. Below I offer comments on your responses, along with how they affect my assessment:
>
> Q1. Quantitative Comparisons and Baselines
>
> Your extension of the evaluation to include RMSE comparison with real satellite data significantly strengthens the case for OrbitZoo’s physical fidelity. The fuel usage analysis under varying perturbation conditions and comparison to classical Hohmann transfers is insightful. The data on performance under simplified vs. perturbed conditions helps ground the realism and learning capability of the RL agents.
>
> This response improves confidence in the validity of the environment for high-fidelity MARL experimentation. It would be helpful if the final version of the paper explicitly included the fuel/altitude tables and discussed variance more clearly.
>
> Q2. Scalability
>
> The clarification on linear scaling with body count, and the availability of parallelization through Orekit, is encouraging. It would be good to explicitly discuss the tradeoffs users might encounter (e.g., memory vs. fidelity, or computational bottlenecks for 100+ body scenarios) in the final version.
>
> Q3. Interface for Noisy Sensors and Onboard Models
>
> Good to know that OrbitZoo’s modular design supports integration with onboard frameworks and that modeling of line-of-sight constraints is already in place. I recommend emphasizing future work on sensor and comms noise modeling, as that would enhance realism and support broader applications (e.g., autonomous fault recovery, anomaly detection).
>
> Q4. Policy Robustness and Physical Plausibility
>
> Your explanation regarding thrust parametrization and validation logic is thorough. I appreciate the handling of unphysical behavior via Orekit exceptions and early termination hooks (e.g., Satellite.has_fuel()). This reinforces OrbitZoo’s viability as a research testbed without encouraging reward hacking or implausible behaviors.
>
> I commend the extended discussion on uncertainty, adversarial MARL, and simulation of degraded sensing/perception. Framing OrbitZoo as a platform capable of adversarial training is a strong point—especially for domains like SSA and satellite security. I would encourage future work or examples demonstrating robustness under high uncertainty or competitive agents.
>
> I maintain my overall rating of 5 (Accept) and appreciate the thoughtful response. The revised paper would benefit from incorporating the quantitative analysis tables and expanded discussion into the main body rather than only in the appendices.

---

> > ### Author Response · Authors · 2025-08-01
> >
> > Thank you for your continued engagement and supportive follow-up!
> >
> > We’re glad to hear that the **extended evaluation, scalability clarification, and thrust control mechanisms addressed your concerns effectively.**
> >
> > We appreciate your suggestions on further improvements, including:
> >
> > - Moving the fuel and altitude tables to the main paper.
> >
> > - Explicitly discussing variance and computational trade-offs in large-scale scenarios.
> >
> > - Highlighting future extensions on sensor noise and communication degradation.
> >
> > We will incorporate these into the final version. We also fully agree with your point about OrbitZoo enabling **robust, adversarial, and uncertainty-aware training**, particularly relevant for autonomy, Space Situational Awareness (SSA), and fault recovery domains. These are active directions we plan to explore in ongoing work.
> >
> > We sincerely thank you for the thoughtful review and support.

---

### Official Review · Reviewer_paFx · 2025-07-04

**Clarity:** 2
**Significance:** 1
**Originality:** 2
**Rating:** 2
**Confidence:** 2

**Summary:**

This paper introduces OrbitZoo, an environment designed for high-fidelity orbital data generation and RL development. The main contribution of this environment includes high-fidelity orbital data generation built on Python, supporting multi-agent reinforcement learning and modular design for customization and visualization. The authors also present some experiments to validate effectiveness of OrbitZoo.

**Questions:**

See above

**Ethical Concerns:**

["NO or VERY MINOR ethics concerns only"]

**Limitations:**

See above

**Quality:**

2

**Strengths And Weaknesses:**

The paper is well organized and easy to follow. The main issue of this paper to me is that this paper feels more like a technical report on a specialized tool for studying orbital dynamics. There is no general benchmarking results or new development of RL algorithms. As such, NeurIPS is probably not the best venue for this work and its contribution to general ML community is limited. It's likely that a more domain-specific conference would be more appropriate.

---

> ### Author Rebuttal · Authors · 2025-07-30
>
> We thank you for your review, and appreciate your comment on the **organization and clarity shown in the paper**, since orbital dynamics is usually a challenging topic that may require some prior knowledge.
>
> We also appreciate your perspective regarding the positioning of this work. We would like to clarify that OrbitZoo is **not merely a specialized simulator**, but rather an **infrastructure designed to enable a new area of machine learning research in orbital dynamics** (L61-63).
>
> We recognize that NeurIPS provides a dedicated Datasets and Benchmarks track, typically intended for standardized datasets or clearly defined benchmarking suites aimed at algorithm comparison. However, OrbitZoo is fundamentally different: it is neither merely a static dataset nor primarily a standardized benchmarking suite. Instead, **OrbitZoo is a novel, modular RL environment purpose-built to support ML-driven innovation and experimentation specifically within orbital dynamics**—an area demanding high-fidelity modeling, realistic physical constraints, and challenging RL-specific aspects (e.g., multi-agent coordination, partial observability, sparse reward structure, and sim-to-real transfer).
>
> The core motivation behind OrbitZoo lies in **enabling the ML community to define, study, and rigorously test** entirely new classes of RL problems that were previously inaccessible or inadequately explored due to the lack of sufficiently realistic, standardized, and extensible environments. By bridging ML algorithmic innovation with realistic orbital physics through an intuitive MARL interface (PettingZoo), OrbitZoo advances foundational RL research questions rather than solely evaluating established algorithms against fixed benchmarks.
>
> We thus strongly believe OrbitZoo aligns with NeurIPS’s main track objectives of fostering new ML research directions, rather than the more evaluation-focused nature typically associated with the Datasets and Benchmarks track.
>
> If there are remaining questions or concerns regarding OrbitZoo, we are available for clarifications.

---

> > ### Author Response · Authors · 2025-08-05
> > **Follow up**
> >
> > Dear Reviewer paFx,
> >
> > Thank you again for your initial review. We understand that your overall assessment raised a concern about track fit and the nature of the contribution, and we’ve provided a response clarifying our rationale for submitting to the main track, as well as the research-enabling intent of OrbitZoo.
> >
> > Some of the points you raised (e.g., about OrbitZoo being a research enabler rather than a benchmark suite) were also echoed by another reviewer. We’ve since clarified that OrbitZoo is not only grounded in a high-fidelity physical domain, but also introduces RL-specific challenges --- like coordination under constraints and sim-to-real transfer --- which are underrepresented in current benchmarks.
> >
> > If you have any remaining questions or would like further clarification, we would be very happy to engage. We appreciate your time and thank you again for your contribution to the review process.

---

> > ### Author Response · Authors · 2025-08-07
> > **Open to Further Discussion**
> >
> > Dear Reviewer paFx,
> >
> > Thank you again for your initial feedback. We wanted to reiterate that OrbitZoo is not intended as a static benchmark or domain-specific simulator, but as a modular RL environment to support the study of underexplored MARL challenges in physically grounded settings, such as coordination under actuation limits, sparse rewards, and sim-to-real transfer.
> >
> > Several structural revisions have been made to highlight this distinction, and we remain available for any additional clarification or feedback you may have.

---

### Official Review · Reviewer_LSvo · 2025-07-04

**Clarity:** 3
**Significance:** 3
**Originality:** 3
**Rating:** 4
**Confidence:** 4

**Summary:**

The article introduces OrbitZoo, a modular, high-fidelity, and multi-agent reinforcement learning (MARL) environment for orbital dynamics. Built on top of the industry-standard Orekit library and interfaced via Python, OrbitZoo enables realistic satellite simulations, including perturbative forces and customizable thrust models. The framework integrates with the PettingZoo MARL API and provides interactive 3D visualization. Empirical demonstration of its capabilities through various experiments, including single-agent Hohmann transfers, multi-agent constellation coordination, and validation against Starlink ephemeris data. The paper aims to address the lack of standardized, high-fidelity environments for applying reinforcement learning to realistic space scenarios.

**Questions:**

This is a higher number of questions than expected, but these are rather short and well-targeted based on the weaknesses listed above:

1. Related work taxonomy: What principles guided the choice and separation of the four categories in Section 2? Could the classification be reorganized to improve clarity and avoid overlap?
2. Feature specification: The main text lacks a detailed overview of OrbitZoo's configurable parameters and environment structure (e.g., thrust constraints, observation/action spaces, default models). Could this be summarized in a table or example use case?
3. Challenge articulation (Section 3.2): The listed challenges often read as system properties rather than research obstacles. Could the authors reframe these to highlight RL-relevant aspects (e.g., uncertainty, scalability, delayed rewards)?
4. Fidelity-performance trade-off: The choice of Orekit implies significant computational cost. Appendix C benchmarks this, but this critical trade-off is omitted from the main discussion. Could a summary be added to Section 4 or 5?
5. Experiment scope vs. depth: The experiments aim to cover a broad range of missions but are presented succinctly, often without learning curves or ablation details. Are these meant to be illustrative use cases or benchmarkable evaluations? Could this section be reorganised?
6. Federated learning: How is federated learning implemented in OrbitZoo, and what experimental configuration is used in Section 5.2? This concept appears abruptly and is not clearly contextualized.
7. Multi-objective RL: Have the authors considered extending the environment or its reward structure to support multi-objective optimization? This is especially relevant in scenarios involving fuel efficiency, time, and risk trade-offs.
8. Appendix content: Could selected content from Appendix D (e.g., D.2 comparison, D.6 validation) be promoted to the main paper? These elements are essential to support the framework’s claims of realism and generality.

**Ethical Concerns:**

["NO or VERY MINOR ethics concerns only"]

**Final Justification:**

I have no further remarks or questions. While I will maintain the overall score based on the level of contribution, I have increased the clarity score to reflect the enhancements made during the rebuttal phase.

**Limitations:**

The manuscript does not very explicitly discuss limitations or potential risks. While the primary use case is scientific, autonomy in orbital environments raises questions around safety, robustness, and possible misuse. A brief discussion of intended scope, deployment assumptions, or responsible use (e.g., civilian vs. defense) would be appropriate. Scalability to large constellations or real-time applications is also not addressed.

**Paper Formatting Concerns:**

Minor ones:

- Figures 2–4 are hard to read and understand.
- Some acronyms (e.g., DDPG) are used without being defined at first mention.

**Quality:**

3

**Strengths And Weaknesses:**

Strengths:
- The work addresses a significant and timely gap in the reinforcement learning community by providing a standardized, high-fidelity, and publicly accessible environment for satellite operations.
-  OrbitZoo builds upon the validated capabilities of Orekit, ensuring accurate modeling of orbital propagation, including atmospheric drag, solar radiation pressure, and third-body effects.
- The modular architecture, integration with the standard library PettingZoo, and availability of interactive visualization are important assets.
- The validation using real Starlink ephemerides empirically demonstrates the applicability of the simulation engine to real-world scenarios.
- Several experimental case studies illustrate the flexibility of the environment across different classes of orbital control problems, including both single-agent and multi-agent settings.
- The appendices contain a significant number of additional contributions (e.g., computational performance benchmarks, comparison with state-of-the-art environments, detailed mission setups) that could benefit from greater visibility in the main text.

Weaknesses:
- The taxonomy introduced in Section 2 lacks conceptual coherence, mixing levels of abstraction and including overlapping or loosely defined categories.
- Section 3.2 aims to discuss challenges in MARL for orbital dynamics but largely describes system components without framing them as reinforcement learning challenges (e.g., reward sparsity, partial observability, exploration complexity).
- While the use of Orekit introduces realism, the trade-off between physical fidelity and computational cost is insufficiently discussed in the main text. Appendix C provides relevant data that should be included in the main paper.
- Many core features of OrbitZoo, such as agent observation and action space definitions, configurable parameters, and default settings, are only briefly touched upon or deferred to supplementary material.
- Section 5.1 introduces an “RL agent” but omits details about the specific algorithm used. More generally, the empirical results prioritize breadth over depth: while multiple use cases are presented, each is described briefly, limiting insight into policy behavior, convergence, or comparative performance.
- Federated learning appears in Section 5 without prior mention or sufficient explanation of its implementation or relevance within the OrbitZoo framework.
- No discussion is provided on multi-objective formulations, despite their clear relevance for many orbital control problems and knowing the growing importance of MORL and MOMARL.
- Too much of the most valuable technical content, including benchmark comparisons, advanced scenario configurations, and realism validation, is relegated to the appendix, reducing its impact.
- (Detail): Several criteria in Table 1 are undefined (e.g., “continuous control”) and the terminology is inconsistent (e.g., "open-source" vs. "publicly available").

---

> ### Author Rebuttal · Authors · 2025-07-30
>
> We sincerely appreciate your thorough review. We are especially grateful for the recognition of the contributions of OrbitZoo, including its **high-fidelity orbital modeling** via Orekit, integration with PettingZoo to **support both single- and multi-agent reinforcement learning**, and overall **modular architecture**. We also appreciate the positive remarks on the **empirical validation** using real Starlink ephemerides, the **inclusion of diverse case studies**, and the additional resources provided in the appendices. These elements reflect our intent to offer a comprehensive and extensible platform for advancing RL research in the space domain.
>
> Several changes have been made to our paper that reflect some of your concerns, which we will mention first. Then, we will address your questions, and remaining weaknesses and limitations, which are clearly **very relevant** for our work.
> ___
> ## Changes
> ### Change 1
> L70 (Section 2) - Reorganization of the previous four categories into three: (1) Tools for Generating and Simulating Orbital Data; (2) Single-Agent Reinforcement Learning in Orbital Dynamics; (3) Multi-Agent Reinforcement Learning in Orbital Dynamics. We also provide the missing context to the terminologies provided in Table 1.
> ### Change 2
> L148 (Section 3.2) - Although the structure remains unchanged, for each challenge we explicitly state its implication for reinforcement learning, especially in the multi-agent context.
> ### Change 3
> L173 (Section 3.3) - We now address the challenge of exploration in Section 3.2., and change the text to mention the importance of having a modular framework that supports different types of RL algorithms, not just those with continuous action spaces. The ability for OrbitZoo to model multiple objectives (MORL) by using one or several agents through its modular design is now also mentioned in that section (3.2).
> ### Change 4
> L179 (Section 3.4) - We extend our discussion on MARL to include both fully cooperative scenarios (using the Dec-POMDP framework) and partial/fully competitive scenarios (using the MA-POMDP framework), together with the importance of Federated Learning (FL) and Federated RL (FRL) in orbital dynamics in these types of scenarios.
> ### Change 5
> L246 (Section 4.3) - We now include “Realistic Thrust Modeling” into the “Bridging the Reality Gap” challenge, and discuss how scalability challenges (including the trade-off between computational cost and realism) are handled by OrbitZoo in “Multi-Agent Coordination”. Aditionally, we now mention OrbitZoo’s ability to simulate FRL.
> ### Change 6
> L277 (Section 5.1) - In the “Setup” paragraph, we briefly characterize what is the agent (a satellite) and its strategy to apply thrusts (RL algorithm).
> ### Change 7
> L1375 (Appendix H) - We have extended our disscussion in sim-to-real transfer, including real-time applications and integration with existing systems.
> ### Change 8
> We have added a new section "*Framework Architecture and Extensibility*" before Appendix A, focused on the details of the OrbitZoo framework, including classes, methods, and use-cases.
> ___
> ## Questions
>
> > Q1. Related work taxonomy: What principles guided the choice and separation of the four categories in Section 2? Could the classification be reorganized to improve clarity and avoid overlap?
>
> Yes it could, and we thank you for this helpful observation. **We agree that those four categories can be reorganized into three**, as mentioned in the first change. Since OrbitZoo attempts to create a bridge between orbital dynamics and the RL areas, we first focus on tools for generating such data, followed by how this data is commonly used in RL/MARL missions, and finally how there is no standardization due to the nature of the needed fidelity for each mission. While we previously created separate sections to address single- and multi-agent missions, we are of the opinion that merging these sections improves clarity. **[Change 1](#change-1) reflects this concern**.
>
> > Q2. Related work taxonomy: What principles guided the choice and separation of the four categories in Section 2? Could the classification be reorganized to improve clarity and avoid overlap?
>
> Yes, and we thank you for your advice. Although we think that configurable parameters are too extensive to include in the main paper, **we agree that it is useful to add a completely new section in the appendix** that includes the detailed architecture and methods, together with the detailed description of each parameter and some visual use-cases that clearly promote the usage of OrbitZoo. **We wrote this section before appendix A, as stated by [Change 8](#change-8)**.
>
> > Q3. The listed challenges often read as system properties rather than research obstacles. Could the authors reframe these to highlight RL-relevant aspects (e.g., uncertainty, scalability, delayed rewards)?
>
> Yes. According to [Change 2](#change-2), **we have reframed the challenges** to create a connection between orbital dynamics and RL challenges.
>
> > Q4. The choice of Orekit implies significant computational cost. Appendix C benchmarks this, but this critical trade-off is omitted from the main discussion. Could a summary be added to Section 4 or 5?
>
> Definitely. According to [Change 5](#change-5), **we now talk about scalability issues in Section 4.3**.
>
> > Q5. The experiments aim to cover a broad range of missions but are presented succinctly, often without learning curves or ablation details. Are these meant to be illustrative use cases or benchmarkable evaluations? Could this section be reorganised?
>
> These are **illustrative experiments meant to demonstrate the range of scenarios OrbitZoo supports**, rather than benchmark performance. **Some ablations are included** (e.g., reward shaping for the Hohmann maneuver, RL algorithms for the CAM mission), but in-depth comparisons are not the focus. We acknowledge that further analysis (e.g., on observation/action spaces or perturbation levels) would be valuable. To reflect our current focus on multi-agent setups, **we reorganized the experiments into single-agent and multi-agent categories**.
>
> > Q6. How is federated learning implemented in OrbitZoo, and what experimental configuration is used in Section 5.2? This concept appears abruptly and is not clearly contextualized.
>
> **[Change 4](#change-4) and [Change 5](#change-5) reflect your concerns here**. While FL is typically performed on several devices, OrbitZoo can simulate learning in this scenario by applying network constraints, where — applied to orbital dynamics — vertices correspond to satellites and edges to active communication between satellites, and where communication constraints are normally related to distance and visibility between satellites. In the GEO mission, we assume that visibility remains constant throughout the episode, since at every moment each satellite observes the anomalies of all other satellites without constraints. In this scenario, agents are partially cooperating (similar to a MA-POMDP), where they need to find a balance between individual (orbit) and collective (anomalies) goals. **Horizontal FL is applied by simply averaging network parameters after local updates**, which in practice would be done, for example, by a ground station. **More complex RL algorithms** that tackle challenges like heterogeneity between satellites and dynamic observation of agents **can also be developed** and easily implemented and tested in OrbitZoo, as it is currently being done by us.
>
> > Q7. Have the authors considered extending the environment or its reward structure to support multi-objective optimization?
>
> **OrbitZoo already supports multi-objective RL (MORL) due to its reward structure**. In the *vanilla* OrbitZoo (as seen in Figure 1), reward is just a non-implemented standardized function that expects to output a dictionary of values for each agent. The actual value for each agent can be anything — including vectors [1] — which should be properly handled by the developer in the mission they are working on. Because of this, **it is possible to implement MORL in several ways**, from using just one agent with a vector of rewards, to using several agents with scalar rewards. This is a very interesting area of research.
> ___
> ## Weaknesses
> - All identified weaknesses but the fifth ("*Section 5.1 introduces an “RL agent” but omits details about the specific algorithm used.*") are directly addressed in our answers to the questions you posed. [Change 6](#change-6) aims to clarify what "RL agent" means in that context.
> ___
> ##  Limitations
> - **We provide an overview on the potential positive and negative impacts of OrbitZoo on appendix H**, where we highlight the possible misuse and safety concerns that naturally result form learning RL policies in a simulated environment. However, we agree that the issues you mention should be further discussed. OrbitZoo is not optimized for strict real-time execution, but its modular design and the use of Orekit enable integration with real-time systems in principle. **We have extended section H to include these limitations**, as reflected in [Change 7](#change-7);
> - As mentioned in the fourth question, we now address **scalability within OrbitZoo in Section 4.3, in addition to the quantitative evaluation in appendix C**.
> ___
> ## Formatting Concerns
>
> - The text shown in those images (timestamp) is a customizable component of the interface. **We have removed that timestamp from images in Figure 2**, as they are not relevant for what is being shown, and **Figure 4 was changed** for a better understanding of the time evolution;
> - Together with the major changes applied in Section 2 ([Change 1](#change-1)), **we have corrected non-defined acronyms** on first mentions, and simultaneously cited relevant papers for them.
>
> [1] Hayes, C.F., Rădulescu, R., Bargiacchi, E. et al. A practical guide to multi-objective reinforcement learning and planning. Auton Agent Multi-Agent Syst 36, 26 (2022).

---

> > ### Author Response · Authors · 2025-08-05
> > **Follow up**
> >
> > Dear Reviewer LSvo,
> >
> > Thank you again for your detailed and constructive review!
> >
> > We’ve posted a rebuttal addressing all of your questions, including clarifications on the taxonomy, RL challenges, experiment depth, and fidelity-performance trade-offs. We also highlighted the changes made from the appendix to the main text and refined Section 3.2 as suggested.
> >
> > If there is anything that remains unclear or open for discussion, we would be very grateful for a chance to address it.
> >
> > We deeply value your feedback and are glad to see alignment on the potential of OrbitZoo as an RL research enabler.

---

> > ### Comment · Reviewer_LSvo · 2025-08-05
> >
> > Thank you for your detailed answers and the substantial changes made to the revised manuscript. Most of my comments have been adequately addressed either in your responses or through the modifications. Please find my follow-up remarks below:
> >
> > - **Q1 & Change 1:** Thank you for this change and for clarifying why the taxonomy was narrowed down to three categories. I am satisfied with the proposed update.
> >
> > - **Q2 & Change 8:** The addition of the new section in the Appendix is a valuable improvement. I am satisfied with this update.
> >
> > - **Q3:** Addressed satisfactorily.
> >
> > - **Q4:** Addressed satisfactorily.
> >
> > - **Q5:** Thank you for these clarifications and changes. It would be helpful to make it explicit in the manuscript that these are illustrative examples. Regarding the further analysis you acknowledge as valuable, would it be possible to mention in the future work section that such extended analyses are planned?
> >
> > - **Q6 & Changes 4 + 5:** Your arguments and updates are satisfactory.
> >
> > - **Q7:** Thank you for the clarification, and apologies for overlooking this point earlier. As this is an active and interesting research area, it would be beneficial to include a brief note in the paper outlining the current MORL capabilities and perspectives enabled by OrbitZoo.
> >
> > - **Limitations:** The updates in Section 4.3 and in the appendix are satisfactory.
> >
> > - **Formatting concerns:** Addressed satisfactorily.

---

> > > ### Author Response · Authors · 2025-08-05
> > >
> > > Dear Reviewer LSvo,
> > >
> > > Thank you very much for your continued engagement and thoughtful follow-up. We are grateful that most of your concerns have been fully addressed and truly appreciate your careful reading.
> > >
> > > Regarding **Q5**, we will make it explicit in the main text that the experiments are illustrative use cases, and we will add to the conclusion or future work section that extended analysis — including convergence and ablations — is part of our ongoing efforts.
> > >
> > > For **Q7**, we agree that highlighting MORL capabilities is valuable and will include a note outlining the current support and our future plans for multi-objective formulations.
> > >
> > > Thank you again for your careful and constructive feedback throughout!

---

> > > > ### Comment · Reviewer_LSvo · 2025-08-07
> > > >
> > > > Dear Authors,
> > > >
> > > > Thank you for the constructive exchange and the improvements made to the manuscript. While I will maintain the overall score based on the level of contribution, I have increased the clarity score to reflect the enhancements made during the rebuttal phase.

---

> > > > > ### Author Response · Authors · 2025-08-07
> > > > > **Thanks!**
> > > > >
> > > > > Dear Reviewer LSvo,
> > > > >
> > > > > Thank you very much for your follow-up and for acknowledging the clarity improvements! We truly appreciate your engagement throughout the process.
> > > > >
> > > > > In response to the joint reviewer feedback (A1–A4), we have now restructured Section 4 to sharpen the distinction between environment fidelity and RL challenges, and to frame the paper more clearly around enabling research on coordination, constraints, and safety in physically grounded MARL settings. We also revised the conclusion to maintain a measured tone, and repositioned implementation details as reproducibility tools rather than contributions.
> > > > >
> > > > > Your suggestions were instrumental in guiding these changes, and we’re grateful for the clarity and focus they brought to the final version.
> > > > >
> > > > > So, thank you!

---

### Comment · Reviewer_8aNh · 2025-08-07
**More questions to the authors**

Dear authors,

thank you for your extensive replies so far. In a discussion between me and other reviewers, the **following open aspects** have been identified that we would like to communicate to you.

**A1.** We would like to sharpen the distinction between environment fidelity and the RL challenges in the final version.

**A2.** We would like to take care to avoid overselling, in particular in the conclusion, the final version.

**A3.** Parts of your paper lean towards implementation details (e.g., visualization details, configuration mechanisms, etc.) that would be more fit for a technical report or documentation rather than a scientific exposition. We suggest to streamline or reframe these parts in the final version to emphasize the role in enabling reproducible RL experimentation, rather than as primary contributions.

**A4.** While A3 might offer some opportunity to safe space, we would like to see a clearer discussion on this instead: A high-quality promotion of OrbitZoo as MARL environment should identify related research gaps in this environment and performance of a couple of go-to methods that would demonstrate these. More specifically, the paper could very much benefit by further explicitly stating:

* What types of RL challenges are **underexplored** in existing benchmarks (e.g., coordination under physics, fuel constraints, adversarial maneuvers),
* How OrbitZoo creates **unique opportunities** to study them, and
* A **brief benchmark** of a few standard methods with analysis of their limitations in this setting.

At the moment, we see that the current version makes a credible case for OrbitZoo's _potential_ as a research enabler, but t the paper would benefit from a more explicit mapping between its design and the open RL research questions it helps address.


Can you comment on these points?

---

> ### Author Response · Authors · 2025-08-07
> **Revisions based on A1-A4. Thank you for your constructive feedback!**
>
> Thank you for the constructive joint feedback and continued discussion!
> We are glad to see alignment on OrbitZoo’s potential and we welcome the opportunity to strengthen the final version. Below, we address aspects A1–A4 with specific actions we have taken for the camera-ready paper. As the response has a character limit, we will summarize changes, but we can post multiple answers if reviewers need to see the full text that was added.
>
> ---
>
> ## Aspect 1
>
> We reestructured Sec 4, moving lower-level engineering details in 4.1 and 4.2 to new appendices "Simulation and Propagation Settings" and "Available Mission Templates".
>
> The new section 4 is titled **"Reinforcement Learning Challenges in OrbitZoo".**
> The content of the section was changed to meet our objective for the paper. The initial paragraph is the following:
>
> "In the emerging era of autonomous systems operating in the physical world, reinforcement learning is evolving beyond synthetic benchmarks and addressing real-world constraints such as actuation limits, uncertainty, safety, and coordination. OrbitZoo is designed to support this shift. It provides a modular testbed to rigorously investigate RL challenges grounded in high-fidelity dynamics while abstracting just enough to enable algorithmic insight. Below, we outline key research questions that OrbitZoo helps explore."
>
> Then, we go over the RL challenges, as discussed with the reviewers, namely
>
> 1. Sim-to-Real Transfer and Grounded Dynamics.
> 2. Continuous Control and Thrust Constraints.
> 3. Multi-Agent Coordination with Partial Observability.
> 4. Reward Shaping under Sparse and Delayed Feedback.
> 5. Adversarial and Safety-Critical Learning.
>
>
> Finally, we add a paragraph on Visualization and Debugging, which we consider an important practical aspect of RL.
>
> Each subsection describes the challenge, and how OrbitZoo’s abstractions, interfaces, and scenarios concretely support experimentation on it, such as decentralized agents under thrust constraints, or safety-aware policies in adversarial maneuvers.
>
> ---
>
> ## Aspect 2
>
> We appreciate the reminder to maintain a measured tone, as you had previously noted. We added a limitations section, and **we revised the conclusion to better reflect OrbitZoo's intended scope and remove any overstated claims**:
>
> "OrbitZoo provides a modular, high-fidelity environment to support the study of reinforcement learning in realistic orbital settings. While grounded in orbital dynamics, its abstractions and challenge structures are representative of broader autonomy domains. The platform supports reproducible experimentation and enables the exploration of autonomy for future space applications."
>
>
> ---
>
> ## Aspect 3
>
> Thank you for this helpful suggestion. In the revised version, we have reframed implementation-focused content (such as visualization, JSON configuration, wrappers) to emphasize their role in enabling reproducible RL experimentation. Specifically:
> - Visualization is now briefly discussed as a tool to support debugging and interpretability in partially observable, multi-agent missions (see A1).
> - Configuration and modular wrappers are described as part of OrbitZoo’s goal to lower the barrier to RL experimentation in high-fidelity settings — enabling researchers to rapidly define new missions, tune physical realism, and experiment with alternative observability models.
>
> We have moved these details to an appendix (the framework’s architecture, design, core modules, and use cases) as as also noted in our response to Reviewer LSvo (Change 8).
>
> ---
>
> ## Aspect 4
>
> We appreciate this suggestion and agree that more explicit mapping between OrbitZoo’s design and the RL research gaps it helps address, improves the final paper. Our intent has always been to position OrbitZoo as a platform to enable this kind of research, and we now state this more clearly.
>
> In the revised version:
>
> - The new Section 4 (“Reinforcement Learning Challenges in OrbitZoo”) outlines underexplored problems such as coordination with physics-aware constraints, control under thrust and fuel limitations, adversarial and federated multi-agent settings, and sparse or delayed reward shaping. The conjunction of these challenges is specific to RL problems in space especially with strong physics models offered with the Orekit substrate. Nevertheless, models satisfying these challenges are suited to a broader set of problems as discussed previously.
> - We updated Sec 5 to contextualize better the baselines we do present (e.g., PPO and DDPG for maneuver planning) and to comment more directly on their observed limitations in our experiments, such as high sample complexity, variance under perturbations, and partial observability. We are writing a section connecting benchmarks and other environments with the challenges of OrbitZoo, now outside of space domain.
> - We added a summary table at the end of Sec. 4, with a systematic treatment of each Research Gap, How OrbitZoo Enables It, and	Limitations of Standard Methods.

---

> > ### Comment · Reviewer_8aNh · 2025-08-08
> >
> > Dear authors, trusting that you implement all these changes, I have further increased my score to weakly accept. Thank you for your serious effort to address all our critiques.

---

> > > ### Author Response · Authors · 2025-08-08
> > > **Thank you, for helping us improve this paper!**
> > >
> > > Dear Reviewer 8aNh,
> > >
> > > Thank you sincerely for your updated assessment and for the detailed, thoughtful engagement throughout the discussion!
> > >
> > > Your comments and suggestions helped us significantly improve the clarity, focus, and framing of the paper. We are committed to implementing the remaining refinements in the final version as discussed.
> > >
> > > With appreciation,
> > > The authors

---

### Decision · Program_Chairs · 2025-09-17

**Decision:**

Accept (poster)

**Comment:**

The initial reviews were very mixed (ratings 1–5), with concerns expressed as to whether this was more of a technical report than a scientific paper.
After extensive discussion and due to the extensive revisions made by the authors, it is now clear that the revised manuscript OrbitZoo is positioned as a research tool for investigating fundamental RL challenges in physically grounded environments.
With these changes, the work makes a valuable contribution to the RL community by providing a modular, high-precision environment for investigating multi-agent coordination in physically grounded environments.

I agree with the majority of the reviewers and recommend accepting the manuscript.